# Harmonized Cone for Feasible and Non-conflict Directions in Training Physics-Informed Neural Networks

**Dohyun Bu**[*] **Yujung Byun**[*] **Jong-Seok Lee**[†]
Korea Advanced Institute of Science and Technology
{dohyun.bu, yujung4169, jongseok.lee}@kaist.ac.kr

## Abstract

Physics-Informed Neural Networks (PINNs) have emerged as a powerful tool for solving PDEs, yet training is difficult due to a multi-objective loss that couples PDE residuals, initial/boundary conditions, and auxiliary physics terms. Existing remedies often yield infeasible scaling factors or conflicting update directions, resulting in degraded performance. In this paper, we show that training PINNs requires jointly considering feasible scaling factors and a non-conflict direction. Through a geometric analysis of per-loss gradients, we define the *harmonized cone* as the intersection of their primal and dual cones, which characterizes directions that are simultaneously feasible and non-conflicting. Building on this, we propose HARMONIC (HARMONIzed Cone gradient descent), a training procedure that computes updates within the harmonized cone by leveraging the Double Description method to aggregate extreme rays. Theoretically, we establish convergence guarantees in nonconvex settings and prove the existence of a nontrivial harmonized cone. Across standard PDE benchmarks, HARMONIC generally outperforms state-of-the-art methods while ensuring feasible and non-conflict updates.

## 1 Introduction

Partial differential equations (PDEs) provide a foundational framework for modeling phenomena across science and engineering (Morton & Mayers, 2005). However, for many PDEs of practical interest, analytic solutions are seldom available. In practice, numerical methods—such as finite differences (Grossmann et al., 2007), finite elements (Bathe, 2006), and spectral methods (Boyd, 2001)—approximate solutions to the governing equations by discretizing space and time. Such discretizations are prone to error (Papež et al., 2014) and often become unstable on nonlinear or high-dimensional problems (Zhu et al., 2019). Neural networks, with their continuous representations well-suited for automatic differentiation (Baydin et al., 2018; Yu et al., 2018), offer a mesh-free alternative for solving PDEs. Building on this capability, Physics-Informed Neural Networks (PINNs) (Raissi et al., 2019) incorporate the governing equations into the loss function by penalizing PDE residuals while simultaneously enforcing initial and boundary conditions (Karniadakis et al., 2021). As PINNs have gained traction (Raissi et al., 2020; Sahli Costabal et al., 2020; Wang et al., 2020; Kochkov et al., 2021), research has focused on improving their accuracy through strategies such as objective reformulation (Eshaghi et al., 2025), adaptive sampling (Wu et al., 2023), and robust initialization (Lee et al., 2025). In parallel, auxiliary terms—including integral constraints (Yuan et al., 2022) and output-conditioned penalties (Yu et al., 2022)—have been incorporated, giving rise to additional loss components that more faithfully encode complex PDE-driven dynamics.

Even with this strong advantage, incorporating multiple losses inevitably complicates PINN training. PDE residuals and initial/boundary conditions are enforced simultaneously, and this multi-loss formulation often produces imbalanced gradients during backpropagation—arising from numerical stiffness (Wang et al., 2021), mismatched convergence rates across different terms (Wang et al., 2022), and incompatible optimal solutions among losses (Lin et al., 2019). To address these challenges, two main streams of research have emerged. The first focuses on reweighting, which seeks

---

[*]Equal contribution.
[†]Corresponding author.

to minimize a non-negative weighted sum of loss terms by adaptively adjusting their relative magnitudes (Xiang et al., 2022; Bischof & Kraus, 2025). While this alleviates scale-induced bias, it fails to account for scenarios where gradients from different losses *conflict* in their directions. Since PINNs require all loss components to be driven simultaneously toward zero, this multi-loss formulation can also be understood in terms of multi objectives being optimized concurrently. From this perspective, the second pursues multi-objective optimization (MOO), which directly enforces non-conflict updates by ensuring nonnegative inner products with all loss-specific gradients (Hwang & Lim, 2024; Liu et al., 2025). However, focusing solely on non-conflict often produces an *infeasible* direction that cannot be realized as the gradient of a reweighted objective with non-negative coefficients, as formally defined in Section 3.1. Although several methods have attempted to integrate both perspectives, there is still no consensus on an optimal strategy for combining reweighting and MOO.

To show the necessity of considering both *infeasible* and *conflict*, we illustrate the bi-objective toy example in Fig. 1, defined by $L_1$ and $L_2$ over the parameter vector $\boldsymbol{\theta}:=(\theta_1, \theta_2)$, a setting commonly used in the MOO literature (Liu et al., 2021a; Senushkin et al., 2023). When updating solely along the conic combination $\boldsymbol{g}_{\mathrm{cc}}=\lambda_1 \nabla_{\boldsymbol{\theta}} L_1 + \lambda_2 \nabla_{\boldsymbol{\theta}} L_2$ with equal weights $\lambda_1=\lambda_2=1$, the update is dominated by the larger-magnitude term and may converge to a suboptimal point. Conversely, when updating only along a non-conflict direction $\boldsymbol{g}_{\mathrm{cf}}$ satisfying $\nabla_{\boldsymbol{\theta}} L_j^\top \cdot \boldsymbol{g}_{\mathrm{cf}}=1$ for $j=1,2$, the optimization avoids direct conflicts. Yet it converges prematurely to a point outside the Pareto front, since $\boldsymbol{g}_{\mathrm{cf}}$ can correspond to an *infeasible* direction. As shown in the right panel of Fig. 1, combining the two principles—requiring the update to lie in (i) the feasible region and (ii) the non-conflict region defined by the loss gradients—guides the trajectory toward a Pareto-optimal solution shared by both losses.

Given the necessity of avoiding both *infeasible* and *conflict*, we propose a gradient-based method using the **HARMONI**zed **C**one gradient descent (**HARMONIC**). Concretely, the harmonized cone is defined as the intersection of the primal and dual cones of the per-loss gradients, such that any update direction within this region can be expressed as a conic combination of the loss gradients and simultaneously has nonnegative inner products with all of them.

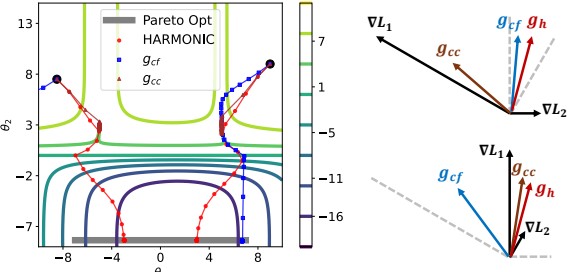

Figure 1: Visualization of optimization trajectories with respect to the objective values and the update directions, where each arrow represents the update direction under specific loss-gradient conditions, and - - - indicates the line orthogonal to the loss gradients. Initialization points are indicated by ●, and the Pareto front is shown in ▬. ▬ corresponds to $\boldsymbol{g}_{cc}$, which converges to a suboptimal point. ▬ denotes $\boldsymbol{g}_{cf}$, where one trajectory reaches the Pareto front but becomes over-emphasized toward $L_1$, while the other diverges in the opposite direction away from the Pareto front. ▬ represents **HARMONIC** ($\boldsymbol{g}_h$), with all trajectories successfully converging to the Pareto front.

This work is the first to provide a geometric analysis that explicitly characterizes both the scaling factors of the losses and the directions of the corresponding gradients, offering a principled understanding of infeasible and conflict in general multi-loss settings. Furthermore, we establish theoretical convergence of **HARMONIC**, showing that the proposed intersection retains the desirable Pareto-stationary convergence behavior. **HARMONIC** unifies reweighting and non-conflict optimization without extra asymptotic cost. Experiments on the PINNacle benchmark (Zhongkai et al., 2024) show that it generally outperforms state-of-the-art methods within this unified framework.

## 2 RELATED WORKS

**Reweighting Strategies.** A central line of research aims to alleviate the issue of imbalanced gradients through adaptive loss weighting. LRA (Wang et al., 2021) rescales individual losses based on gradient statistics, while GradNorm (Chen et al., 2018) introduces trainable scaling factors that balance the relative rate of loss reduction with respect to their initial magnitudes. SoftAdapt (Heydari et al., 2019) dynamically adjusts weights by measuring the per-loss change between consecutive

Table 1: Comparison of representative approaches against four criteria: applicability to *Multiple* loss terms (at least three), guarantee of *Feasible* update gradients with respect to all loss terms, guarantee of *Non-Conflict* update directions with respect to all loss terms, and guarantee of *Non-Degenerate* scaling factors for all loss terms (✓ = satisfied, ✗ = not satisfied).

| Method Class | Multiple | Feasible | Non-Conflict | Non-Degenerate |
|---|:---:|:---:|:---:|:---:|
| LRA, NTK, ReLoBRaLo | ✓ | ✓ | ✗ | ✗ |
| MGDA | ✓ | ✓ | ✓ | ✗ |
| PCGrad, CAGrad, IMTL-G | ✓ | ✗ | ✗ | ✗ |
| Aligned-MTL, ConFIG | ✓ | ✗ | ✓ | ✗ |
| DCGD (Center) | ✗ | ✓ | ✓ | ✓ |
| **HARMONIC** | ✓ | ✓ | ✓ | ✓ |

iterations. Recently, ReLoBRaLo (Bischof & Kraus, 2025) integrates these strategies into a unified framework, achieving robust performance across PDEs. In parallel, leveraging eigenvalue of Neural Tangent Kernel (NTK) (Wang et al., 2022), they dynamically update the loss weights to balance the average convergence rates among losses.

**MOO Perspectives.** Recent advances in multi-task learning (MTL) recast gradient-based updates through the lens of multi-objective optimization. MGDA (Sener & Koltun, 2018) seeks the minimum-norm vector inside the convex hull of task gradients. PCGrad (Yu et al., 2020) reduces conflicts by projecting pairwise gradients before aggregation. CAGrad (Liu et al., 2021a) maximizes the minimum loss decrease across tasks, bounded within a Euclidean ball around the mean gradient. IMTL-G (Liu et al., 2021b) aims to balance task gradients with respect to both their directions and magnitudes. Aligned-MTL (Senushkin et al., 2023) enforces a spectral lower bound on gradient alignment. These perspectives have also been extended to PINNs (Zhou et al., 2023), which is made possible by the fact that each loss term in PINNs is required to approach zero simultaneously. Consequently, this motivates the use of MOO-based gradient strategies in the PINN setting. ConFIG (Liu et al., 2025) constructs a pseudo-inverse-based update that guarantees strictly positive inner products with all gradients and equal projection lengths across losses. DCGD (Hwang & Lim, 2024) introduces a geometric analysis of the mean gradient, aiming to balance not only directional conflicts but also magnitude disparities

**Limitations.** Table 1 summarizes the limitations of existing approaches. Reweighting strategies do not explicitly account for gradient conflicts, and thus may result in conflicting update directions. MGDA guarantees non-conflicting updates, but it does not enforce non-degenerate scaling factors for all loss terms, which can result in some gradients being entirely suppressed. PCGrad, CAGrad, and IMTL-G attempt to mitigate such sacrifices by heuristically modifying gradient directions, yet they provide no guarantees. Aligned-MTL and ConFIG further ensure non-conflict for all loss terms, but their updates may fall outside the feasible region. DCGD satisfies the last three criteria but is difficult to extend beyond two losses. To the best of our knowledge, prior work has not simultaneously addressed all four desiderata.

## 3 PRELIMINARIES

### 3.1 PRIMAL AND DUAL GRADIENT CONE

**Notation.** Let $m, d$ denote the number of loss functions and the dimension of the parameter vector $\boldsymbol{\theta}$, respectively, with $d \gg m$. For each $j=1,\ldots,m$, define the loss gradient $\boldsymbol{g}_j := \nabla_{\boldsymbol{\theta}} L_j \in \mathbb{R}^d$. Collecting these, we form the gradient matrix $\boldsymbol{G} := [\boldsymbol{g}_1, \ldots, \boldsymbol{g}_m] \in \mathbb{R}^{d \times m}$, which we assume to be of full rank. Define matrix $\boldsymbol{D} := [\boldsymbol{g}_{\neg 1}^-, \ldots, \boldsymbol{g}_{\neg m}^-] \in \mathbb{R}^{d \times m}$ as the Moore–Penrose pseudo-inverse of $\boldsymbol{G}^\top$, and let $\mathcal{A}(\boldsymbol{G}) \in \mathbb{R}^d$ denote the update direction. Additionally, $\boldsymbol{0}_m$ and $\boldsymbol{1}_m$ denote the $m$-dimensional vectors whose entries are all 0 and all 1, respectively.

**Primal Gradient Cone Region.** A primal gradient cone $\mathbb{K}$ is generated by $\boldsymbol{G}$ (Vertex representation, V-representation) and simultaneously represented by $\boldsymbol{D}$ (Half-space representation, H-representation).

**Definition 1** (Primal Gradient Cone). *We define the primal gradient cone as*

$$\mathbb{K} := \{\boldsymbol{G}\boldsymbol{\lambda} \mid \boldsymbol{\lambda} \in \mathbb{R}_+^m\},$$

*or equivalently*

$$\mathbb{K} := \{\boldsymbol{x} \in \mathbb{R}^d \mid \boldsymbol{D}^\top \boldsymbol{x} \geq \boldsymbol{0}_m\},$$

*where $\mathbb{R}_+^m$ denotes the $m$-dimensional nonnegative orthant.*

When $\mathcal{A}(\boldsymbol{G})$ lies within $\mathbb{K}$, it can be written as $\mathcal{A}(\boldsymbol{G}) = \nabla_{\boldsymbol{\theta}}\left(\sum_j \lambda_j L_j\right)$ for weights satisfying $\lambda_j \geq 0$ for all $j$, which defines the *feasible* region. Conversely, if $\mathcal{A}(\boldsymbol{G})$ lies outside $\mathbb{K}$, some weights become negative, leading to an *infeasible* region. The detailed proof is provided in the Appendix A.1.

**Dual Gradient Cone Region.** A dual gradient cone $\mathbb{K}^*$ is defined as the set of vectors whose inner products with all $\boldsymbol{g}_j$ are nonnegative.

**Definition 2** (Dual Gradient Cone). *The dual gradient cone of $\mathbb{K}$ is defined as*

$$\mathbb{K}^* := \{\boldsymbol{y} \in \mathbb{R}^d \mid \boldsymbol{G}^\top \boldsymbol{y} \geq \boldsymbol{0}_m\},$$

*or equivalently*

$$\mathbb{K}^* := \{\boldsymbol{D}\boldsymbol{w} \mid \boldsymbol{w} \in \mathbb{R}_+^m\}.$$

If $\mathcal{A}(\boldsymbol{G})$ is inside $\mathbb{K}^*$, $\mathcal{A}(\boldsymbol{G})$ ensures that none of the loss terms increase, defining the *non-conflict* region. Conversely, when $\mathcal{A}(\boldsymbol{G})$ falls outside $\mathbb{K}^*$, some loss inevitably increases, leading to a *conflict* region. The detailed proof is provided in the Appendix A.2.

### 3.2 EMPIRICAL OBSERVATIONS AND ISSUES IN TRAINING PINNS

**Limitations of Single-Aspect Strategies.** As noted in Section 2, prior approaches have mainly relied on either the reweighting or the non-conflict strategies. Using the one-dimensional Burgers equation, Fig. 2 shows training curves and test MSE, highlighting the drawbacks of considering only one of these aspects. Fig. 2a shows that when reweighting update direction $\mathcal{A}_{cc}(\boldsymbol{G}) := \frac{1}{m}\sum_{j=1}^m \boldsymbol{g}_j/\|\boldsymbol{g}_j\|$ falls outside the dual cone $\mathbb{K}^*$, the test MSE starts to increase. This indicates that *conflict* arises, causing one of the training losses to increase as the update is aligned opposite to its corresponding loss gradient. In contrast, Fig. 2b demonstrates that although all training losses decrease, the test MSE rises whenever *non-conflict* update direction $\mathcal{A}_{cf}(\boldsymbol{G})$, with $\boldsymbol{G}^\top \mathcal{A}_{cf}(\boldsymbol{G}) = \boldsymbol{1}_m$, leaves the primal cone $\mathbb{K}$. This indicates that even when *conflict* is mitigated, *infeasible* directions lead to degraded test performance.

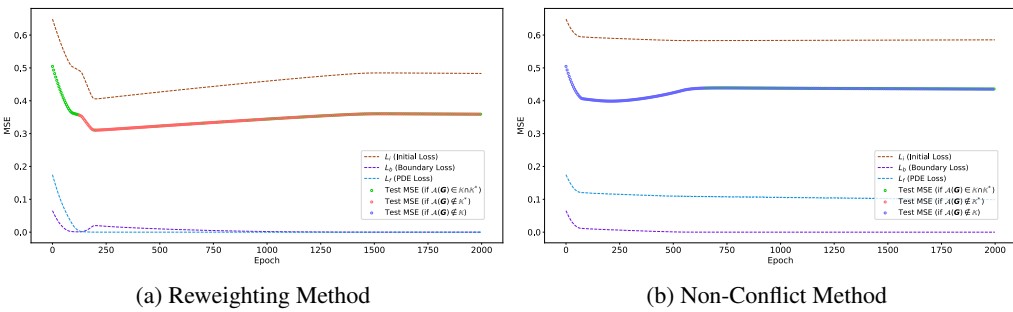

(a) Reweighting Method          (b) Non-Conflict Method

Figure 2: Training curves for the initial loss $L_i$, boundary loss $L_b$, PDE residual loss $L_f$, and test MSE for one-dimensional Burgers equation. --- denotes the $L_i$ trajectory, --- denotes the $L_b$ trajectory, --- denotes the $L_f$ trajectory, ○ indicates the test MSE when the update follows a feasible and non-conflict direction, ○ indicates the test MSE when the update follows a conflict direction, and ○ indicates the test MSE when the update follows an infeasible direction.

**Infeasible and Conflict Directions Exposed by Cosine Values.** To further investigate these challenges in training PINNs, we record the cosine values between $\mathcal{A}(\boldsymbol{G})$ and each $\boldsymbol{g}_{\neg j}^-$ (Fig. 3a, 3b), as well as with $\boldsymbol{g}_j$ (Fig. 3c, 3d), for all $j$ throughout training. As shown in Fig. 3a and Fig. 3c, under the reweighting method, the cosine values between the PDE loss gradient and $\mathcal{A}_{cc}(\boldsymbol{G})$ are frequently close to 1, indicating that the updates are often biased toward the PDE loss. In particular, Fig. 3c shows that more than half of the cosine values between the initial loss gradient and $\mathcal{A}_{cc}(\boldsymbol{G})$ are negative, suggesting that, from the perspective of the initial loss, the updates are often made in a deteriorating direction. For the non-conflict method, Fig. 3d shows that the cosine values between

$\mathcal{A}_{cf}(\boldsymbol{G})$ and both the PDE loss gradient and the initial loss gradient are mostly close to $0$, indicating relatively less improvement compared to the boundary loss. In Fig. 3b, the cosine values between $\mathcal{A}_{cf}(\boldsymbol{G})$ and the extreme rays of $\mathbb{K}^*$ corresponding to the initial and PDE loss gradients are negative, indicating that $\mathcal{A}_{cf}(\boldsymbol{G})$ moves in an *infeasible* direction.

## 4 METHODOLOGY

In this section, we define a *harmonized cone*, the region that is simultaneously *feasible* and *non-conflict*, and present a geometric analysis considering both the scaling factors and directions of all loss gradients. Then, we introduce our algorithm, which ensures that $\mathcal{A}(\boldsymbol{G})$ remains inside this region.

### 4.1 HARMONIZED CONE REGION

A harmonized cone $\mathbb{H}$ is defined as the set of vectors that have nonnegative inner product values with both $\boldsymbol{g}_j$ and $\boldsymbol{g}_{\neg j}^-$ for all $j$.

**Definition 3** (Harmonized Cone). *The set*

$$\mathbb{H} := \{\boldsymbol{z} \in \mathbb{R}^d \mid \boldsymbol{G}^\top \boldsymbol{z} \geq \boldsymbol{0}_m, \boldsymbol{D}^\top \boldsymbol{z} \geq \boldsymbol{0}_m\},$$

*or equivalently*

$$\mathbb{H} := \{\boldsymbol{R}\boldsymbol{\alpha} \mid \boldsymbol{\alpha} \in \mathbb{R}_+^p\},$$

*is called the* harmonized cone *of $\boldsymbol{G}$, where $\boldsymbol{R} \in \mathbb{R}^{d \times p}$ denotes the generating matrix of $\mathbb{H}$.*

When $\mathcal{A}(\boldsymbol{G})$ lies within $\mathbb{H}$, it is both *feasible* and *non-conflict*. Conversely, if $\mathcal{A}(\boldsymbol{G}) \notin \mathbb{H}$, then it is either *infeasible*, which may lead to degraded test performance, or one of the losses inevitably increases even with sufficiently small step sizes due to the *conflict*. The detailed proof is provided in the Appendix B.1.

Theorem 1 establishes the necessary and sufficient conditions for $\mathcal{A}(\boldsymbol{G})$ to remain inside $\mathbb{H}$, expressed in terms of the relative magnitudes between the gradients of all loss terms. In addition, Theorem 2 provides the same characterization from a dual perspective.

**Theorem 1.** *Suppose that the gradient matrix at iteration $t$ is denoted by $\boldsymbol{G}^{(t)}$, and let $\mathbb{H}^{(t)}$ be the harmonized cone induced by $\boldsymbol{G}^{(t)}$. Define $\boldsymbol{\lambda}^{(t)} = [\lambda_1^{(t)}, \ldots, \lambda_m^{(t)}]^\top \in \mathbb{R}_+^m$ as a scaling vector, where each component $\lambda_j$ is the scaling factor corresponding to the loss $L_j$. Then, a feasible update direction $\boldsymbol{G}^{(t)}\boldsymbol{\lambda}^{(t)} \in \mathbb{H}^{(t)}$ if and only if*

$$\boldsymbol{G}^{(t)\top}\boldsymbol{G}^{(t)}\boldsymbol{\lambda}^{(t)} \geq \boldsymbol{0}_m.$$

**Theorem 2.** *Suppose that the Moore–Penrose pseudo-inverse of transposed gradient matrix at iteration $t$ is denoted by $\boldsymbol{D}^{(t)}$, and let $\mathbb{H}^{(t)}$ be the harmonized cone induced by $\boldsymbol{D}^{(t)}$. Define $\boldsymbol{w}^{(t)} = [w_1^{(t)}, \ldots, w_m^{(t)}]^\top \in \mathbb{R}_+^m$ as a scaling vector, where each component $w_j$ is the scaling factor corresponding to $\boldsymbol{g}_{\neg j}^-$. Then, a non-conflict update direction $\boldsymbol{D}^{(t)}\boldsymbol{w}^{(t)} \in \mathbb{H}^{(t)}$ if and only if*

$$\boldsymbol{D}^{(t)\top}\boldsymbol{D}^{(t)}\boldsymbol{w}^{(t)} \geq \boldsymbol{0}_m.$$

Theorems 1 and 2 establish clear criteria for when reweighting or non-conflict can lead to adverse training in PINNs, with detailed proofs provided in the Appendix B.2, B.3. Together, these results highlight the *harmonized cone* as a unifying concept that integrates reweighting and non-conflict approaches.

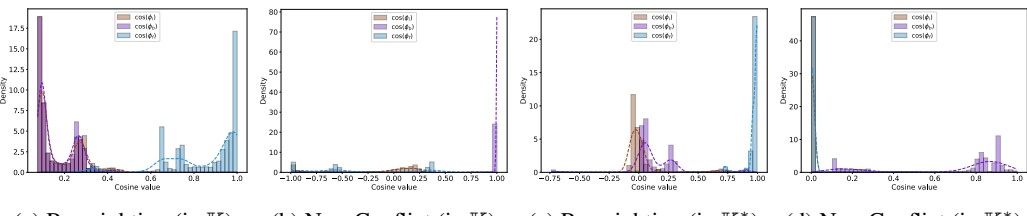

(a) Reweighting (in $\mathbb{K}$)    (b) Non-Conflict (in $\mathbb{K}$)    (c) Reweighting (in $\mathbb{K}^*$)    (d) Non-Conflict (in $\mathbb{K}^*$)

Figure 3: Histograms of cosine values $\cos(\phi_j)$ between $\mathcal{A}(\boldsymbol{G})$ and $\boldsymbol{g}_{\neg j}^-$, and between $\mathcal{A}(\boldsymbol{G})$ and $\boldsymbol{g}_j$.

---

**Algorithm 1** HARMONIZed Cone gradient descent (HARMONIC)

---

**Require:** $\boldsymbol{G}^{(t)} \in \mathbb{R}^{d \times m}$, $\eta^{(t)}$ (learning rate), $\boldsymbol{\theta}^{(t)}$ (network parameters).

1: **Form H-representation** (Theorem 1): $\boldsymbol{A}^{(t)} \leftarrow \begin{bmatrix} \boldsymbol{I}_m \\ \boldsymbol{G}^{(t)\top} \boldsymbol{G}^{(t)} / \|\boldsymbol{G}\|^2 \end{bmatrix}$.

2: **Double Description (H→V)**: compute the V-representation of $\{\boldsymbol{\lambda} \in \mathbb{R}^m \mid \boldsymbol{A}^{(t)} \boldsymbol{\lambda} \geq \boldsymbol{0}_m\}$:
   Initialize $\mathcal{R}^{(0)} = \{\boldsymbol{e}_1, \ldots, \boldsymbol{e}_m\}$, where $\boldsymbol{e}_j$ is the $j$-th standard basis vector.
   For each constraint row $\boldsymbol{a}_\ell^\top$ of $\boldsymbol{G}^{(t)\top} \boldsymbol{G}^{(t)} / \|\boldsymbol{G}^{(t)}\|^2$:
      Partition $\mathcal{R}^{(\ell-1)}$ into $(\mathcal{R}_\ell^+, \mathcal{R}_\ell^0, \mathcal{R}_\ell^-)$.
      Update $\mathcal{R}^{(\ell)} \leftarrow \mathcal{R}_\ell^+ \cup \mathcal{R}_\ell^0 \cup \{(\boldsymbol{a}_\ell^\top \boldsymbol{\pi}^+)\boldsymbol{\pi}^- - (\boldsymbol{a}_\ell^\top \boldsymbol{\pi}^-)\boldsymbol{\pi}^+ \mid \boldsymbol{\pi}^+ \in \mathcal{R}_\ell^+, \ \boldsymbol{\pi}^- \in \mathcal{R}_\ell^-\}$.
   Remove duplicates to obtain $\boldsymbol{\Pi}^{(t)} = [\boldsymbol{\pi}_1^{(t)}, \ldots, \boldsymbol{\pi}_p^{(t)}] \in \mathbb{R}^{m \times p}$.

3: **Map rays to parameter space**: for each $j=1, \ldots, p$, set $\boldsymbol{r}_j^{(t)} \leftarrow \boldsymbol{G}^{(t)} \boldsymbol{\pi}_j^{(t)} \in \mathbb{R}^d$.

4: **Normalize directions**: Set $\boldsymbol{d}^{(t)} \leftarrow \sum_{j=1}^p \boldsymbol{r}_j^{(t)} / \|\boldsymbol{r}_j^{(t)}\|$ and $\hat{\boldsymbol{d}}^{(t)} \leftarrow \boldsymbol{d}^{(t)} / \|\boldsymbol{d}^{(t)}\|$.

5: **Aggregate rays**: set $\mathcal{A}_h(\boldsymbol{G}^{(t)}) \leftarrow (\boldsymbol{1}_m^\top \boldsymbol{G}^{(t)\top} \hat{\boldsymbol{d}}^{(t)}) \hat{\boldsymbol{d}}^{(t)}$.

6: **Update**: $\boldsymbol{\theta}^{(t+1)} \leftarrow \boldsymbol{\theta}^{(t)} - \eta^{(t)} \mathcal{A}_h(\boldsymbol{G}^{(t)})$

---

## 4.2 HARMONIZED CONE GRADIENT DESCENT

With the theoretical foundation in Section 4.1, we now propose HARMONIZed Cone gradient descent (HARMONIC), which ensures that $\mathcal{A}(\boldsymbol{G})$ remains inside $\mathbb{H}$. A representation matrix $\boldsymbol{A}^{(t)}$ with respect to $\boldsymbol{\lambda}^{(t)}$ is derived from $\boldsymbol{\lambda}^{(t)} \geq \boldsymbol{0}_m$ and $\boldsymbol{G}^{(t)\top} \boldsymbol{G}^{(t)} \boldsymbol{\lambda}^{(t)} \geq \boldsymbol{0}_m$. Through the Double Description Method (Motzkin et al., 1953), the H-representation $\boldsymbol{A}^{(t)}$ is converted into the V-representation, given by the generating matrix $\boldsymbol{\Pi}^{(t)} = [\boldsymbol{\pi}_1^{(t)}, \ldots, \boldsymbol{\pi}_p^{(t)}] \in \mathbb{R}^{m \times p}$. Specifically, for each constraint row $\boldsymbol{a}_\ell^\top$ of $\boldsymbol{G}^{(t)\top} \boldsymbol{G}^{(t)} / \|\boldsymbol{G}^{(t)}\|^2$, the current ray set is partitioned into $\mathcal{R}_\ell^+ = \{\boldsymbol{\pi} \mid \boldsymbol{a}_\ell^\top \boldsymbol{\pi} > 0\}$, $\mathcal{R}_\ell^0 = \{\boldsymbol{\pi} \mid \boldsymbol{a}_\ell^\top \boldsymbol{\pi} = 0\}$, and $\mathcal{R}_\ell^- = \{\boldsymbol{\pi} \mid \boldsymbol{a}_\ell^\top \boldsymbol{\pi} < 0\}$, and new candidate rays are generated from all $(\boldsymbol{\pi}^+, \boldsymbol{\pi}^-) \in \mathcal{R}_\ell^+ \times \mathcal{R}_\ell^-$ via $(\boldsymbol{a}_\ell^\top \boldsymbol{\pi}^+) \boldsymbol{\pi}^- - (\boldsymbol{a}_\ell^\top \boldsymbol{\pi}^-) \boldsymbol{\pi}^+$. The resulting candidate is then pruned to obtain the final generator matrix $\boldsymbol{\Pi}^{(t)}$. Each $\boldsymbol{G}^{(t)} \boldsymbol{\pi}_j$ is an extreme ray of $\mathbb{H}^{(t)}$ and corresponds to a column vector of $\boldsymbol{R}$ in Definition 3. Finally, the update $\mathcal{A}_h(\boldsymbol{G}^{(t)})$ is obtained by averaging and normalizing the extreme rays of $\mathbb{H}^{(t)}$ to form a unified direction, and the direction is then scaled by the sum of the inner products with all loss gradients. The proposed method is summarized in Algorithm 1.

## 4.3 COMPARISON WITH RESPECT TO EXISTING APPROACHES

In this section, we illustrate how HARMONIC updates its search direction in comparison with the other approaches. We then examine a toy example in which we visualize the trajectory converging toward the Pareto front and compare it with the trajectories generated by the other methods. Finally, we demonstrate the advantages of HARMONIC through additional illustrative examples, building on the same experimental setting applied to the other methods in Section 3.2

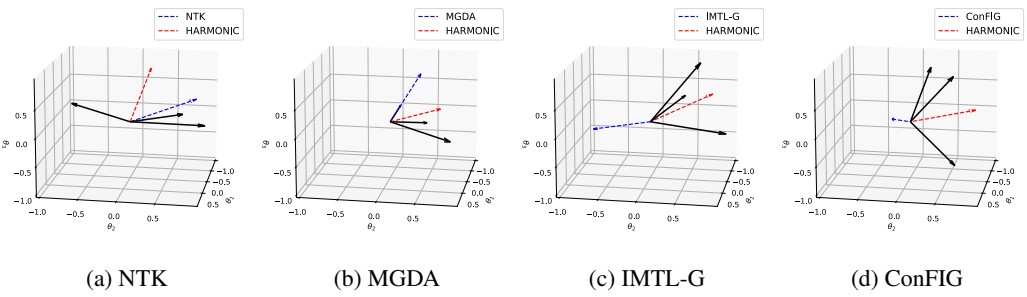

(a) NTK      (b) MGDA      (c) IMTL-G      (d) ConFIG

Figure 4: Unsatisfied cases of the existing methods. $\longrightarrow$ denotes the gradient of each loss. $\dashrightarrow$ represents $\mathcal{A}(\boldsymbol{G})$ of the existing methods. $\dashrightarrow$ represents $\mathcal{A}(\boldsymbol{G})$ of HARMONIC.

Fig. 4 visualizes, in a three-dimensional parameter space with three loss gradients, how $\mathcal{A}(\boldsymbol{G})$ is selected under each of the four methods in comparison with HARMONIC. We compare HARMONIC with NTK (Wang et al., 2022), MGDA (Sener & Koltun, 2018), IMTL-G (Liu et al., 2021b), and ConFIG (Liu et al., 2025). From Fig. 4a, NTK can lead to updates that lie within the primal gradient cone $\mathbb{K}$, i.e., the *feasible* region constructed by the loss gradients. However, such updates may still incur *conflict*, as the update direction can form an obtuse angle with some of the individual gradients. On the other hand, as shown in Fig. 4b, MGDA produces updates that are not only *feasible* but also *non-conflict*. Yet, when the inner product between any pair of loss gradients is nonnegative, the update direction is biased toward the gradient with the smallest magnitude. In such cases, some loss terms may fail to decrease, since the update direction can align almost exclusively with the gradient of the smallest magnitude. IMTL-G, illustrated in Fig. 4c, may also result in updates that lie in an *infeasible* region, potentially inducing *conflict*. Finally, as shown in Fig. 4d, ConFIG ensures *non-conflict* updates; however, the resulting direction may fall outside the *feasible* region. Further details and discussions are provided in Appendix E.1.

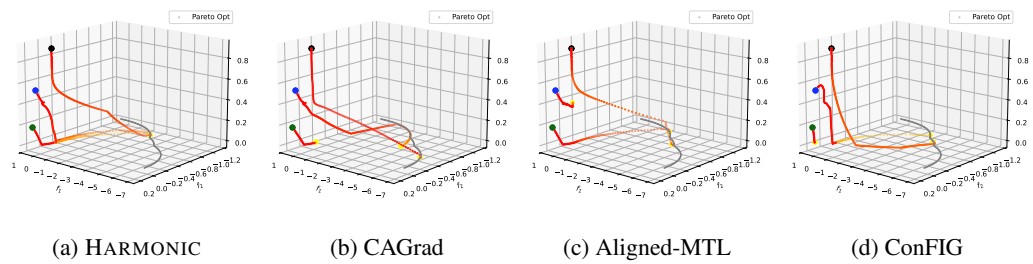

| (a) HARMONIC | (b) CAGrad | (c) Aligned-MTL | (d) ConFIG |

Figure 5: Visualization of optimization trajectories for the toy example. The starting points of the trajectories are denoted by ● (0.8, 0.8, 0.5, 0.5), ● (0.3, 0.9, 0.2, 0.8), and ● (0.1, 0.8, 0.1, 0.7). And the trajectories are shown fading from ● (start) to ● (end). The Pareto front is given by ●.

We construct a toy example in a four-dimensional parameter space to evaluate whether recent MOO methods can successfully converge to the Pareto front for a three-objective minimization task. Comparing against recent MOO algorithms such as CAGrad (Liu et al., 2021a), Aligned-MTL (Senushkin et al., 2023), ConFIG, we evaluate HARMONIC under the same conditions. All methods are initialized from the same three starting points, and the optimization trajectories are visualized over 20,000 steps with a learning rate of 0.001. Details of the experimental setup are provided in Appendix E.2.

As shown in Fig. 5, HARMONIC achieves convergence to the Pareto front from all initial points. In contrast, since CAGrad does not guarantee non-conflict updates, its iterates are pulled toward the mean-loss gradient even after reaching the Pareto front, eventually converging to regions dominated by a single objective. On the other hand, the trajectories of Aligned-MTL and ConFIG, both of which enforce non-conflicting updates, converge stably once they reach the Pareto front. Yet their updates can occasionally step outside the feasible region, causing premature convergence before the Pareto front for certain initializations.

Taken together, these comparisons highlight that HARMONIC resolves the challenges faced by existing methods across all loss gradient scenarios. In Fig. 6a, we empirically demonstrate that updates with HARMONIC decrease all training losses without degrading test performance. Furthermore, Fig. 6b, 6c show that, for all $j$, both $\boldsymbol{g}_j$ and $\boldsymbol{g}_{\neg j}^-$ have strictly positive cosine values with $\mathcal{A}(\boldsymbol{G})$, which is not only a *feasible* but also an *non-conflict* direction, indicating that $\mathcal{A}(\boldsymbol{G})$ always lies in the interior of both $\mathbb{K}$ and $\mathbb{K}^*$.

## 4.4 THEORETICAL ANALYSES

We show that HARMONIC can converge to a Pareto stationary point even in possibly nonconvex settings, and theoretically establish that the intersection of $\mathbb{K}$ and $\mathbb{K}^*$ always exists, excluding the trivial solution.

**Theorem 3.** *Let $L_1, L_2, \ldots, L_m$ be differentiable and possibly nonconvex objectives, and assume that the total gradient $\sum_{j=1}^{m} \nabla_{\boldsymbol{\theta}} L_j$ is Lipschitz continuous with constant $\mu > 0$. Consider updates along HARMONIC, denoted by $\mathcal{A}_h(\boldsymbol{G})$, with step size $\eta \leq \frac{2}{\mu}$, where $M > 0$ denotes the minimum co-*

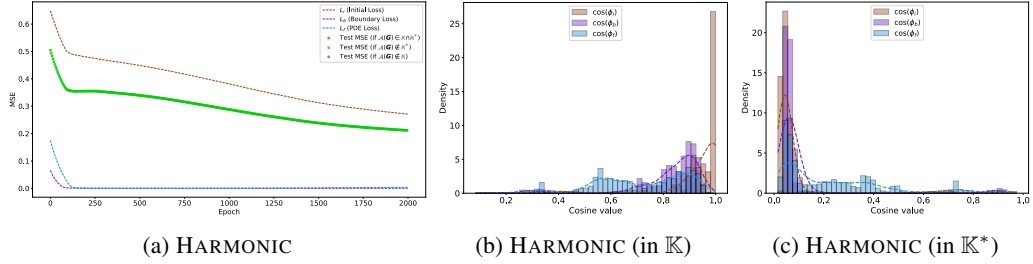

(a) HARMONIC        (b) HARMONIC (in $\mathbb{K}$)        (c) HARMONIC (in $\mathbb{K}^*$)

Figure 6: (left) Training curve (a) and histograms of cosine values $\cos(\phi)$ between $\mathcal{A}(G)$ and $g_{\neg j}^-$ (b), and between $\mathcal{A}(G)$ and $g_j$ (c).

*sine value between $\mathcal{A}_h(G)$ and each $g_j$. Then the sequence either converges to a Pareto-stationary point, or, after $T$ iterations, satisfies the bound*

$$\frac{1}{T}\sum_{t=0}^{T-1}\|\sum_{j=1}^{m}\nabla_{\boldsymbol{\theta}}L_j^{(t)}\|^2 \ \leq\ \frac{2}{\eta M T}\sum_{j=1}^{m}\big(L_j^{(0)}-L_j^{(T)}\big),$$

*where $T$ denotes the total number of iterations.*

Theorem 3 states that HARMONIC converges to a stationary point at a rate of $\mathcal{O}(1/\sqrt{T})$ in the nonconvex setting. The detailed proof of the theorem is provided in the Appendix B.4.

**Theorem 4.** *Let the convex gradient hull be defined as $\mathbb{U} = \big\{\, G\boldsymbol{\lambda} \in \mathbb{R}^d \ \big|\ \boldsymbol{\lambda} \geq \mathbf{0}_m,\ \mathbf{1}_m^\top\boldsymbol{\lambda} = 1 \,\big\}$, which is contained in the primal gradient cone $\mathbb{K}$. Then there always exists a vector*

$$\boldsymbol{u}^\star \ =\ \underset{\boldsymbol{u}\in\mathbb{U}}{\arg\min}\ \|\boldsymbol{u}\|$$

*such that $\|\boldsymbol{u}^\star\| > 0$. Consequently, $\boldsymbol{u}^\star$ belongs to $\mathbb{H}$, and hence $\mathbb{H}$ containing $\boldsymbol{u}^\star$ necessarily exists.*

This guarantees that HARMONIC can always operate within a nontrivial harmonized cone, ensuring its applicability across all training scenarios. Additional details and proofs are deferred to Appendix B.5.

## 5 EXPERIMENTS

In this section, we compare HARMONIC with various methods, including MultiAdam (Yao et al., 2023), LRA (Wang et al., 2021), ReLoBRaLo (Bischof & Kraus, 2025), MGDA (Sener & Koltun, 2018), PCGrad (Yu et al., 2020), CAGrad (Liu et al., 2021a), IMTL-G (Liu et al., 2021b), Aligned-MTL (Senushkin et al., 2023), and ConFIG (Liu et al., 2025).

We evaluate performance on the PINNacle benchmark (Zhongkai et al., 2024) for PDEs and the A-PINN (Yuan et al., 2022) benchmark for the integro-differential equation (IDE). In these experiments, we consider five benchmark equations: popular PDEs (Wave1d-C, Poisson2d-C), a high-dimensional PDE (HNd), an IDE (Volterra1d), and inverse problem (HInv). All experimental settings follow PINNacle, except that we use a network architecture with 50 neurons and 3 layers, trained for 50,000 iterations. All methods are evaluated under two learning rate settings (0.001, and 0.0001), and we report the best performance for each method. Details of the experimental settings and additional results on more equations are provided in Appendix F, G and H.1.

**Comparision on Benchmark Equations.** In Table 2, we highlight **the best** and the second best methods based on the mean over five random seeds. The numbers of objectives for each benchmark are: 4 (Wave1d-C, Volterra1d), and 3 (Poisson2d-C, HNd, HInv). All objectives are treated as independent losses. Experimental results demonstrate that HARMONIC generally outperforms all benchmarks. Consistent with prior empirical findings—namely, that DCGD (Hwang & Lim, 2024) (reporting results with two losses) and ConFIG (Liu et al., 2025) (reporting results with three losses) achieve performance gains through non-conflict strategies—our experiments confirm that non-conflict-based methods outperform reweighting approaches not only with three but also with four loss terms. In particular, for Poisson2d-C, all methods except HARMONIC exhibit poor performance, whereas HARMONIC achieves substantially superior performance.

Table 2: Average relative $L^2$ errors (with standard deviations) estimated across 5 random seeds for each algorithm.

| Method | Wave1d-C | Poisson2d-C | HNd | Volterra1d | HInv |
|---|---|---|---|---|---|
| MultiAdam | 1.0958 (0.1004) | 0.6733 (0.0820) | 0.0040 (0.0028) | 0.0004 (0.0002) | 0.1446 (0.0245) |
| LRA | 0.2608 (0.0542) | 0.5712 (0.1595) | 0.0020 (0.0011) | 0.0070 (0.0065) | 0.0546 (0.0131) |
| ReLoBRaLo | 0.3542 (0.0089) | 0.6602 (0.0221) | **0.0004 (0.0000)** | **0.0002 (0.0000)** | 1.5999 (0.0576) |
| MGDA | 0.3995 (0.0135) | 0.6777 (0.0239) | 0.0009 (0.0001) | 0.0011 (0.0019) | 2.4224 (1.6765) |
| PCGrad | 0.2256 (0.0701) | 0.6918 (0.0190) | 0.0005 (0.0002) | 0.0004 (0.0004) | 0.0660 (0.0152) |
| CAGrad | 0.1186 (0.0071) | 0.6969 (0.0197) | 0.0005 (0.0001) | 0.0014 (0.0024) | 0.6435 (1.0862) |
| IMTL-G | 0.2724 (0.3372) | 0.5039 (0.0684) | 0.0006 (0.0001) | 0.0735 (0.0989) | 0.0498 (0.0075) |
| Aligned-MTL | 0.2302 (0.0514) | 0.2847 (0.2363) | 0.0006 (0.0002) | 0.0037 (0.0044) | 0.0804 (0.0457) |
| ConFIG | 0.0668 (0.0279) | 0.6856 (0.0274) | 0.0006 (0.0001) | 0.0018 (0.0024) | 0.0466 (0.0068) |
| HARMONIC | **0.0655 (0.0293)** | **0.0214 (0.0179)** | 0.0005 (0.0000) | 0.0003 (0.0001) | **0.0461 (0.0098)** |

**Mitigating Unsatisfied Cases.** We conducted additional experiments to assess the effect of constraining $\mathcal{A}(\boldsymbol{G})$ to lie inside $\mathbb{H}$. In Fig. 7, the left panels of Fig. 7a and Fig. 7b visualize, for ReLoBRaLo and ConFIG respectively on Poisson2d-C, whether $\mathcal{A}(\boldsymbol{G})$ enters $\mathbb{H}$ during training. In both cases, the test relative $L^2$ error decreases at first but then remains nearly constant beyond a certain point, indicating that the training has settled into an undesirable suboptimal point. By contrast, the right panels of Fig. 7a and Fig. 7b start from the same initialization but, whenever the baseline update would drive $\mathcal{A}(\boldsymbol{G})$ outside $\mathbb{H}$, we instead apply HARMONIC to keep the update inside $\mathbb{H}$. Invoking Theorem 1, 2, we mathematically certify whether $\mathcal{A}(\boldsymbol{G})$ lies inside or outside $\mathbb{H}$ and, based on this certification, choose the update rule used to form $\mathcal{A}(\boldsymbol{G})$. Under this intervention, the test relative $L^2$ error converges close to zero, indicating that optimization does not stall at a suboptimal point but instead progresses toward a more desirable minimum.

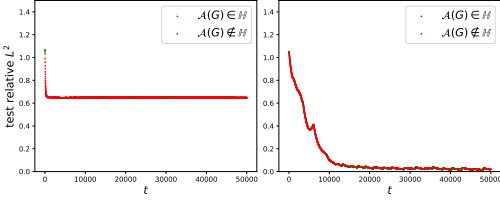 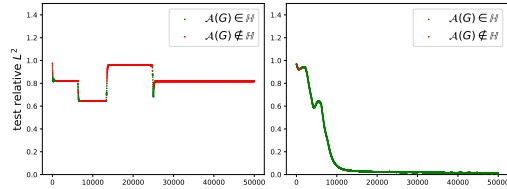

(a) (left) ReLoBRaLo only (right) with HARMONIC     (b) (left) ConFIG only (right) with HARMONIC

Figure 7: Training curve for Poisson2d-C, where • denotes $\mathcal{A}(\boldsymbol{G}) \in \mathbb{H}$ and • denotes $\mathcal{A}(\boldsymbol{G}) \notin \mathbb{H}$.

We then extend the visualization experiment to the same settings as Table 2. For each baseline method (ReLoBRaLo, ConFIG, and CAGrad), whenever $\mathcal{A}(\mathbf{G}) \notin \mathbb{H}$, we enforce the update to remain inside $\mathbb{H}$ by applying HARMONIC. We denote the resulting variants as $\mathbb{H}$-ReLoBRaLo, $\mathbb{H}$-CAGrad, and $\mathbb{H}$-ConFIG, and their performance is reported in Table 3. The results in Table 3 show that these variants consistently outperform their original counterparts. In particular, all methods demonstrate performance improvements on the benchmark equations, with especially substantial gains observed for Poisson2d-C.

Table 3: Average relative $L^2$ errors (with standard deviations) estimated across 5 random seeds for each algorithm. Red-colored entries indicate results that are better than their counterparts.

| Method | Wave1d-C | Poisson2d-C | HNd | Volterra1d | HInv |
|---|---|---|---|---|---|
| $\mathbb{H}$-ReLoBRaLo | 0.3274 (0.0392) | 0.0948 (0.1763) | 0.0030 (0.0056) | 0.0179 (0.0351) | 0.0467 (0.0126) |
| $\mathbb{H}$-CAGrad | 0.1265 (0.0463) | 0.0312 (0.0122) | 0.0015 (0.0024) | 0.0007 (0.0004) | 0.0518 (0.0070) |
| $\mathbb{H}$-ConFIG | 0.0407 (0.0149) | 0.0094 (0.0022) | 0.0006 (0.0002) | 0.0064 (0.0096) | 0.0558 (0.0124) |

**Computational Cost.** Table 4 reports the average seconds per 100 epochs for each method across the benchmark datasets. Across all benchmarks, MGDA and CAGrad consistently required longer runtimes, as each epoch involves an iterative optimization procedure. By comparison, HARMONIC, despite enforcing both the *feasible* and *non-conflict* conditions, is generally faster, or at least maintains a computational cost comparable to ConFIG. In particular, computing the V-representation of the $2m \times m$ matrix in HARMONIC is practical in terms of wall-clock time. Appendix H.3 further confirms that HARMONIC scales similarly even with eight loss components.

Table 4: Average computational costs (in seconds) with standard deviations from 100 epochs across 100 repeats. **The worst** methods are highlighted.

| Method | Wave1d-C | Poisson2d-C | HNd | Volterra1d | HInv |
|---|---|---|---|---|---|
| MultiAdam | 1.53 (0.06) | 1.08 (0.07) | 5.29 (0.01) | 1.03 (0.05) | 2.36 (0.01) |
| LRA | 1.39 (0.05) | 1.03 (0.04) | 5.46 (0.03) | 1.10 (0.05) | 2.61 (0.05) |
| ReLoBRaLo | 0.51 (0.01) | 0.46 (0.02) | 2.24 (0.01) | 0.41 (0.02) | 1.09 (0.02) |
| MGDA | **3.19 (0.52)** | **3.00 (0.53)** | 6.40 (0.32) | **4.01 (1.38)** | 3.13 (0.13) |
| PCGrad | 1.58 (0.07) | 1.16 (0.08) | 5.40 (0.02) | 1.15 (0.05) | 2.48 (0.03) |
| CAGrad | 2.48 (0.24) | 1.90 (0.14) | **6.41 (0.11)** | 1.91 (0.21) | **3.45 (0.13)** |
| IMTL-G | 1.56 (0.06) | 1.09 (0.05) | 5.34 (0.01) | 1.02 (0.05) | 2.41 (0.02) |
| Aligned-MTL | 1.49 (0.07) | 0.97 (0.03) | 5.35 (0.01) | 1.01 (0.05) | 2.42 (0.02) |
| ConFIG | 1.54 (0.05) | 1.12 (0.06) | 5.39 (0.03) | 1.06 (0.04) | 2.44 (0.03) |
| HARMONIC | 1.56 (0.07) | 1.03 (0.07) | 5.35 (0.01) | 1.05 (0.03) | 2.42 (0.02) |

## 6 CONCLUSIONS

In this work, we proposed the concept of the harmonized cone to address challenges that arise from multiple losses in the training of PINNs. Defined as the intersection of the primal and dual cones of per-loss gradients, the harmonized cone characterizes a region that simultaneously guarantees feasible and non-conflict. Building on this geometric analysis, we introduced HARMONIC, a gradient-based method that leverages the Double Description method to aggregate extreme rays, thereby enforcing that the update direction remains within the interior of the harmonized cone. From a theoretical perspective, we established the existence of a nontrivial harmonized cone and proved that HARMONIC converges to Pareto-stationary solutions in nonconvex settings. Empirical results on the PDEs and IDE benchmarks demonstrate that HARMONIC consistently achieves either superior performance compared to the state-of-the-art baselines. Furthermore, we showed that HARMONIC incurs no substantial computational overhead compared to the existing methods.

For future work, a compelling direction is to incorporate objective preferences, as studied in MOO (Chen et al., 2024), to guide update directions within the harmonized cone. Another promising direction is to extend the framework beyond first-order optimization and explore second-order methods for PINNs (Wang et al., 2025).

## 7 REPRODUCIBILITY STATEMENT

The proofs of the main results are provided in Appendix A and B. Details on how the figures in the main text were generated are described in Appendix C, D, and E. Appendix F provides complete descriptions of the benchmarks used in our experiments as well as the experimental settings, while Appendix G presents the implementation details of the baseline methods considered for comparison. In addition, we provide the benchmark data files and fully reproducible code in the supplementary material.

## 8 ACKNOWLEDGEMENTS

This research was supported in part by the National Research Foundation of Korea grant funded by the Korea government (MSIT) (Grant number RS-2020-NR049544, RS-2022-NR068758, and RS-2024-00361377) and in part by the Korea Evaluation Institute of Industrial Technology grant funded by the Korea government (MOTIR) (Grant number RS-2025-25458052).

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

## A    Proofs for Section 3

### A.1    Detail proof for Definition 1

**Definition 1**    (Primal Gradient Cone). We define the primal gradient cone as

$$\mathbb{K} := \{\boldsymbol{G\lambda} \mid \boldsymbol{\lambda} \in \mathbb{R}_+^m\}, \tag{1}$$

or equivalently

$$\mathbb{K} := \{\boldsymbol{x} \in \mathbb{R}^d \mid \boldsymbol{D}^\top \boldsymbol{x} \geq \boldsymbol{0}_m\}, \tag{2}$$

where $\mathbb{R}_+^m$ denotes the $m$-dimensional nonnegative orthant.

*Proof.* A cone $\mathbb{K}$ is generated by conic combinations of finitely many column vectors of the V-representation $\boldsymbol{G}$.

**Theorem 5** (Weyl's Theorem). *Any nonempty finitely generated cone is finitely constrained.*

By Theorem 5, the cone $\mathbb{K}$ can also be expressed in terms of an H-representation $\boldsymbol{H}$ with finitely many columns.

**Corollary 1** (Weyl Duality). *Given matrix $\boldsymbol{G} : d \times m$, consider $\mathbb{K} = \{\boldsymbol{G\lambda} \mid \boldsymbol{\lambda} \in \mathbb{R}_+^m\}, \mathbb{L} = \{\boldsymbol{x} \in \mathbb{R}^d \mid \boldsymbol{G}^\top \boldsymbol{x} \geq \boldsymbol{0}_m\}$. Then $\mathbb{K}^* = \mathbb{L}, \mathbb{L}^* = \mathbb{K}$, where * denotes dual relationship.*

By Corollary 1, $\boldsymbol{H}$ corresponds to the V-representation of the dual cone $\mathbb{K}^*$.

Consider $\boldsymbol{G}^\top$ and its pseudo-inverse $\boldsymbol{D}$. Since $\boldsymbol{G}$ has full rank, we have

$$\boldsymbol{G}^\top \boldsymbol{D} = \boldsymbol{I}_m. \tag{3}$$

This implies

$$\boldsymbol{g}_k \cdot \boldsymbol{g}_{\neg j}^- = 1, \qquad\qquad \text{if } k = j, \tag{4}$$

$$\boldsymbol{g}_k \cdot \boldsymbol{g}_{\neg j}^- = 0, \qquad\qquad \text{if } k \neq j. \tag{5}$$

This is equivalent to the process of activating all constraints, except the one associated with the $j$-th column of $\boldsymbol{G}$, to yield the extreme ray. Hence, each $\boldsymbol{g}_{\neg j}^-$ is an extreme ray of the dual cone $\mathbb{K}^*$.

Consequently, $\boldsymbol{D}$ can be regarded as one of the valid H-representations of $\mathbb{K}$, and we may equivalently write

$$\mathbb{K} := \{\boldsymbol{x} \in \mathbb{R}^d \mid \boldsymbol{D}^\top \boldsymbol{x} \geq \boldsymbol{0}_m\}. \tag{6}$$

□

### A.2    Detail proof for Definition 2

**Definition 2**    (Dual Gradient Cone). The dual gradient cone of $\mathbb{K}$ is defined as

$$\mathbb{K}^* := \{\boldsymbol{y} \in \mathbb{R}^d \mid \boldsymbol{G}^\top \boldsymbol{y} \geq \boldsymbol{0}_m\}, \tag{7}$$

or equivalently

$$\mathbb{K}^* := \{\boldsymbol{Dw} \mid \boldsymbol{w} \in \mathbb{R}_+^m\}. \tag{8}$$

*Proof.* By definition, the dual cone $\mathbb{K}^*$ can be represented in the H-representation by finitely many column vectors of $\boldsymbol{G}$.

**Theorem 6** (Minkowski's Theorem). *Any polyhedral cone is nonempty and finitely generated.*

By Theorem 6, the inequality representation

$$\{\boldsymbol{y} \in \mathbb{R}^d \mid \boldsymbol{G}^\top \boldsymbol{y} \geq \boldsymbol{0}_m\} \tag{9}$$

admits an equivalent V-representation $\boldsymbol{V}$ with finitely many columns.

By Corollary 1, $\boldsymbol{V}$ corresponds to the H-representation of the primal cone $\mathbb{K}$. Furthermore, from the result of Proof A.1, since $\boldsymbol{D}$ is an H-representation of $\mathbb{K}$, it can equivalently be regarded as a valid V-representation of $\mathbb{K}^*$. □

# B    PROOFS FOR SECTION 4

## B.1    DETAIL PROOF FOR DEFINITION 3

**Definition 3**    (Harmonized Cone). The set

$$\mathbb{H} := \{\boldsymbol{z} \in \mathbb{R}^d \mid \boldsymbol{G}^\top \boldsymbol{z} \geq \boldsymbol{0}_m, \boldsymbol{D}^\top \boldsymbol{z} \geq \boldsymbol{0}_m\}, \tag{10}$$

or equivalently

$$\mathbb{H} := \{\boldsymbol{R}\boldsymbol{\alpha} \mid \boldsymbol{\alpha} \in \mathbb{R}_+^p\}, \tag{11}$$

is called the *harmonized cone* of $\boldsymbol{G}$, where $\boldsymbol{R} \in \mathbb{R}^{d \times p}$ denotes the generating matrix of $\mathbb{H}$.

*Proof.*  By definition, the harmonized cone $\mathbb{H}$ is the intersection of $\mathbb{K}$ and $\mathbb{K}^*$.

Since $\mathbb{H}$ must simultaneously satisfy the constraints given by the H-representations of both $\mathbb{K}$ and $\mathbb{K}^*$, it can be written as

$$\{\boldsymbol{z} \in \mathbb{R}^d \mid \boldsymbol{G}^\top \boldsymbol{z} \geq \boldsymbol{0}_m, \ \boldsymbol{D}^\top \boldsymbol{z} \geq \boldsymbol{0}_m\}. \tag{12}$$

By Theorem 6, the inequality representation admits an equivalent V-representation $\boldsymbol{R}$ with finitely many columns. □

## B.2    DETAIL PROOF FOR THEOREM 1

**Theorem 1.**    Suppose that the gradient matrix at iteration $t$ is denoted by $\boldsymbol{G}^{(t)}$, and let $\mathbb{H}^{(t)}$ be the harmonized cone induced by $\boldsymbol{G}^{(t)}$. Define $\boldsymbol{\lambda}^{(t)}{=}[\lambda_1^{(t)}, \dots, \lambda_m^{(t)}]^\top \in \mathbb{R}_+^m$ as a scaling vector, where each component $\lambda_j$ is the scaling factor corresponding to the loss $L_j$. Then, a *feasible* update direction $\boldsymbol{G}^{(t)}\boldsymbol{\lambda}^{(t)} \in \mathbb{H}^{(t)}$ if and only if

$$\boldsymbol{G}^{(t)\top} \boldsymbol{G}^{(t)} \boldsymbol{\lambda}^{(t)} \geq \boldsymbol{0}_m. \tag{13}$$

*Proof.*  A *feasible* update direction $\boldsymbol{G}^{(t)}\boldsymbol{\lambda}^{(t)}$, which lies in the primal cone $\mathbb{K}$, must satisfy Definition 3 in order to belong to the harmonized cone.

If we denote $\boldsymbol{z} = \boldsymbol{G}^{(t)}\boldsymbol{\lambda}^{(t)}$, then it must hold that

$$\boldsymbol{G}^{(t)\top} \boldsymbol{G}^{(t)} \boldsymbol{\lambda}^{(t)} \geq \boldsymbol{0}_m, \tag{14}$$

$$\boldsymbol{D}^{(t)\top} \boldsymbol{G}^{(t)} \boldsymbol{\lambda}^{(t)} \geq \boldsymbol{0}_m. \tag{15}$$

Since $\boldsymbol{D}^{(t)\top}\boldsymbol{G}^{(t)}{=}\boldsymbol{I}_m$, for $\boldsymbol{\lambda}^{(t)}$ defined in $\mathbb{R}_+^m$, the condition reduces to requiring $\boldsymbol{G}^{(t)\top}\boldsymbol{G}^{(t)}\boldsymbol{\lambda}^{(t)} \geq \boldsymbol{0}_m$. Thus, $\boldsymbol{G}^{(t)}\boldsymbol{\lambda}^{(t)}$ belongs to the harmonized cone if and only if this condition is satisfied. □

## B.3    DETAIL PROOF FOR THEOREM 2

**Theorem 2.**    Suppose that the Moore–Penrose pseudo-inverse of transposed gradient matrix at iteration $t$ is denoted by $\boldsymbol{D}^{(t)}$, and let $\mathbb{H}^{(t)}$ be the harmonized cone induced by $\boldsymbol{D}^{(t)}$. Define $\boldsymbol{w}^{(t)}{=}[w_1^{(t)}, \dots, w_m^{(t)}]^\top \in \mathbb{R}_+^m$ as a scaling vector, where each component $w_j$ is the scaling factor corresponding to $\boldsymbol{g}_{\neg j}^-$. Then, a *non-conflict* update direction $\boldsymbol{D}^{(t)}\boldsymbol{w}^{(t)} \in \mathbb{H}^{(t)}$ if and only if

$$\boldsymbol{D}^{(t)\top} \boldsymbol{D}^{(t)} \boldsymbol{w}^{(t)} \geq \boldsymbol{0}_m. \tag{16}$$

*Proof.*  A *non-conflict* update direction $\boldsymbol{D}^{(t)}\boldsymbol{w}^{(t)}$, which lies in the dual cone $\mathbb{K}^*$, must satisfy Definition 3 in order to belong to the harmonized cone.

If we denote $\boldsymbol{z} = \boldsymbol{D}^{(t)}\boldsymbol{w}^{(t)}$, then it must hold that

$$\boldsymbol{G}^{(t)\top} \boldsymbol{D}^{(t)} \boldsymbol{w}^{(t)} \geq \boldsymbol{0}_m, \tag{17}$$

$$\boldsymbol{D}^{(t)\top} \boldsymbol{D}^{(t)} \boldsymbol{w}^{(t)} \geq \boldsymbol{0}_m. \tag{18}$$

Since $\boldsymbol{G}^{(t)\top}\boldsymbol{D}^{(t)}{=}\boldsymbol{I}_m$, for $\boldsymbol{w}^{(t)}$ defined in $\mathbb{R}_+^m$, the condition reduces to requiring $\boldsymbol{D}^{(t)\top}\boldsymbol{D}^{(t)}\boldsymbol{w}^{(t)} \geq \boldsymbol{0}_m$. Thus, $\boldsymbol{D}^{(t)}\boldsymbol{w}^{(t)}$ belongs to the harmonized cone if and only if this condition is satisfied. □

### B.4 DETAIL PROOF FOR THEOREM 3

**Theorem 3.** Let $L_1, L_2, \ldots, L_m$ be differentiable and possibly nonconvex objectives, and assume that the total gradient $\sum_{j=1}^{m} \nabla_{\boldsymbol{\theta}} L_j$ is Lipschitz continuous with constant $\mu > 0$. Consider updates along HARMONIC, denoted by $\mathcal{A}_h(\boldsymbol{G})$, with step size $\eta \leq \frac{2}{\mu}$, where $M > 0$ denotes the minimum cosine value between $\mathcal{A}_h(\boldsymbol{G})$ and each $\boldsymbol{g}_j$. Then the sequence either converges to a Pareto-stationary point, or, after $T$ iterations, satisfies the bound

$$\frac{1}{T} \sum_{t=0}^{T-1} \|\sum_{j=1}^{m} \nabla_{\boldsymbol{\theta}} L_j^{(t)}\|^2 \leq \frac{2}{\eta M T} \sum_{j=1}^{m} (L_j^{(0)} - L_j^{(T)}), \tag{19}$$

where $T$ denotes the total number of iterations.

*Proof.* By the descent lemma (Nesterov, 2018), an update along $\mathcal{A}_h(\boldsymbol{G}^{(t)})$ with a step size $\eta \leq \frac{2}{\mu}$ gives us

$$\sum_{j=1}^{m} L_j^{(t+1)} \leq \sum_{j=1}^{m} L_j^{(t)} - \frac{\eta}{2} \|\mathcal{A}_h(\boldsymbol{G}^{(t)})\|^2, \tag{20}$$

where the superscript indicates the optimization iteration index.

Recall from Algorithm 1 that $\hat{\boldsymbol{d}}^{(t)} := \boldsymbol{d}^{(t)}/\|\boldsymbol{d}^{(t)}\|$ satisfies $\|\hat{\boldsymbol{d}}^{(t)}\| = 1$ and $\hat{\boldsymbol{d}}^{(t)} \in \mathbb{H}^{(t)} \subset \mathbb{K}^{*(t)}$, so that $\boldsymbol{g}_j^{(t)\top} \hat{\boldsymbol{d}}^{(t)} > 0$ for all $j$. Also, Algorithm 1 defines

$$\mathcal{A}_h(\boldsymbol{G}^{(t)}) = \left(\boldsymbol{1}_m^{\top} \boldsymbol{G}^{(t)\top} \hat{\boldsymbol{d}}^{(t)}\right) \hat{\boldsymbol{d}}^{(t)} = \left(\sum_{j=1}^{m} \boldsymbol{g}_j^{(t)\top} \hat{\boldsymbol{d}}^{(t)}\right) \hat{\boldsymbol{d}}^{(t)}. \tag{21}$$

Therefore

$$
\begin{aligned}
\|\mathcal{A}_h(\boldsymbol{G}^{(t)})\| &= \left\|\left(\sum_{j=1}^{m} \boldsymbol{g}_j^{(t)\top} \hat{\boldsymbol{d}}^{(t)}\right) \hat{\boldsymbol{d}}^{(t)}\right\| \\
&= \sqrt{\left(\sum_{j=1}^{m} \boldsymbol{g}_j^{(t)\top} \hat{\boldsymbol{d}}^{(t)}\right) \hat{\boldsymbol{d}}^{(t)} \hat{\boldsymbol{d}}^{(t)\top} \left(\sum_{j=1}^{m} \boldsymbol{g}_j^{(t)\top} \hat{\boldsymbol{d}}^{(t)}\right)} \\
&= \left|\sum_{j=1}^{m} \boldsymbol{g}_j^{(t)\top} \hat{\boldsymbol{d}}^{(t)}\right| \cdot \|\hat{\boldsymbol{d}}^{(t)}\| \\
&= \sum_{j=1}^{m} \boldsymbol{g}_j^{(t)\top} \hat{\boldsymbol{d}}^{(t)} \qquad (\forall_j \ \boldsymbol{g}_j^{(t)\top} \hat{\boldsymbol{d}}^{(t)} > 0) \\
&= \sum_{j=1}^{m} \boldsymbol{g}_j^{(t)\top} \frac{\mathcal{A}_h(\boldsymbol{G}^{(t)})}{\|\mathcal{A}_h(\boldsymbol{G}^{(t)})\|} \\
&= \sum_{j=1}^{m} \|\boldsymbol{g}_j^{(t)}\| \frac{\boldsymbol{g}_j^{(t)\top} \mathcal{A}_h(\boldsymbol{G}^{(t)})}{\|\boldsymbol{g}_j^{(t)}\| \|\mathcal{A}_h(\boldsymbol{G}^{(t)})\|}.
\end{aligned}
\tag{22}
$$

And the triangle inequality where $\sum_{j=1}^{m}\|\boldsymbol{g}_j\| \geq \|\sum_{j=1}^{m}\boldsymbol{g}_j\|$, we have

$$
\begin{aligned}
\sum_{j=1}^{m} L_j^{(t+1)} &\leq \sum_{j=1}^{m} L_j^{(t)} - \frac{\eta}{2}\|\mathcal{A}_h(\boldsymbol{G}^{(t)})\|^2 \\
&= \sum_{j=1}^{m} L_j^{(t)} - \frac{\eta}{2}\Big(\sum_{j=1}^{m}\|\boldsymbol{g}_j^{(t)}\|\Big)^2 \cdot \Big(\frac{\boldsymbol{g}_j^{\top}\mathcal{A}_h(\boldsymbol{G}^{(t)})}{\|\boldsymbol{g}_j\|\|\mathcal{A}_h(\boldsymbol{G}^{(t)})\|}\Big)^2 \\
&\leq \sum_{j=1}^{m} L_j^{(t)} - \frac{\eta}{2}\|\sum_{j=1}^{m}\boldsymbol{g}_j^{(t)}\|^2 \cdot \Big(\frac{\boldsymbol{g}_j^{\top}\mathcal{A}_h(\boldsymbol{G}^{(t)})}{\|\boldsymbol{g}_j\|\|\mathcal{A}_h(\boldsymbol{G}^{(t)})\|}\Big)^2 \\
&= \sum_{j=1}^{m} L_j^{(t)} - \frac{\eta}{2}\|\sum_{j=1}^{m}\nabla_{\boldsymbol{\theta}} L_j^{(t)}\|^2 \cdot \Big(\frac{\boldsymbol{g}_j^{\top}\mathcal{A}_h(\boldsymbol{G}^{(t)})}{\|\boldsymbol{g}_j\|\|\mathcal{A}_h(\boldsymbol{G}^{(t)})\|}\Big)^2.
\end{aligned}
\tag{23}
$$

Summing this inequality over $t = 0, 1, \ldots, T-1$ results in

$$
\sum_{t=0}^{T-1}\|\sum_{j=1}^{m}\nabla_{\boldsymbol{\theta}} L_j^{(t)}\|^2 \leq \frac{2}{\eta}\sum_{t=0}^{T-1}\sum_{j=1}^{m}\Big\{\big(L_j^{(t)} - L_j^{(t+1)}\big)\cdot\Big(\frac{\|\boldsymbol{g}_j\|\|\mathcal{A}_h(\boldsymbol{G}^{(t)})\|}{\boldsymbol{g}_j^{\top}\mathcal{A}_h(\boldsymbol{G}^{(t)})}\Big)^2\Big\}.
\tag{24}
$$

By defining $\min_{0\leq t\leq T-1}\Big(\frac{\boldsymbol{g}_j^{\top}\mathcal{A}_h(\boldsymbol{G}^{(t)})}{\|\boldsymbol{g}_j\|\|\mathcal{A}_h(\boldsymbol{G}^{(t)})\|}\Big)^2 = M$, we have

$$
\sum_{t=0}^{T-1}\|\sum_{j=1}^{m}\nabla_{\boldsymbol{\theta}} L_j^{(t)}\|^2 \leq \frac{2}{\eta M}\sum_{j=1}^{m}(L_j^{(0)} - L_j^{(T)}).
\tag{25}
$$

Finally, averaging both sides leads to

$$
\frac{1}{T}\sum_{t=0}^{T-1}\|\sum_{j=1}^{m}\nabla_{\boldsymbol{\theta}} L_j^{(t)}\|^2 \leq \frac{2}{\eta M T}\sum_{j=1}^{m}(L_j^{(0)} - L_j^{(T)}).
\tag{26}
$$

As $T \to \infty$, either the right-hand side in the inequality goes to $0$, implying that the minimal gradient norm converges to zero, which means a stationary point, or the inequality above holds. $\qquad\square$

### B.5 DETAIL PROOF FOR THEOREM 4

**Theorem 4.** Let the convex gradient hull be defined as $\mathbb{U} = \big\{\boldsymbol{G}\boldsymbol{\lambda} \in \mathbb{R}^d \mid \boldsymbol{\lambda} \geq \boldsymbol{0}_m, \ \boldsymbol{1}_m^{\top}\boldsymbol{\lambda} = 1\big\}$, which is contained in the primal gradient cone $\mathbb{K}$. Then there always exists a vector

$$
\boldsymbol{u}^{\star} = \arg\min_{\boldsymbol{u}\in\mathbb{U}}\|\boldsymbol{u}\|
\tag{27}
$$

such that $\|\boldsymbol{u}^{\star}\| > 0$. Consequently, $\boldsymbol{u}^{\star}$ belongs to $\mathbb{H}$, and hence $\mathbb{H}$ containing $\boldsymbol{u}^{\star}$ necessarily exists.

*Proof.* For each $i$, define

$$
\boldsymbol{v}_i = \boldsymbol{g}_i - \boldsymbol{u}^{\star}.
\tag{28}
$$

Since $\mathbb{U}$ is convex, it holds that

$$
\boldsymbol{u}^{\star} + \epsilon\boldsymbol{v}_i \in \mathbb{U}, \qquad \forall \epsilon \in [0,1].
\tag{29}
$$

By optimality of $\boldsymbol{u}^{\star}$ we have

$$
\|\boldsymbol{u}^{\star} + \epsilon\boldsymbol{v}_i\|^2 \geq \|\boldsymbol{u}^{\star}\|^2.
\tag{30}
$$

Expanding the left-hand side gives

$$
2\epsilon(\boldsymbol{u}^{\star}\cdot\boldsymbol{v}_i) + \epsilon^2\|\boldsymbol{v}_i\|^2 \geq 0.
\tag{31}
$$

For sufficiently small $\epsilon > 0$, this implies

$$
\boldsymbol{u}^{\star}\cdot\boldsymbol{v}_i \geq 0.
\tag{32}
$$

Substituting $\boldsymbol{v}_i = \boldsymbol{g}_i - \boldsymbol{u}^{\star}$ yields

$$
\boldsymbol{g}_i\cdot\boldsymbol{u}^{\star} \geq \|\boldsymbol{u}^{\star}\|^2 > 0.
\tag{33}
$$

Because $\boldsymbol{G}$ is full rank, $\boldsymbol{u}^{\star}$ cannot be a trivial solution. Moreover, since $\boldsymbol{u}^{\star}$ is nontrivial, the set $\mathbb{H}$ containing $\boldsymbol{u}^{\star}$ cannot be empty. Equivalently, the cone generated by $\boldsymbol{G}$ and its dual necessarily intersect. $\qquad\square$

## C DETAILED DISCUSSION FOR SECTION 1

In this section, we provide a detailed account of the bi-objective toy example introduced in Fig. 1. We consider $\boldsymbol{\theta} = (\theta_1, \theta_2) \in \mathbb{R}^2$ and define two objectives:

$$L_1(\boldsymbol{\theta}) = \big(c_1(\theta_2)\, f_{1a}(\boldsymbol{\theta}) + c_2(\theta_2)\, f_{1b}(\boldsymbol{\theta})\big) \cdot s, \tag{34}$$

$$L_2(\boldsymbol{\theta}) = c_1(\theta_2)\, f_{2a}(\boldsymbol{\theta}) + c_2(\theta_2)\, f_{2b}(\boldsymbol{\theta}), \tag{35}$$

where

$$f_{1a}(\boldsymbol{\theta}) = \log\big(\max\big(0.5(-\theta_1 - 7) - \tanh(-\theta_2),\, 5{\times}10^{-6}\big)\big) + 6, \tag{36}$$

$$f_{2a}(\boldsymbol{\theta}) = \log\big(\max\big(0.5(-\theta_1 + 3) + \tanh(-\theta_2) + 2,\, 5{\times}10^{-6}\big)\big) + 6, \tag{37}$$

$$f_{1b}(\boldsymbol{\theta}) = \frac{(-\theta_1 + 7)^2 + 0.1(-\theta_2 - 8)^2}{10} - 20, \tag{38}$$

$$f_{2b}(\boldsymbol{\theta}) = \frac{(-\theta_1 - 7)^2 + 0.1(-\theta_2 - 8)^2}{10} - 20, \tag{39}$$

and the switching coefficients

$$c_1(\theta_2) = \max\big(\tanh(0.5\,\theta_2),\, 0\big), \qquad c_2(\theta_2) = \max\big(\tanh(-0.5\,\theta_2),\, 0\big). \tag{40}$$

The composite loss is then

$$L_0(\boldsymbol{\theta}) = L_1(\boldsymbol{\theta}) + L_2(\boldsymbol{\theta}). \tag{41}$$

Optimization is carried out using Adam with learning rate $0.001$ for $80{,}000$ epochs. We initialize at $(-8.5, 7.5)$ and $(9.0, 9.0)$, and trajectories are plotted every $1{,}000$ epochs. The hyperparameter $s$ is varied to investigate how rescaling $L_1$ affects convergence behavior.

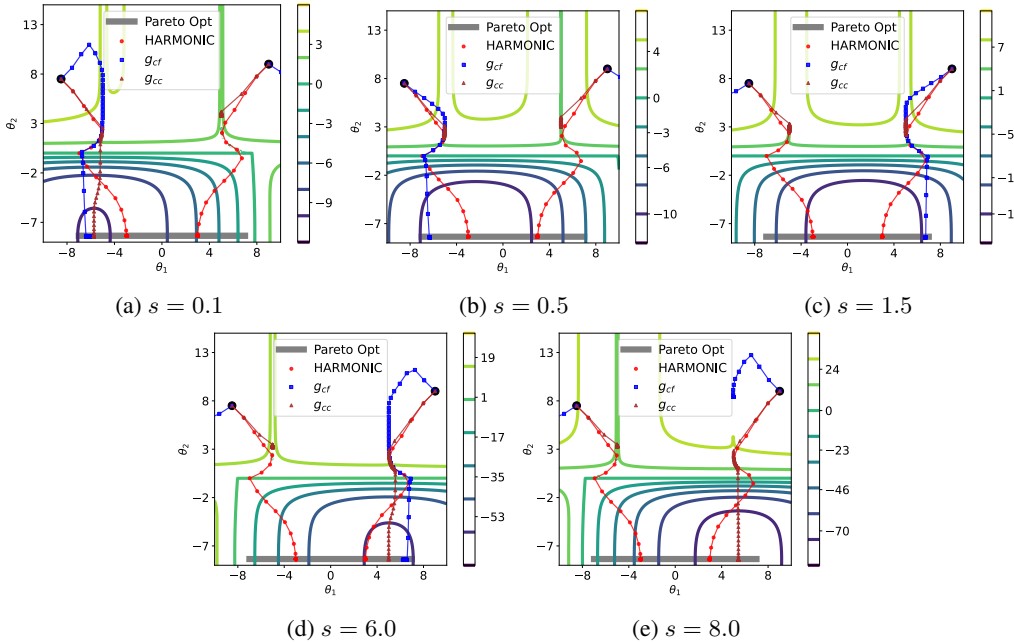

Figure 8: Optimization trajectories for the toy example under different values of $s$. Initialization points are indicated by ●, and the Pareto front is shown in ▬. ▬ indicates $\boldsymbol{g}_{cc}$, ▬ denotes $\boldsymbol{g}_{cf}$, and ▬ represents **HARMONIC**.

The observations across different settings are summarized as follows:

- $s = 0.1$: From the initial point $(-8.5, 7.5)$, all methods converged to the Pareto set. From $(9.0, 9.0)$, only HARMONIC succeeded in reaching the Pareto set.
- $s = 0.5$: From $(-8.5, 7.5)$, both HARMONIC and the non-conflict update $\boldsymbol{g}_{cf}$ converged to the Pareto set. From $(9.0, 9.0)$, only HARMONIC reached the Pareto set.

- $s = 1.5$: From $(-8.5, 7.5)$, HARMONIC alone converged to the Pareto set. From $(9.0, 9.0)$, both HARMONIC and $\boldsymbol{g}_{cf}$ converged.

- $s = 6.0$: From $(-8.5, 7.5)$, only HARMONIC converged. From $(9.0, 9.0)$, all methods successfully converged to the Pareto set.

- $s = 8.0$: From $(-8.5, 7.5)$, only HARMONIC converged. From $(9.0, 9.0)$, both HARMONIC and the conic-combination update $\boldsymbol{g}_{cc}$ converged.

Overall, in all tested cases HARMONIC consistently converged to the Pareto set, whereas alternative update rules exhibited sensitivity to both the scaling factor and the initialization point. This demonstrates the necessity of jointly enforcing *feasible* and *non-conflict*, as argued in Section 1.

## D    EXPERIMENTAL SETUP FOR FIG. 2, FIG. 3 AND FIG. 6

To produce the training curves and cosine-value histograms in Fig. 2, Fig. 3 and Fig. 6, we trained a physics-informed neural network (PINN) on the one-dimensional Burgers equation under the following controlled setting. The network architecture consisted of four fully connected layers with 32 neurons per layer using hyperbolic tangent activation function, and weights were initialized by Xavier normal initialization with zero bias. Training was performed for a total of 2,000 epochs using stochastic gradient descent (SGD) with learning rate 0.001. We sampled 10,000 points each for the PDE residual, initial condition, and boundary condition terms.

# E  DETAILED DISCUSSION FOR SECTION 4

## E.1  UNSATISFIED CASES OF THE EXISTING METHODS (SECTION 4.3)

In this section, we provide detailed numerical evidence complementing the visualizations. For each method, we present the gradient matrix $G^\top = [g_1, g_2, g_3]^\top$. We then report the update direction of HARMONIC $\mathcal{A}_h(G)$ and that of the comparison method $\mathcal{A}_c(G)$. Next, we compute inner products with two types of test vectors:

- $\mathbb{K}^*$-**test:** the inner products $\langle g_j, \mathcal{A}(G) \rangle$ between each gradient $g_j$ and $\mathcal{A}(G)$. If any of them is negative, the update is *conflicting*.

- $\mathbb{K}$-**test:** the inner products $\langle g_{\neg j}^-, \mathcal{A}(G) \rangle$, where $g_{\neg j}^-$ denotes the defining rays of the dual gradient cone $\mathbb{K}^*$. If any of them is negative, the update lies in an *infeasible* region. A degenerate case where only one inner product equals 1 while the others are 0 indicates that the update ignores all but one gradient.

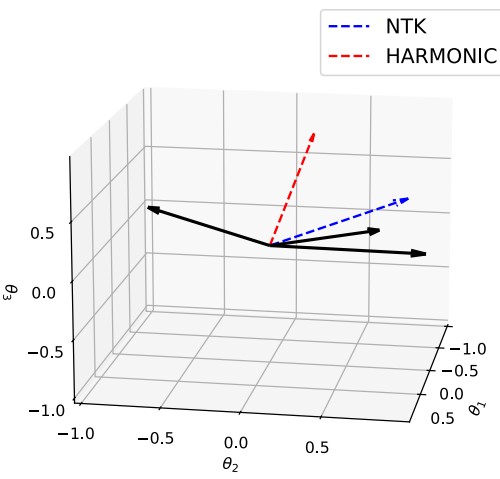

Figure 9: Unsatisfied cases of NTK. $\longrightarrow$ denotes the gradient of each loss. $--\rightarrow$ represents $\mathcal{A}_c(G)$ of the existing methods. $--\rightarrow$ represents $\mathcal{A}_h(G)$ of HARMONIC.

**NTK (Fig. 9).**

$$G^\top = \begin{bmatrix} -0.4278 & -0.8925 & 0.1431 \\ 0.5179 & 0.7754 & 0.3613 \\ -0.0979 & 0.9941 & -0.0472 \end{bmatrix},$$

$$\mathcal{A}_h(G) = [-0.4067, 0.1670, 0.6891]^\top, \quad \mathcal{A}_c(G) = [-0.0026, 0.2923, 0.1524]^\top.$$

$\mathbb{K}^*$-test:

$$\langle g_j, \mathcal{A}_h(G) \rangle = (0.1236, 0.1679, 0.1733) > 0, \quad \langle g_j, \mathcal{A}_c(G) \rangle = (-0.2380, 0.2804, 0.2837).$$

NTK induces conflict with $g_1$.

**MGDA (Fig. 10).**

$$G^\top = \begin{bmatrix} 0.0137 & 0.1432 & 0.3131 \\ 0.3571 & 0.5491 & 0.1414 \\ 0.9823 & 0.9361 & 0.0552 \end{bmatrix},$$

$$\mathcal{A}_h(G) = [0.4320, 0.6414, 0.3868]^\top, \quad \mathcal{A}_c(G) = [0.0137, 0.1432, 0.3131]^\top.$$

$\mathbb{K}$-test:

$$\langle g_{\neg j}^-, \mathcal{A}_h(G) \rangle = (0.9674, 0.4974, 0.2455) > 0, \quad \langle g_{\neg j}^-, \mathcal{A}_c(G) \rangle = (1.0000, 0.0000, 0.0000) \geq 0.$$

MGDA produces a degenerate update, aligning only with $g_1$.

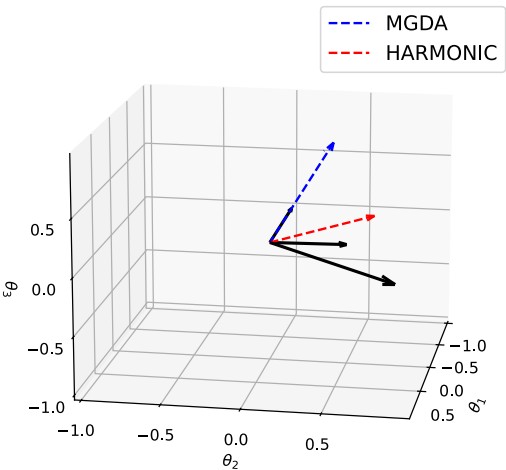

Figure 10: Unsatisfied cases of MGDA. $\longrightarrow$ denotes the gradient of each loss. $\dashrightarrow$ represents $\mathcal{A}_c(\boldsymbol{G})$ of the existing methods. $\dashrightarrow$ represents $\mathcal{A}_h(\boldsymbol{G})$ of HARMONIC.

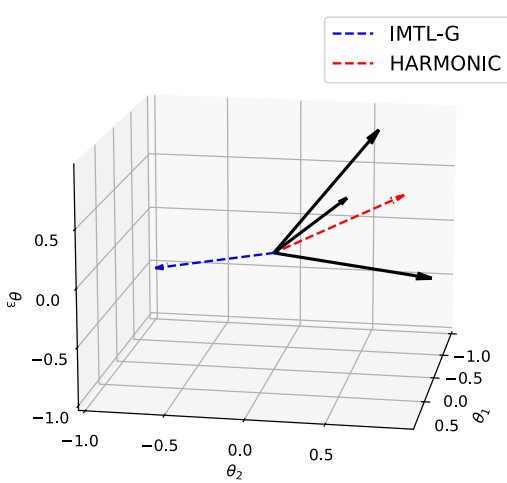

Figure 11: Unsatisfied cases of IMTL-G. $\longrightarrow$ denotes the gradient of each loss. $\dashrightarrow$ represents $\mathcal{A}_c(\boldsymbol{G})$ of the existing methods. $\dashrightarrow$ represents $\mathcal{A}_h(\boldsymbol{G})$ of HARMONIC.

**IMTL-G (Fig. 11).**

$$\boldsymbol{G}^\top = \begin{bmatrix} -0.1323 & 1.0037 & -0.2090 \\ -0.3367 & 0.4206 & 0.3984 \\ -0.9204 & 0.5455 & 0.8364 \end{bmatrix},$$

$$\mathcal{A}_h(\boldsymbol{G}) = [-0.4360, 0.6668, 0.3361]^\top, \quad \mathcal{A}_c(\boldsymbol{G}) = [0.8040, -0.6127, 0.1140]^\top.$$

$\mathbb{K}$-test:

$$\langle \boldsymbol{g}_{\neg j}^-, \mathcal{A}_h(\boldsymbol{G}) \rangle = (0.3225, 0.4974, 0.2455) > 0, \quad \langle \boldsymbol{g}_{\neg j}^-, \mathcal{A}_c(\boldsymbol{G}) \rangle = (-1.2588, 4.6542, -2.3954).$$

$\mathbb{K}^*$-test:

$$\langle \boldsymbol{g}_j, \mathcal{A}_h(\boldsymbol{G}) \rangle = (0.6566, 0.5612, 1.0462) > 0, \quad \langle \boldsymbol{g}_j, \mathcal{A}_c(\boldsymbol{G}) \rangle = (-0.7451, -0.4830, -0.9788) < 0.$$

IMTL-G yields infeasible and conflicting updates.

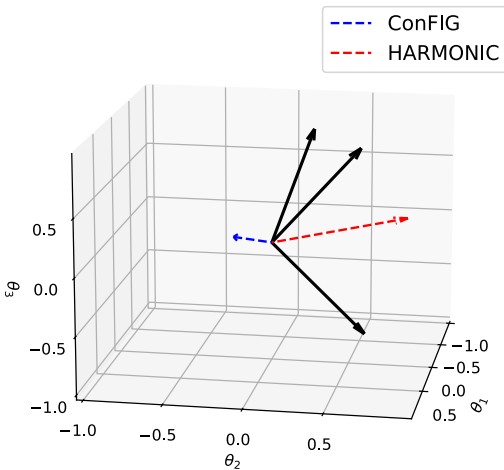

Figure 12: Unsatisfied cases of ConFIG. $\longrightarrow$ denotes the gradient of each loss. $\dashrightarrow$ represents $\mathcal{A}_c(\boldsymbol{G})$ of the existing methods. $\dashrightarrow$ represents $\mathcal{A}_h(\boldsymbol{G})$ of HARMONIC.

**ConFIG (Fig. 12).**

$$\boldsymbol{G}^\top = \begin{bmatrix} -0.1807 & 0.2457 & 0.9523 \\ 0.4623 & 0.6682 & -0.5829 \\ -0.1050 & 0.5583 & 0.8230 \end{bmatrix},$$

$$\mathcal{A}_h(\boldsymbol{G}) = [0.2071, 0.7769, 0.3001]^\top, \quad \mathcal{A}_c(\boldsymbol{G}) = [5.5958, -0.4348, 2.2241]^\top.$$

$\mathbb{K}^*$-test:

$$\langle \boldsymbol{g}_{\neg j}^-, \mathcal{A}_h(\boldsymbol{G}) \rangle = (0.4187, 0.6966, 0.3735) > 0, \quad \langle \boldsymbol{g}_{\neg j}^-, \mathcal{A}_c(\boldsymbol{G}) \rangle = (75.9019, 26.7388, -66.1925).$$

ConFIG violates feasibility.

Overall, these numerical checks confirm the unsatisfied cases of existing approaches, and demonstrate that HARMONIC uniquely guarantees both *feasible* and *non-conflict* simultaneously.

### E.2 TOY EXAMPLE FOR COMPARING MULTI-GRADIENT METHODS (SECTION 4.3)

To provide a controlled environment for analyzing gradient interactions among objectives, we construct a three-objective optimization problem

$$f : \mathbb{R}^4 \to \mathbb{R}^3, \qquad \boldsymbol{x} = (x_1, x_2, x_3, x_4)^\top \mapsto f(\boldsymbol{x}) = \big(f_1(\boldsymbol{x}), f_2(\boldsymbol{x}), f_3(\boldsymbol{x})\big),$$

with the search domain restricted to $\boldsymbol{x} \in [0, 1]^4$.

**Radial terms in** $(x_3, x_4)$    We begin by defining three quadratic distances

$$g(\boldsymbol{x}) = (x_3 - 0.5)^2 + (x_4 - 0.5)^2, \tag{42}$$

$$g_1(\boldsymbol{x}) = (x_3 - 0.45)^2 + (x_4 - 0.55)^2, \tag{43}$$

$$g_2(\boldsymbol{x}) = (x_3 - 0.55)^2 + (x_4 - 0.45)^2. \tag{44}$$

The terms $g_1$ and $g_2$ create two nearby basins of attraction that determine which objective, $f_1$ or $f_2$, dominates the local behavior.

**Switching mechanism in** $(x_1, x_2)$    We introduce a smooth logarithmic transform

$$\phi(a) = \log\big(\max(|a|, \varepsilon)\big) + 6, \qquad \varepsilon = 6 \times 10^{-6}, \tag{45}$$

and two gating functions

$$c_1(\boldsymbol{x}) = \max\big(\tanh(20(x_2 - 0.5)), 0\big), \qquad c_2(\boldsymbol{x}) = \max\big(\tanh(-20(x_2 - 0.5)), 0\big). \tag{46}$$

These gates satisfy $c_1 \approx 1$ for $x_2 > 0.5$ and $c_2 \approx 1$ for $x_2 < 0.5$, switching the curvature of the landscape across the line $x_2 = 0.5$.

Using these components, we define four shape functions:

$$z_1^{\log}(\boldsymbol{x}) = \phi(0.5(-20(x_1 - 0.5) - 7) - \tanh(-20(x_2 - 0.5))), \tag{47}$$

$$z_2^{\log}(\boldsymbol{x}) = \phi(0.5(-20(x_1 - 0.5) + 3) + \tanh(-20(x_2 - 0.5)) + 2), \tag{48}$$

$$z_1^{\mathrm{sq}}(\boldsymbol{x}) = \frac{(-20(x_1 - 0.5) + 7)^2 + 0.1(-20(x_2 - 0.5) - 20)^2}{10} - 20, \tag{49}$$

$$z_2^{\mathrm{sq}}(\boldsymbol{x}) = \frac{(-20(x_1 - 0.5) - 7)^2 + 0.1(-20(x_2 - 0.5) - 20)^2}{10} - 20. \tag{50}$$

The two composite shape functions are

$$z_1 = z_1^{\log} c_1 + z_1^{\mathrm{sq}} c_2, \qquad z_2 = z_2^{\log} c_1 + z_2^{\mathrm{sq}} c_2. \tag{51}$$

**Definition of the objectives**  With a global scaling parameter $s > 0$ (we use $s = 6$ in our experiments), we define

$$f_1(\boldsymbol{x}) = \frac{z_1(\boldsymbol{x})}{10} \, s \, g_1(\boldsymbol{x}), \tag{52}$$

$$f_2(\boldsymbol{x}) = \frac{z_2(\boldsymbol{x})}{10} \, g_2(\boldsymbol{x}), \tag{53}$$

$$f_3(\boldsymbol{x}) = (1 + g(\boldsymbol{x})) \sin\left(\frac{\pi}{2} x_1\right) \sin\left(\frac{\pi}{2} x_2\right). \tag{54}$$

The optimization problem is therefore

$$\min_{\boldsymbol{x} \in [0,1]^4} ( f_1(\boldsymbol{x}),\ f_2(\boldsymbol{x}),\ f_3(\boldsymbol{x}) ). \tag{55}$$

**Sign structure and role reversal of $g_1$ and $g_2$**  For typical initial points used in our experiments, the values of $f_1$ and $f_2$ are *positive*, so reducing $g_1$ and $g_2$ decreases both objectives. Along the Pareto front, however, $f_1$ and $f_2$ become *negative*, which reverses the effect of the multiplicative factors: $g_1$ and $g_2$ then act as *maximization* terms. This sign change results in a characteristic "role switch" of the gradients in $(x_3, x_4)$, creating sharp transitions in the interaction between the objectives.

**Visualization**  To illustrate the structural properties of the toy problem, we additionally provide contour plots of the individual objectives under three representative slices of the four-dimensional domain. Each slice fixes two coordinates of $\boldsymbol{x}$ and visualizes $f_1$, $f_2$, and $f_3$ as functions of the remaining two coordinates:

(1) **Slice in $(x_1, x_2)$ (Fig. 13a):** Fix $(x_3, x_4) = (0.5, 0.5)$, and visualize $f_1$, $f_2$, and $f_3$ over $(x_1, x_2) \in [0, 1]^2$. This slice highlights how the log-to-quadratic transition across $x_2 = 0.5$ shapes the curvature of the first two objectives.

(2) **Initial-condition slice in $(x_3, x_4)$ (Fig. 13b):** Fix $(x_1, x_2) = (0.3, 0.8)$ and visualize $f_1$, $f_2$, and $f_3$ over $(x_3, x_4) \in [0, 1]^2$. For this configuration, $f_1$ and $f_2$ are positive, so the factors $g_1$ and $g_2$ act as *minimization* terms, creating distinct attraction regions centered near $(0.45, 0.55)$ and $(0.55, 0.45)$.

(3) **Pareto-condition slice in $(x_3, x_4)$ (Fig. 13c):** Fix $(x_1, x_2) = (0.3, 0.1)$ and visualize $f_1$, $f_2$, and $f_3$ over $(x_3, x_4) \in [0, 1]^2$. In this region, $f_1$ and $f_2$ become negative, reversing the effect of $g_1$ and $g_2$ so that they now behave as *maximization* terms. This results in a qualitatively different landscape and induces sharp gradient-direction changes relevant to multi-gradient optimization.

These three complementary slices allow us to clearly visualize (i) the curvature switching in $(x_1, x_2)$, (ii) the initial positive-objective regime in $(x_3, x_4)$, and (iii) the sign-flipped regime along the Pareto front.

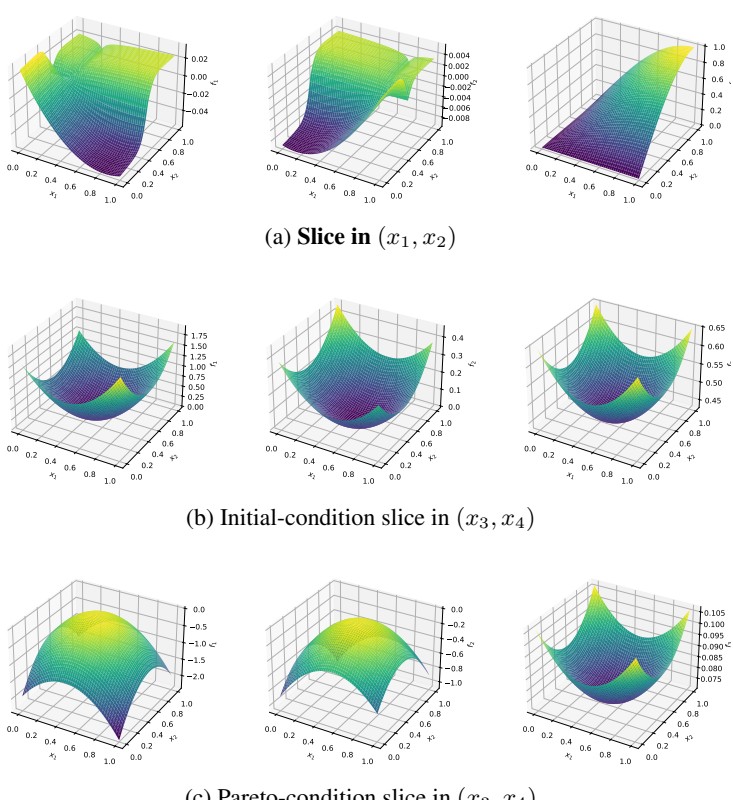

(a) **Slice in** $(x_1, x_2)$

(b) Initial-condition slice in $(x_3, x_4)$

(c) Pareto-condition slice in $(x_3, x_4)$

Figure 13: Three-dimensional surface plots of the objective functions under different slices of the decision space. (a) Slice with respect to $(x_1, x_2)$ while fixing $x_3 = x_4 = 0.5$. (b) Slice with respect to $(x_3, x_4)$ while fixing $x_1 = 0.3, x_2 = 0.8$. (c) Slice with respect to $(x_3, x_4)$ while fixing $x_1 = 0.3, x_2 = 0.1$.

# F  DETAILS OF PDES

## F.1  BURGERS' EQUATIONS

**1D Burgers Equation (Burgers1d)**

The PDE is given by

$$u_t + u\,u_x = \nu\,u_{xx}. \tag{56}$$

The domain is defined as

$$(x,t) \in \Omega \times T = [-1,1] \times [0,1].$$

The initial and boundary conditions are

$$u(x,0) = -\sin(\pi x), \tag{57}$$
$$u(-1,t) = u(1,t) = 0. \tag{58}$$

The parameter is

$$\nu = \frac{0.01}{\pi}. \tag{59}$$

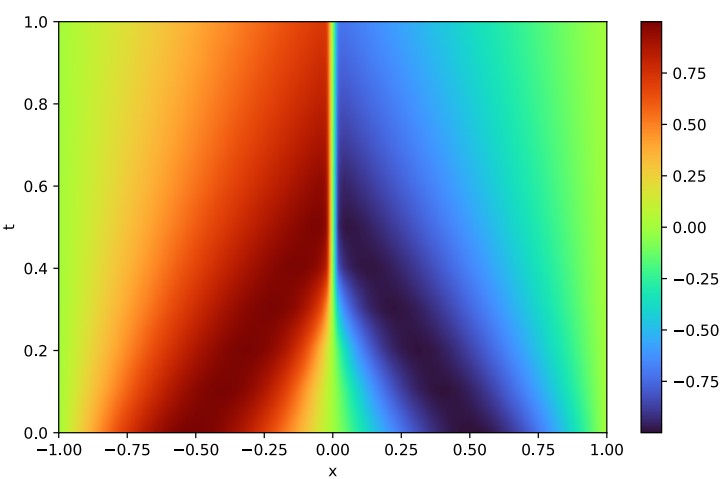

Figure 14: Reference solution of Burgers1d using PINNacle reference data.

**2D Coupled Burgers Equation (Burgers2d)**

The governing equations are given by

$$u_t + uu_x + vu_y = \nu(u_{xx} + u_{yy}), \tag{60}$$
$$v_t + uv_x + vv_y = \nu(v_{xx} + v_{yy}). \tag{61}$$

The domain is defined as

$$(x,y,t) \in \Omega \times T = [0,L]^2 \times [0,1]. \tag{62}$$

The initial conditions are given by

$$\boldsymbol{w}(x,y) = \sum_{i=-L}^{L} \sum_{j=-L}^{L} \Big[ \boldsymbol{a}_{ij} \sin\big(2\pi(ix+jy)\big) + \boldsymbol{b}_{ij} \cos\big(2\pi(ix+jy)\big) \Big], \tag{63}$$

$$[u(x,y,0),\ v(x,y,0)] = 2\,\boldsymbol{w}(x,y) + \boldsymbol{c}. \tag{64}$$

where $\boldsymbol{a}_{ij}, \boldsymbol{b}_{ij}, \boldsymbol{c} \sim \mathcal{N}(0,1) \in R^2$. The boundary conditions are given by

$$u(0,y,t) = u(L,y,t), \tag{65}$$
$$u(x,0,t) = u(x,L,t), \tag{66}$$

$$v(0, y, t) = v(L, y, t), \tag{67}$$
$$v(x, 0, t) = v(x, L, t). \tag{68}$$

The parameters are

$$L = 4, \quad \nu = 0.001. \tag{69}$$

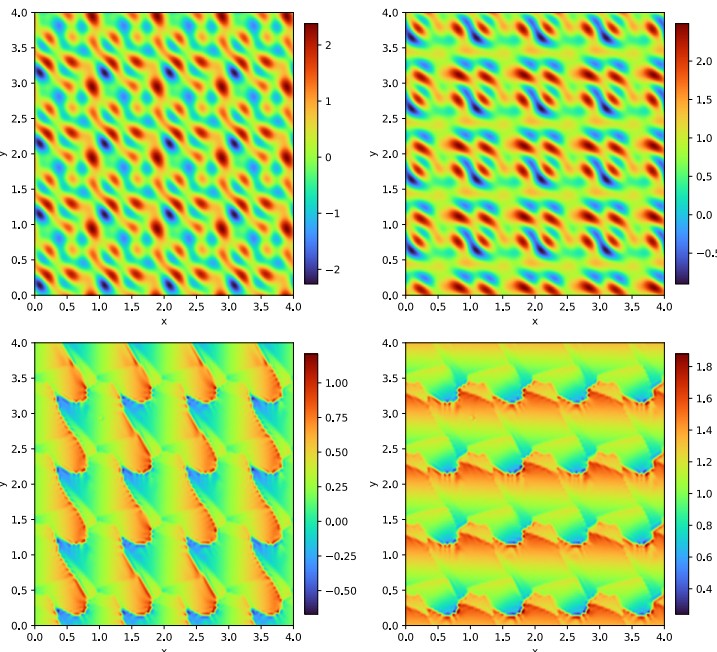

Figure 15: Reference solution of Burgers2d at timesteps $t = 0, 1$, where the left panels show $u$ and the right panels show $v$ from the PINNacle reference data.

**Experiment details**   The model architecture consisted of 3 layers with 50 neurons, and training was performed for 50,000 epochs over 5 independent runs with a learning rate of $10^{-3}$. We employed the `tanh` activation function, Glorot normal initialization (Glorot & Bengio, 2010), and the Adam optimizer.

Table 5: Sampling sizes used in our experiments, following the default configuration of PINNacle.

|                    | Burgers1d | Burgers2d |
| ------------------ | --------- | --------- |
| Collocation points | 8192      | 32768     |
| Initial points     | 2048      | 8192      |
| Boundary points    | 2048      | 8192      |
| Test points        | 8192      | 32768     |

With this experimental setting, the results are presented as follows.

### F.2   POISSON EQUATIONS

**Poisson 2D Classic (Poisson2d-C)**

The PDE is given by

$$u_{xx} + u_{yy} = 0. \tag{70}$$

Table 6: Average of relative $L^2$ errors (with standard deviation) estimated across 5 random seeds for each algorithm. We highlight **the best** and the second methods for Burgers' Equation.

| Method | Burgers1d | Burgers2d |
|---|---|---|
| MultiAdam | 0.8526 (0.1900) | 0.5491 (0.0000) |
| LRA | 0.0427 (0.0267) | **0.5170** (**0.0031**) |
| ReLoBRaLo | **0.0159** (**0.0017**) | 0.5282 (0.0045) |
| MGDA | 0.0505 (0.0737) | 0.5231 (0.0054) |
| PCGrad | 0.0218 (0.0101) | 0.5228 (0.0009) |
| CAGrad | 0.0280 (0.0097) | 0.5244 (0.0017) |
| IMTL-G | 0.3221 (0.2031) | 1.2998 (1.4134) |
| Aligned-MTL | 0.0280 (0.0097) | 0.5248 (0.0018) |
| ConFIG | 0.0264 (0.0108) | 0.5245 (0.0174) |
| HARMONIC | 0.0164 (0.0023) | 0.5439 (0.0126) |

The domain is a rectangle minus four circles $(x, y) \in \Omega = \Omega_{\text{rec}} \setminus R_i$, where $\Omega_{\text{rec}} = [-0.5, 0.5]^2$ is the rectangle and $R_i$ denotes four circle areas:

$$R_1 = \{(x, y) : (x - 0.3)^2 + (y - 0.3)^2 \leq 0.1^2\}, \tag{71}$$

$$R_2 = \{(x, y) : (x + 0.3)^2 + (y - 0.3)^2 \leq 0.1^2\}, \tag{72}$$

$$R_3 = \{(x, y) : (x - 0.3)^2 + (y + 0.3)^2 \leq 0.1^2\}, \tag{73}$$

$$R_4 = \{(x, y) : (x + 0.3)^2 + (y + 0.3)^2 \leq 0.1^2\}. \tag{74}$$

The boundary condition is

$$u(x, y) = 0, (x, y) \in \partial R_i, \tag{75}$$

$$u(x, y) = 1, (x, y) \in \partial \Omega_{\text{rec}}. \tag{76}$$

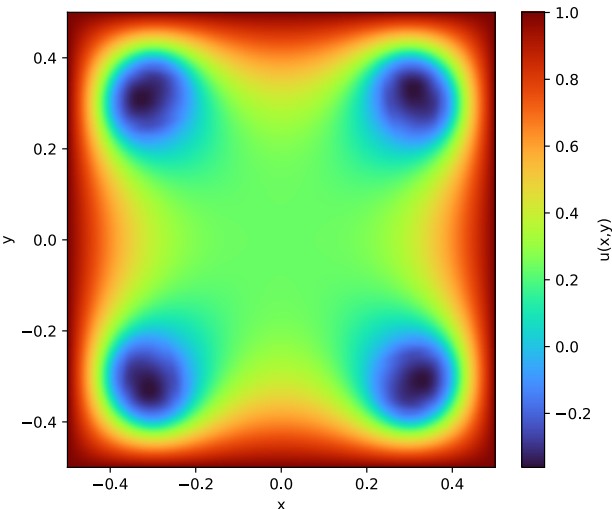

Figure 16: Reference solution of Poisson2d-C using PINNacle reference data.

**Poisson-Boltzmann (Helmholtz) 2D Irregular Geometry (Poisson2d-CG)**

The PDE is

$$-\nabla u + k^2 u = f(x, y), \tag{77}$$

where $f(x)$ is defined as

$$f(x) = A \cdot (\Sigma_i \mu_i^2 + x_i^2) \cdot \sin(\mu_1 \pi x_1) \sin(\mu_2 \pi x_2). \tag{78}$$

The domain is $(x, y) \in \Omega = [-1, 1]^2$ with several circles removed. The circles $\Omega_{\mathrm{rec}} = \cup_{i=1}^4 R_i$ are

$$R_1 = \{(x, y) : (x - 0.5)^2 + (y - 0.5)^2 \leq 0.2^2\}, \tag{79}$$
$$R_2 = \{(x, y) : (x - 0.4)^2 + (y + 0.4)^2 \leq 0.4^2\}, \tag{80}$$
$$R_3 = \{(x, y) : (x + 0.2)^2 + (y + 0.7)^2 \leq 0.1^2\}, \tag{81}$$
$$R_4 = \{(x, y) : (x + 0.6)^2 + (y - 0.5)^2 \leq 0.3^2\}. \tag{82}$$

The boundary conditions are

$$u(x, y) = 0.2, \quad (x, y) \in \partial\Omega_{\mathrm{rec}}, \tag{83}$$
$$u(x, y) = 1, \quad (x, y) \in \partial\Omega_{\mathrm{circle}}. \tag{84}$$

The parameters are

$$\mu_1 = 1, \ \mu_2 = 4, \ k = 8, \ A = 10. \tag{85}$$

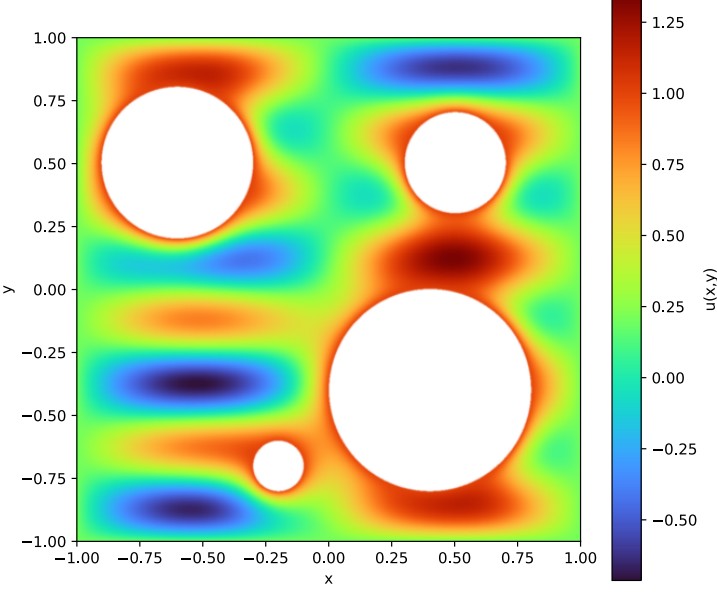

Figure 17: Reference solution of Poisson2d-CG using PINNacle reference data.

**Experiment details** The model architecture consisted of 3 layers with 50 neurons, and training was performed for 50,000 epochs over 5 independent runs with a learning rate of $10^{-3}$. We employed the tanh activation function, Glorot normal initialization, and the Adam optimizer.

Table 7: Sampling sizes used in our experiments, following the default configuration of PINNacle.

|  | **Poisson2d-C** | **Poisson2d-CG** |
|---|---|---|
| Collocation points | 8192 | 8192 |
| Initial points | 2048 | 2048 |
| Boundary points | 2048 | 2048 |
| Test points | 8192 | 8192 |

With this experimental setting, the results are presented as follows.

### F.3 HEAT EQUATIONS

**2D Heat with Varying Coefficients (Heat2d-VC)**

The governing equation is

$$u_t - \nabla \cdot (a(x, y)\nabla u) = f(x, y, t). \tag{86}$$

Table 8: Average of relative $L^2$ errors (with standard deviation) estimated across 5 random seeds for each algorithm. We highlight **the best** and the second methods for Poisson equations.

| Method | Poisson2d-C | Poisson2d-CG |
|---|---|---|
| MultiAdam | 0.6733 (0.0820) | 0.8104 (0.0120) |
| LRA | 0.5712 (0.1595) | 0.3154 (0.0448) |
| ReLoBRaLo | 0.6602 (0.0221) | 0.6338 (0.0259) |
| MGDA | 0.6777 (0.0239) | 0.7486 (0.0281) |
| PCGrad | 0.6918 (0.0190) | 0.1914 (0.0450) |
| CAGrad | 0.7806 (0.1553) | 0.3615 (0.0621) |
| IMTL-G | 0.5039 (0.0684) | 1.0175 (1.1243) |
| Aligned-MTL | 0.2847 (0.2363) | 0.7024 (0.0416) |
| ConFIG | 0.6954 (0.4296) | **0.1252 (0.1941)** |
| HARMONIC | **0.0214 (0.0179)** | 0.1460 (0.1810) |

The domain is

$$(x, y) \in \Omega \times T = [0, 1]^2 \times [0, 5]. \tag{87}$$

The function $a(x, y)$ is chosen similarly to Darcy flow but with an exponential GRF.

The function $f(x, y, t)$ is defined as

$$f(x, y, t) = A \sin(m_1 \pi x) \sin(m_2 \pi y) \sin(m_3 \pi t). \tag{88}$$

The initial and boundary conditions are

$$u(x, y, 0) = 0, \quad (x, y) \in \Omega, \tag{89}$$
$$u(x, y, t) = 0, \quad (x, y) \in \partial\Omega. \tag{90}$$

The parameters are

$$A = 200, \ m_1 = 1, \ m_2 = 5, \ m_3 = 1. \tag{91}$$

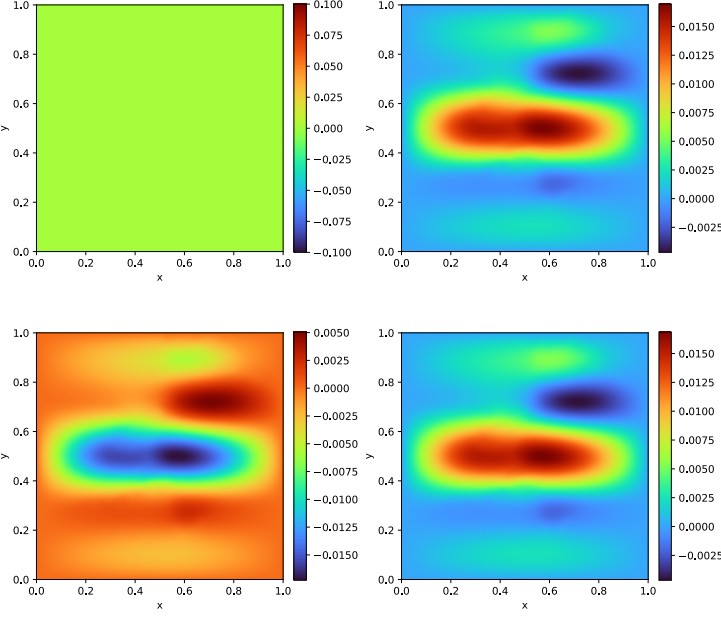

Figure 18: Reference solution of Heat2d-VC at timesteps $t = 0, 0.3, 0.7, 1$ using PINNacle reference data.

**2D Heat Multi-Scale (Heat2d-MS)**

The PDE equation is given by

$$u_t - \frac{1}{(500\pi)^2} u_{xx} - \frac{1}{\pi^2} u_{yy} = 0. \tag{92}$$

The domain is $(x, y) \in \Omega \times T = [0, 1]^2 \times [0, 5]$.

The initial and boundary conditions are

$$u(x, y, 0) = \sin(20\pi x) \sin(\pi y), \qquad (x, y) \in \Omega, \tag{93}$$
$$u(x, y, t) = 0, \qquad (x, y) \in \partial\Omega, \ t \in T. \tag{94}$$

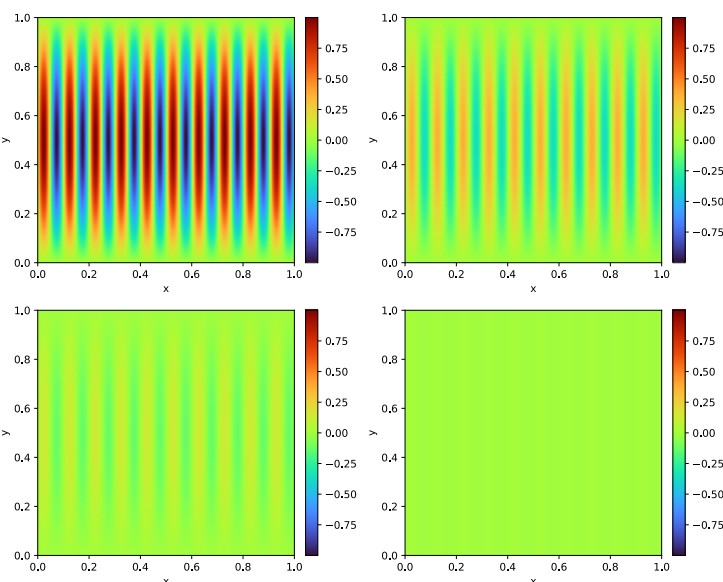

Figure 19: Reference solution of Heat2d-MS at timesteps $t = 0, 1, 2, 5$ using PINNacle reference data.

## 2D Heat Complex Geometry (Heat2d-CG)

The PDE is

$$u_t - \nabla u = 0. \tag{95}$$

The domain is defined as

$$(x, y, t) \in \Omega \times T = ([-8, 8] \times [-12, 12] \setminus \cup_i R_i) \times [0, 3]. \tag{96}$$

The initial condition is

$$u(x, y, 0) = 0. \tag{97}$$

The boundary conditions are

$$-n \cdot (-\nabla u) = 5 - u, \tag{98}$$
$$-n \cdot (-\nabla u) = 1 - u, \tag{99}$$
$$-n \cdot (-\nabla u) = 0.1 - u. \tag{100}$$

## N Dimensional Heat Equation (HNd)

The governing equation is

$$u_t = k\Delta u + f(\boldsymbol{x}, t), \tag{101}$$

where

$$f(\boldsymbol{x}, t) = -\frac{1}{d} \|\boldsymbol{x}\|_2^2 \exp\left(\tfrac{1}{2} \|\boldsymbol{x}\|_2^2 + t\right). \tag{102}$$

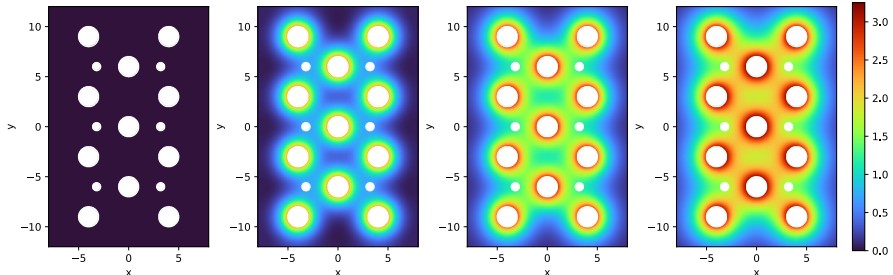

Figure 20: Reference solution of Heat2d-CG at timesteps $t = 0, 1, 2, 3$ using PINNacle reference data.

The domain is $(\boldsymbol{x}, t) \in \Omega \times T$. The geometric domain $\Omega$ is a unit sphere in $d$-dimensional space and $T = [0, 1]$. We choose $d$ as 5.

The initial and boundary conditions are

$$\boldsymbol{n} \cdot \nabla u = g(\boldsymbol{x}, t), \ (\boldsymbol{x}, t) \in \partial\Omega \times [0, 1], \tag{103}$$
$$u(\boldsymbol{x}, 0) = g(\boldsymbol{x}, 0), \ \boldsymbol{x} \in \Omega, \tag{104}$$

where

$$g(\boldsymbol{x}, t) = \exp\left(\tfrac{1}{2}\|\boldsymbol{x}\|_2^2 + t\right). \tag{105}$$

We choose dimension $d = 5$ and the parameter is

$$k = \frac{1}{d}. \tag{106}$$

**Experiment details** The model architecture consisted of 3 layers with 50 neurons for Heat2d-VC, Heat2d-CG and HNd, while Heat2d-MS was configured with 5 layers and 100 neurons. Training was performed for 50,000 epochs over 5 independent runs with a learning rate of $10^{-3}$. We employed the tanh activation function, Glorot normal initialization, and the Adam optimizer.

Table 9: Sampling sizes used in our experiments, following the default configuration of PINNacle.

|  | **Heat2d-VC** | **Heat2d-MS** | **Heat2d-CG** | **HNd** |
|---|---|---|---|---|
| Collocation points | 32768 | 32768 | 32768 | 32768 |
| Initial points | 8192 | 8192 | 8192 | 8192 |
| Boundary points | 8192 | 8192 | 8192 | 8192 |
| Test points | 32768 | 32768 | 32768 | 32768 |

With this experimental setting, the results are presented as follows.

Table 10: Average of relative $L^2$ errors (with standard deviation) estimated across 5 random seeds for each algorithm. We highlight **the best** and the second methods for Heat.

| Method | **Heat2d-VC** | **Heat2d-MS** | **Heat2d-CG** | **HNd** |
|---|---|---|---|---|
| MultiAdam | 1.1038 (0.0708) | 0.9987 (0.0021) | 0.7049 (0.0407) | 0.0322 (0.0338) |
| LRA | 0.2984 (0.0474) | **0.0278 (0.0070)** | 0.2930 (0.1242) | 0.0020 (0.0011) |
| ReLoBRaLo | 1.8305 (0.1208) | 0.0355 (0.0212) | 0.1057 (0.0034) | 0.0034 (0.0067) |
| MGDA | 1.2157 (0.1851) | 0.0824 (0.1416) | 0.1205 (0.0063) | 0.0048 (0.0088) |
| PCGrad | 1.1198 (0.1314) | 0.0324 (0.0161) | 0.0991 (0.0063) | 0.0028 (0.0019) |
| CAGrad | 0.3355 (0.0437) | 0.0442 (0.0176) | 0.1173 (0.0013) | **0.0005 (0.0001)** |
| IMTL-G | 1.2430 (0.4677) | 0.9933 (0.0027) | 0.0989 (0.0037) | 0.0006 (0.0002) |
| Aligned-MTL | 0.9080 (0.6703) | 0.9849 (0.4311) | 0.1189 (0.0070) | 0.0006 (0.0002) |
| ConFIG | 0.2812 (0.1329) | 0.8121 (0.4211) | 0.1195 (0.0107) | 0.0006 (0.0001) |
| Harmonic | **0.1927 (0.0071)** | 0.0317 (0.0331) | **0.0918 (0.0062)** | **0.0005 (0.0000)** |

### F.4 WAVE EQUATION

**1D Wave Equation (Wave1d-C)**

The governing equation is

$$u_{tt} - 4\,u_{xx} = 0. \tag{107}$$

The domain is $(x,t) \in \Omega \times T = [0,1] \times [0,1]$.
The initial condition is

$$u(x,0) = \sin(\pi x) + \frac{1}{2}\sin(4\pi x), \tag{108}$$

$$u_t(x,0) = 0. \tag{109}$$

The boundary conditions are

$$u(0,t) = u(1,t) = 0. \tag{110}$$

The analytical solution of this problem is

$$u(x,t) = \sin(\pi x)\cos(2\pi t) + \frac{1}{2}\sin(4\pi x)\cos(8\pi t). \tag{111}$$

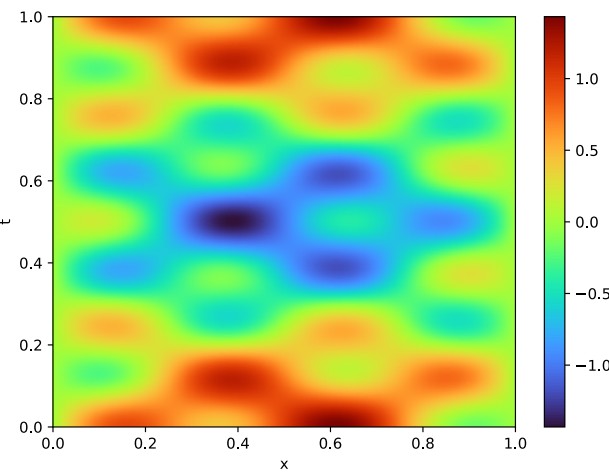

Figure 21: Reference solution of Wave1d-C using PINNacle reference data.

**Experiment details**   The model architecture consisted of 3 layers with 50 neurons. Training was performed for 50,000 epochs over 5 independent runs with a learning rate of $10^{-3}$. We employed the tanh activation function, Glorot normal initialization, and the Adam optimizer.

Table 11: Sampling sizes used in our experiments, following the default configuration of PINNacle.

|  | **Wave1d-C** |
| --- | --- |
| Collocation points | 8192 |
| Initial points | 2048 |
| Boundary points | 2048 |
| Test points | 8192 |

With this experimental setting, the results are presented as follows.

### F.5 NAVIER-STOKES EQUATIONS

**2D NS lid-driven flow (NS2d-C)**

Table 12: Average of relative $L^2$ errors (with standard deviation) estimated across 5 random seeds for each algorithm. We highlight **the best** and the second methods for Wave equation.

| Method | Wave1d-C |
|---|---|
| MultiAdam | 1.1540 (0.0630) |
| LRA | 0.2608 (0.0542) |
| ReLoBRaLo | 0.4102 (0.0476) |
| MGDA | 0.4627 (0.1359) |
| PCGrad | 0.2256 (0.0701) |
| CAGrad | 0.1186 (0.0071) |
| IMTL-G | 0.7664 (0.4757) |
| Aligned-MTL | 0.3801 (0.1741) |
| ConFIG | 0.0668 (0.0279) |
| HARMONIC | **0.0655 (0.0293)** |

The PDE is given by

$$uu_x + vu_y + p_x = \frac{1}{Re}(u_{xx} + u_{yy}), \tag{112}$$

$$uv_x + vv_y + p_y = \frac{1}{Re}(v_{xx} + v_{yy}), \tag{113}$$

$$u_x + v_y = 0. \tag{114}$$

The domain is $(x, y) \in \Omega = [0, 1]^2$, the top boundary is $\Gamma_1$, the left, right and bottom boundary is $\Gamma_2$. The boundary conditions are

$$u(x, y) = 4x(1 - x), \qquad\qquad y \in \Gamma_1, \tag{115}$$
$$v(x, y) = 0, \qquad\qquad y \in \Gamma_1, \tag{116}$$
$$u(x, y) = 0, v(x, y) = 0, \qquad\qquad y \in \Gamma_2, \tag{117}$$
$$p(0, 0) = 0. \tag{118}$$

The Reynolds number $Re = 100$.

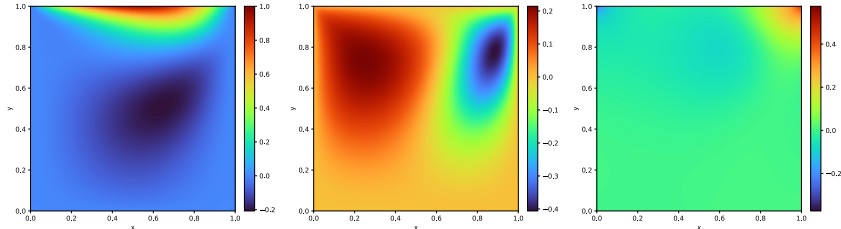

Figure 22: Reference solution of NS2d-C, where the left panel shows $u$, the middle pannel shows $v$, and the right panel shows $p$ from the PINNacle reference data.

**2D Back Step Flow (NS-CG)**

The PDE is given by

$$uu_x + vu_y + p_x = \frac{1}{Re}(u_{xx} + u_{yy}), \tag{119}$$

$$uv_x + vv_y + p_y = \frac{1}{Re}(v_{xx} + v_{yy}), \tag{120}$$

$$u_x + v_y = 0. \tag{121}$$

The domain is defined as $(x, y) \in \Omega = [0, 4] \times [0, 2] \setminus ([0, 2] \times [1, 2] \cup R_i)$ (excluding the top-left quarter). The inlet boundary is $\Gamma_{in}$ and the outlet boundary is $\Gamma_{out}$.

The boundary conditions are

$$u(0, y) = 4y(1 - y), \tag{122}$$
$$v(0, y) = 0, \tag{123}$$
$$p(4, y) = 0, \tag{124}$$
$$u(x, y) = 0, \ v(x, y) = 0, \qquad x \notin \Gamma_{in} \cup \Gamma_{out}. \tag{125}$$

The Reynolds number of $Re = 100$.

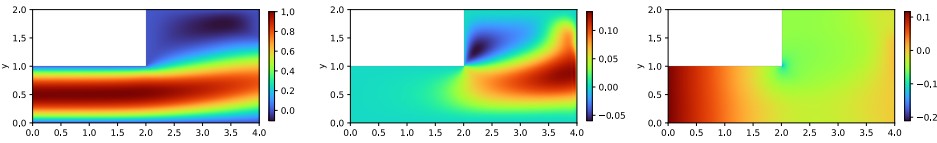

Figure 23: Reference solution of NS-CG, where the left panel shows $u$, the middle pannel shows $v$, and the right panel shows $p$ from the PINNacle reference data.

**Experiment details**  The model architecture consisted of 3 layers with 100 neurons. Training was performed for 50,000 epochs over 5 independent runs with a learning rate of $10^{-3}$. We employed the $\tanh$ activation function, Glorot normal initialization, and the Adam optimizer.

Table 13: Sampling sizes used in our experiments, following the default configuration of PINNacle.

|  | **NS2d-C** | **NS-CG** |
|---|---|---|
| Collocation points | 8192 | 8192 |
| Initial points | 2048 | 2048 |
| Boundary points | 2048 | 2048 |
| Test points | 8192 | 8192 |

With this experimental setting, the results are presented as follows.

Table 14: Average of relative $L^2$ errors (with standard deviation) estimated across 5 random seeds for each algorithm. We highlight **the best** and the second methods for Navier-Stokes equations.

| Method | **NS2d-C** | **NS-CG** |
|---|---|---|
| MultiAdam | 1.3164 (0.3269) | 0.8175 (0.0260) |
| LRA | 0.5027 (0.1197) | 0.2760 (0.0129) |
| ReLoBRaLo | 0.0954 (0.0366) | **0.1212 (0.0093)** |
| MGDA | 0.4316 (0.2851) | 0.1999 (0.0605) |
| PCGrad | **0.0593 (0.0283)** | 0.1229 (0.0171) |
| CAGrad | 0.1372 (0.0945) | 0.2027 (0.0147) |
| IMTL-G | 0.6960 (0.2076) | 0.3463 (0.0984) |
| Aligned-MTL | 0.1980 (0.1516) | 0.1937 (0.0424) |
| ConFIG | 0.9274 (0.4688) | 0.2889 (0.0669) |
| HARMONIC | 0.0767 (0.0639) | 0.1547 (0.0144) |

## F.6  INVERSE PROBLEMS

**Poisson inverse problem (PInv)**

The governing PDE is

$$-\nabla \cdot (a \nabla u) = f(x, y) \tag{126}$$

The source term $f$ is

$$f(x,y) = \frac{2\pi^2 \sin(\pi x)\sin(\pi y)}{1 + x^2 + y^2 + (x-1)^2 + (y-1)^2}$$
$$+ \frac{2\pi\big((2x-1)\cos(\pi x)\sin(\pi y) + (2y-1)\sin(\pi x)\cos(\pi y)\big)}{(1 + x^2 + y^2 + (x-1)^2 + (y-1)^2)^2}. \quad (127)$$

The geometric domain is $(x,y) \in \Omega = [0,1]^2$, and

$$u(x,y) = \sin(\pi x)\sin(\pi y). \quad (128)$$

To ensure the uniqueness of the solution, we impose a boundary condition of $a(x,y)$,

$$a(x,y) = \frac{1}{1 + x^2 + y^2 + (x-1)^2 + (y-1)^2}, \qquad (x,y) \in \partial\Omega. \quad (129)$$

We sample data of $u(x,y)$ with 2500 uniformly distributed $50 \times 50$ points and add Gaussian noise $\mathcal{N}(0, 0.1)$ to it. The ground truth of $a(x,y)$ is

$$a(x,y) = \frac{1}{1 + x^2 + y^2 + (x-1)^2 + (y-1)^2}, \qquad (x,y) \in \Omega. \quad (130)$$

**Heat (Diffusion) inverse problem (HInv)**

The PDE is given by

$$u_t - \nabla \cdot (a\nabla u) = f(x,y,t), \quad (131)$$

where the source function $f(x,y,t)$ is

$$f(x,y,t) = \Big((4\pi^2 - 1)\sin(\pi x)\sin(\pi y)$$
$$+ \pi^2\big(2\sin^2(\pi x)\sin^2(\pi y) - \cos^2(\pi x)\sin^2(\pi y) - \sin^2(\pi x)\cos^2(\pi y)\big)\Big)e^{-t}. \quad (132)$$

The geometric domain is $(x,y,t) \in \Omega \times T = [-1,1]^2 \times [0,1]$, and

$$u = e^{-t}\sin(\pi x)\sin(\pi y). \quad (133)$$

We impose a boundary condition for the diffusion coefficient field as

$$a(x,y) = 2, \partial x \in \Omega. \quad (134)$$

We sample data of $u(x,y,t)$ randomly with 2500 points from the temporal domain $\Omega \times T$ and add Gaussian noise $\mathcal{N}(0, 0.1)$ to it. The ground truth is

$$a(x,y) = 2 + \sin(\pi x)\sin(\pi y), \qquad (x,y) \in \Omega. \quad (135)$$

**Experiment details** The model architecture consisted of 1 layer with 100 neurons for PInv and 3 layers with 50 neurons for HInv. Training was performed for 50,000 epochs over 5 independent runs with a learning rate or $10^{-3}$. We employed the $\texttt{tanh}$ activation function, Glorot normal initialization, and the Adam optimizer.

Table 15: Sampling sizes used in our experiments, following the default configuration of PINNacle.

|  | PInv | HInv |
| --- | --- | --- |
| Collocation points | 8192 | 32768 |
| Initial points | 2048 | 8192 |
| Boundary points | 2048 | 8192 |
| Test points | 8192 | 32768 |

With this experimental setting, the results are presented as follows.

Table 16: Average of relative $L^2$ errors (with standard deviation) estimated across 5 random seeds for each algorithm. We highlight **the best** and the second methods for Inverse problems.

| Method | PInv | HInv |
|---|---|---|
| MultiAdam | 0.2047 (0.0724) | 0.1857 (0.0652) |
| LRA | 0.1324 (0.0273) | 0.0546 (0.0131) |
| ReLoBRaLo | 0.5176 (0.9037) | 1.5999 (0.0576) |
| MGDA | 0.2434 (0.0528) | 2.4224 (1.6765) |
| PCGrad | **0.0909** (**0.0216**) | 0.1269 (0.0267) |
| CAGrad | 0.2080 (0.0484) | 0.6435 (1.0862) |
| IMTL-G | 0.3775 (0.2857) | 0.0515 (0.0105) |
| Aligned-MTL | 0.0915 (0.0194) | 0.0804 (0.0457) |
| ConFIG | 0.1344 (0.0320) | 0.0525 (0.0132) |
| HARMONIC | 0.1183 (0.0271) | **0.0506** (**0.0121**) |

## F.7 VOLTERRA PROBLEMS

**1D nonlinear Volterra**

The IDE is expressed as

$$\frac{du(x)}{dx} + u(x) = \lambda \int_0^x e^{t-x} u(t)\, dt. \tag{136}$$

If we define

$$v(x) = \int_0^x e^{t-x} u(t)\, dt, \tag{137}$$

then the IDE can be written as

$$\frac{du(x)}{dx} + u(x) = \lambda\, v(x). \tag{138}$$

Also, $v(x)$ should satisfy

$$\frac{dv(x)}{dx} = u(x) - \int_0^x e^{t-x} u(t)\, dt = u(x) - v(x). \tag{139}$$

The domain is $x \in \Omega = [0,5]$. The equation has an exact solution

$$u(x) = e^{-x} \cosh(x). \tag{140}$$

The initial condition is

$$u(0) = 1, \tag{141}$$

$$v(0) = 0. \tag{142}$$

The parameter is $\lambda = 1$.

**Volterra system**

The IDEs system is

$$\frac{d^2 u(x)}{dx^2} = 1 - \frac{x^3}{3} - \frac{1}{2}\frac{dv(x)}{dx} + \lambda_1 \int_0^x \left(u^2(t) + v^2(t)\right) dt, \tag{143}$$

$$\frac{d^2 v(x)}{dx^2} = -1 + x^2 - x\, u(x) + \lambda_2 \int_0^x \left(u^2(t) - v^2(t)\right) dt. \tag{144}$$

$$\tag{145}$$

As we define

$$w(x) = \int_0^x \left(u^2(t) + v^2(t)\right) dt, \tag{146}$$

$$p(x) = \int_0^x \left(u^2(t) - v^2(t)\right) dt, \tag{147}$$

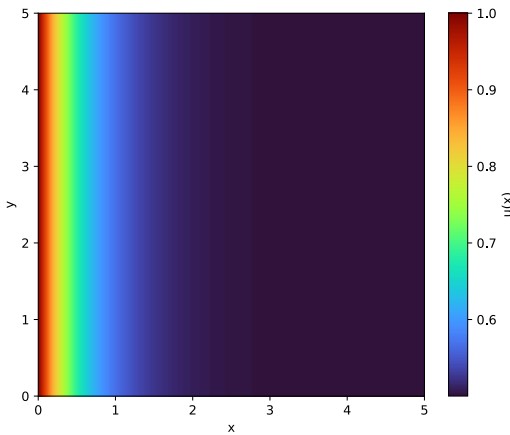

Figure 24: Reference solution of Volterra1d.

we can rewrite IDEs as

$$\frac{d^2 u(x)}{dx^2} = 1 - \frac{x^3}{3} - \frac{1}{2}\frac{dv(x)}{dx} + \lambda_1 w(x), \tag{148}$$

$$\frac{d^2 v(x)}{dx^2} = -1 + x^2 - x\,u(x) + \lambda_2 p(x). \tag{149}$$

$$\tag{150}$$

As follows, auxiliary outputs $w(x)$ and $p(x)$ are governed by

$$\frac{dw(x)}{dx} = u^2(x) + v^2(x), \tag{151}$$

$$\frac{dp(x)}{dx} = u^2(x) - v^2(x). \tag{152}$$

The domain is $x \in \Omega = [0, 1]$. The exact solutions are

$$u(x) = x + e^x, \tag{153}$$
$$v(x) = x - e^x. \tag{154}$$

The initial conditions are

$$\frac{du(0)}{dx} = 2, \frac{dv(0)}{dx} = 0, \tag{155}$$

and

$$u(0) = 1, \ v(0) = -1, \ w(0) = 0, \ p(0) = 0. \tag{156}$$

The parameters are

$$\lambda_1 = \frac{1}{2}, \ \ \lambda_2 = \frac{1}{4}. \tag{157}$$

**Experiment details** The model architecture consisted of 3 layers with 50 neurons for Volterra1d and 5 layers with 100 neurons for Volterra system. Training was performed for 50,000 epochs over 5 independent runs with a learning rate of $10^{-3}$. We employed the tanh activation function, Glorot normal initialization, and the Adam optimizer.

With this experimental setting, the results are presented as follows.

F.8    OTHER PDEs

We conducted experiments on the PDEs included in the PINNacle benchmark, however, we did not evaluate certain PDEs that do not align with the purpose of our study. First, in the two-loss setting, the concept is identical to DCGD(center), and even the computational procedure coincides with that

Table 17: Sampling sizes used in our experiments.

| | 1D Volterra | Volterra system |
|---|---|---|
| Collocation points | 8192 | 8192 |
| Initial points | 2048 | 2048 |
| Boundary points | 2048 | 2048 |
| Test points | 8192 | 8192 |

Table 18: Average of relative $L^2$ errors (with standard deviation) estimated across 5 random seeds for each algorithm. We highlight **the best** and the second methods for Volterra.

| Method | Volterra1d | Volterra system |
|---|---|---|
| MultiAdam | 0.0007 (0.0004) | 0.9998 (0.0000) |
| LRA | 0.0070 (0.0065) | 0.3120 (0.1856) |
| ReLoBRaLo | 0.0036 (0.0044) | 0.0009 (0.0005) |
| MGDA | 0.2806 (0.1263) | 0.0119 (0.0148) |
| PCGrad | 0.0012 (0.0010) | 0.0039 (0.0084) |
| CAGrad | 0.0014 (0.0024) | 0.0059 (0.0092) |
| IMTL-G | 1.3180 (1.3065) | 0.6734 (0.1375) |
| Aligned-MTL | 0.0071 (0.0073) | 0.0268 (0.0575) |
| ConFIG | 0.0018 (0.0024) | 0.1082 (0.1274) |
| HARMONIC | **0.0003** (**0.0001**) | **0.0007** (**0.0008**) |

of ConFIG. Therefore, we excluded Poisson 3D Complex Geometry with Two Domains (Poisson3d-CG), 2D Poisson Equation with many subdomains (Poisson2d-MS), N-Dimensional Poisson Equation (PNd), and the Kuramoto–Sivashinsky Equation (KS), since even when accounting for the governing equation and initial/boundary conditions, they involve only two losses. Second, as reported in PINNacle, the relative $L^2$ error for the 2D Heat Long Time (Heat2d-LT), 2D Navier–Stokes Long Time (NS2d-LT), 2D Wave Equation in Heterogeneous Medium (Wave2d-CG), 2D Multi-Scale Long Time Wave Equation (Wave2d-MS), and the 2D Diffusion–Reaction Gray–Scott Model (GS) typically exceeds 1 or remains close to it across most methods. For these PDEs, we conducted experiments with bot our method and the comparison methods, but the overall performance was unsatisfactory. Since under such circumstances it is not meaningful to claim that one method outperforms another, we chose not to include these results in the report.

## G  DETAILS OF COMPARISON METHODS

We aim to introduce the considerations taken into account for reporting the hyperparameters of each method used in the experiments and for comparing their performance. In addition, we did not perform hyperparameter tuning, since in the multi-loss setting relative superiority cannot be captured by a single quantitative validation metric, making PDE-specific hyperparameter selection unavailable. Accordingly, we adopted the hyperparameter settings that were either commonly used or explicitly recommended in the original papers.

**MultiAdam**  As MultiAdam applies Adam to each loss in a multi-loss setting, it takes the exponential decay rates for the first-moment and second-moment estimates as hyperparameters. In our experiments, we followed the recommendation of MultiAdam (Yao et al., 2023) by setting both hyperparameters to 0.99.

**ReLoBRaLo**  ReLoBRaLo estimates the scaling factors using the following procedure.

$$\lambda_i^{\text{bal}}(t, t') = m \cdot \frac{\exp\left(\dfrac{L_i(t)}{\tau L_i(t')}\right)}{\sum_{j=1}^{m} \exp\left(\dfrac{L_j(t)}{\tau L_j(t')}\right)}, \quad i \in \{1, \ldots, m\} \tag{158}$$

$$\lambda_i^{\text{hist}}(t) = \rho \lambda_i(t-1) + (1-\rho)\lambda_i^{\text{bal}}(t, 0) \tag{159}$$

$$\lambda_i(t) = \alpha \lambda_i^{\text{hist}} + (1-\alpha)\lambda_i^{\text{bal}}(t, t-1) \tag{160}$$

In the above equations, $m$ denotes the number of losses, and $\tau$, $\rho$, and $\alpha$ are hyperparameters. The hyperparameters were set to $\tau = 0.1$, $\rho = 0.9999$, and $\alpha = 0.999$, following the configuration used in the Burgers1d experiments of ReLoBRaLo (Bischof & Kraus, 2025).

**MGDA**  In MGDA, the gradient vectors are first normalized, and the update step is taken in the direction of the minimum-norm point within the convex hull formed by their normalized convex combination. We employed the $\ell_2$ normalization, proposed in MGDA (Sener & Koltun, 2018), which has the advantage of providing a stable definition of the update direction.

**CAGrad**  In CAGrad (Liu et al., 2021a), the hyperparameter $c \in [0, 1)$ constrains the update direction to remain within a certain distance from the average of the loss gradients, and we set $c = 0.5$ in our experiments.

**ConFIG**  In ConFIG, the update direction is originally obtained by solving a least-squares problem to find a vector whose inner product with each loss gradient equals 1. However, this procedure produced NaN values in all five independent runs on NS2d-C, Volterra1d, and Volterra system. Therefore, in our implementation, instead of using the unstable least-squares method, we computed the pseudo-inverse of the loss gradient matrix and multiplied it by $\mathbf{1}_m$ to obtain the update direction instead.

**NTK**  At each iteration, the scaling factors are estimated using the singular values of the kernel matrix, where the kernel matrix is defined as

$$\mathbf{K} = \mathbf{J}\mathbf{J}^\top, \tag{161}$$

and the Jacobian matrix $\mathbf{J}$ has elements corresponding to the derivatives of each loss with respect to the network parameters $\theta$. In practice, our experiments employed networks ranging from 3 layers with 50 neurons to 5 layers with 100 neurons. Thus, computing the Jacobian matrix and subsequently forming the kernel matrix at every iteration would incur a prohibitive computational cost. Moreover, since the number of losses in our experiments can be as large as 10, we considered such computations impractical. In fact, even for the simplest case of the one dimensional Burgers equation, evaluating the Jacobian and constructing the kernel matrix result in an average computational speed of only 0.2 iterations per second, excluding the cost of training itself. By contrast, all other methods except NTK achieved training speeds exceeding 50 iterations per second on average, indicating that the NTK approach is prohibitively inefficient under our experimental setting.

# H ADDITIONAL EXPERIMENTS

## H.1 COMPARISON ACROSS LEARNING RATES

We conducted experiments on the Wave1D-C, Poisson2D-C, HNd, Volterra1D, and HInv problems using two learning-rate configurations: $10^{-3}$, and $10^{-4}$. The results for each setting are reported accordingly.

Table 19: Average of relative $L^2$ errors (with standard deviations) estimated across 5 random seeds for each algorithm. For every learning rate, we highlight **the best** and **the overall best** methods for Wave1d-C.

| Method | $10^{-3}$ | $10^{-4}$ |
|---|---|---|
| MultiAdam | 1.1540 (0.0630) | 1.0958 (0.1004) |
| LRA | 0.2608 (0.0542) | 0.3385 (0.0277) |
| ReLoBRaLo | 0.4102 (0.0476) | 0.3542 (0.0089) |
| MGDA | 0.4627 (0.1359) | 0.3995 (0.0135) |
| PCGrad | 0.2256 (0.0701) | 0.3124 (0.0198) |
| CAGrad | 0.1186 (0.0071) | 0.3313 (0.0428) |
| IMTL-G | 0.7664 (0.4757) | 0.2724 (0.3372) |
| Aligned-MTL | 0.3801 (0.1741) | 0.2302 (0.0514) |
| ConFIG | 0.0668 (0.0279) | **0.1690 (0.0423)** |
| HARMONIC | **0.0655 (0.0293)** | 0.2499 (0.0456) |

Table 20: Average of relative $L^2$ errors (with standard deviations) estimated across 5 random seeds for each algorithm. For every learning rate, we highlight **the best** and **the overall best** methods for Poisson2d-C.

| Method | $10^{-3}$ | $10^{-4}$ |
|---|---|---|
| MultiAdam | 0.6733 (0.0820) | 1.2376 (0.1976) |
| LRA | 0.5712 (0.1595) | 0.6908 (0.0212) |
| ReLoBRaLo | 0.6602 (0.0221) | 0.6629 (0.0208) |
| MGDA | 0.6777 (0.0239) | 0.6840 (0.0192) |
| PCGrad | 0.6918 (0.0190) | 0.9286 (0.2068) |
| CAGrad | 0.7806 (0.1553) | 0.6969 (0.0197) |
| IMTL-G | 0.5039 (0.0684) | 0.9743 (0.6459) |
| Aligned-MTL | 0.2847 (0.2363) | 0.7137 (0.0324) |
| ConFIG | 0.6954 (0.4296) | 0.6856 (0.0274) |
| HARMONIC | **0.0214 (0.0179)** | **0.3579 (0.4681)** |

Table 21: Average of relative $L^2$ errors (with standard deviations) estimated across 5 random seeds for each algorithm. For every learning rate, we highlight **the best** and **the overall best** methods for HNd.

| Method | $10^{-3}$ | $10^{-4}$ |
|---|---|---|
| MultiAdam | 0.0322 (0.0338) | 0.0040 (0.0028) |
| LRA | 0.0020 (0.0011) | 0.0036 (0.0057) |
| ReLoBRaLo | 0.0034 (0.0067) | **0.0004 (0.0000)** |
| MGDA | 0.0048 (0.0088) | 0.0009 (0.0001) |
| PCGrad | 0.0028 (0.0019) | 0.0005 (0.0002) |
| CAGrad | **0.0005 (0.0001)** | 0.0006 (0.0000) |
| IMTL-G | 0.0006 (0.0002) | 0.0006 (0.0001) |
| Aligned-MTL | 0.0006 (0.0002) | 0.0016 (0.0014) |
| ConFIG | 0.0006 (0.0001) | 0.0008 (0.0008) |
| HARMONIC | **0.0005 (0.0000)** | 0.0038 (0.0076) |

Table 22: Average of relative $L^2$ errors (with standard deviations) estimated across 5 random seeds for each algorithm. For every learning rate, we highlight **the best** and **the overall best** methods for Volterra1d.

| Method | $10^{-3}$ | $10^{-4}$ |
|---|---|---|
| MultiAdam | 0.0007 (0.0004) | 0.0004 (0.0002) |
| LRA | 0.0070 (0.0065) | 0.0080 (0.0039) |
| ReLoBRaLo | 0.0036 (0.0044) | **0.0002 (0.0000)** |
| MGDA | 0.2806 (0.1263) | 0.0011 (0.0019) |
| PCGrad | 0.0012 (0.0010) | 0.0004 (0.0004) |
| CAGrad | 0.0014 (0.0024) | 0.0017 (0.0027) |
| IMTL-G | 1.3180 (1.3065) | 0.0735 (0.0989) |
| Aligned-MTL | 0.0071 (0.0073) | 0.0037 (0.0044) |
| ConFIG | 0.0018 (0.0024) | 0.0042 (0.0069) |
| HARMONIC | **0.0003 (0.0001)** | 0.0004 (0.0001) |

Table 23: Average of relative $L^2$ errors (with standard deviations) estimated across 5 random seeds for each algorithm. For every learning rate, we highlight **the best** and **the overall best** methods for HInv.

| Method | $10^{-3}$ | $10^{-4}$ |
|---|---|---|
| MultiAdam | 0.1857 (0.0652) | 0.1446 (0.0245) |
| LRA | 0.0546 (0.0131) | 0.0626 (0.0061) |
| ReLoBRaLo | 1.5999 (0.0576) | 1.7514 (0.0447) |
| MGDA | 2.4224 (1.6765) | 2.8915 (1.6877) |
| PCGrad | 0.1269 (0.0267) | 0.0660 (0.0152) |
| CAGrad | 0.6435 (1.0862) | 2.2489 (0.4788) |
| IMTL-G | 0.0515 (0.0105) | 0.0498 (0.0075) |
| Aligned-MTL | 0.0804 (0.0457) | 0.1902 (0.0532) |
| ConFIG | 0.0525 (0.0132) | 0.0466 (0.0068) |
| HARMONIC | **0.0506 (0.0121)** | **0.0461 (0.0098)** |

## H.2 COMPARISON UNDER TWO LOSS COMPONENTS

We conducted experiments using two loss configurations: the average of PDE losses, and the average of initial and boundary condition losses.

Table 24: Average relative $L^2$ errors (with standard deviations) estimated across 5 random seeds for each algorithm.

| Method | Wave1d-C | Poisson2d-C | HNd | Volterra1d | HInv |
|---|---|---|---|---|---|
| MultiAdam | 1.0020 (0.0035) | 1.3607 (0.0003) | 14.0535 (1.7338) | 0.8254 (0.0009) | 0.1221 (0.0359) |
| LRA | 0.3537 (0.0214) | 0.4903 (0.1593) | 0.0010 (0.0005) | 0.0010 (0.0010) | 0.0687 (0.0094) |
| ReLoBRaLo | 0.4566 (0.0385) | 0.6948 (0.0173) | 0.0010 (0.0016) | 0.0025 (0.0037) | 1.4040 (0.0415) |
| MGDA | 0.4282 (0.0049) | 0.6927 (0.0223) | 0.0046 (0.0030) | 0.0037 (0.0043) | 1.4780 (0.0726) |
| PCGrad | 0.4065 (0.0120) | 0.5727 (0.1399) | 0.0012 (0.0016) | 0.0003 (0.0002) | 0.1587 (0.0259) |
| CAGrad | 0.3692 (0.0236) | 0.6967 (0.0174) | 0.0013 (0.0019) | 0.0008 (0.0010) | 2.0905 (0.2565) |
| IMTL-G | 0.3593 (0.0312) | 0.7119 (0.0168) | 0.0004 (0.0000) | 0.0026 (0.0036) | 0.0514 (0.0135) |
| Aligned-MTL | 0.4865 (0.1849) | 0.6572 (0.0664) | 0.0005 (0.0000) | 0.0028 (0.0036) | 0.2769 (0.4351) |
| ConFIG | 0.3979 (0.0374) | 0.3991 (0.3600) | 0.0005 (0.0001) | 0.0028 (0.0057) | 0.0521 (0.0120) |
| DCGD | 0.4022 (0.0309) | 0.4280 (0.3755) | 0.0005 (0.0001) | 0.0008 (0.0015) | 0.0515 (0.0122) |
| HARMONIC | 0.3882 (0.0215) | 0.4242 (0.3757) | 0.0005 (0.0001) | 0.0002 (0.0001) | 0.0521 (0.0119) |

As can be seen in Table 24, under the two-loss setting, DCGD, ConFIG, and HARMONIC all lie within the harmonized cone, resulting in highly similar performance across the methods.

### H.3 SCALABILITY WITH MORE LOSS COMPONENTS

In addition to the problems presented in the main text(Wave1D, Poisson2D-C, HNd, Volterra1D, and HInv) we also evaluate the computational cost of our method on NS2D-C, which involves eight loss components.

Table 25: Average computational costs (in seconds) with standard deviations from 100 epochs across 100 repeats. **The worst** methods are highlighted.

| Method | NS2d-C |
|---|---|
| MultiAdam | 5.6655 (0.1410) |
| LRA | 6.1194 (0.1649) |
| ReLoBRaLo | 1.2052 (0.0556) |
| MGDA | **9.8877** (**1.6511**) |
| PCGrad | 6.1200 (0.3040) |
| CAGrad | 6.6949 (0.1894) |
| IMTL-G | 5.5782 (0.3040) |
| Aligned-MTL | 5.6768 (0.1596) |
| ConFIG | 5.6790 (0.1842) |
| HARMONIC | 5.6746 (0.1412) |

As shown in Table 25, even with a large number of losses, the computational cost remains comparable to or even lower than that of other methods.

# I    LLM USAGE

Large Language Models (LLMs) were used in a limited capacity to aid in the preparation of this manuscript. Specifically, LLMs were employed solely for writing support tasks such as grammar checking, polishing sentence structure, and improving readability of author-written drafts. They were not involved in research ideation, methodological design, data analysis, or result interpretation. The authors take full responsibility for the content of this paper.

