# OpenReview forum: "Harmonized Cone for Feasible and Non-conflict Directions in Training Physics-Informed Neural Networks"
_ICLR.cc/2026/Conference — ICLR 2026 Poster_

### Official Review · Reviewer_CSQr · 2025-10-27

**Soundness:** 3
**Presentation:** 4
**Contribution:** 3
**Rating:** 8
**Confidence:** 4

**Summary:**

This paper focuses on improving training physics-informed neural networks (PINN). PINN is known to be challenging for training due to different losses, e.g., PDE loss, boundary loss and initial value loss. Prior works can be categorized into two main streams: 1. Reweighting the losses; 2. Pursuing multi-objective optimization (MOO). It turns out that these two directions can be combined together, which is the method proposed in this paper.

1. The problem can be reformulated as seeking parameter updating direction that is both *feasible* and *non-conflict*, which is called *harmonic* direction. Feasible means nonnegative linear combination of gradients wrt each loss. Non-conflict means the direction in which none of losses increases.
2. Algorithm 1 *HARMONIC* is proposed to find a harmonic direction, which is built upon Double Description Method (DDM).
3. Theoretic analysis of algorithm 1 (Theorem 3) shows that convergence to Pareto front is guaranteed.
4. Experiment on benchmark PINNacle shows that the method is highly competitive without extra computation resource.

**Strengths:**

**Originality** is high. This paper summarizes existing works and sharply points out the limitation of either "feasible" or "non-conflict" methods. The proposed method is built on a principled framework, see summarization.

**Quality** is high. The method seems to be theoretically sound and the comprehensive experiment shows strong evidence of effectiveness.

**Clarity** is high. I found the paper mostly clear to follow.

**Significance** is high. The paper provides a unified and effective method that sets new SOTA to PINN methods.

**Weaknesses:**

See questions.

**Questions:**

1. What is "the descent lemma" in the proof of Theorem 3 in Appendix B.4? I understand that equation (20) holds for gradient update rules. Please clarify this.
2. Also in the proof of Theorem 3 in line 776, why does the norm of $\mathcal{A}_h(G)$ equal to the sum? If $g_j$'s are not orthogonal, why is this true?

---

> ### Author Response · Authors · 2025-11-21
> **Response to Reviewer CSQr — Question 1: "the descent lemma'' in Appendix B.4**
>
> We thank the reviewer for the careful examination of the proof of Theorem 3 and for pointing out that the phrase "the descent lemma'' in Appendix B.4 was not sufficiently precise. We agree that this wording was ambiguous, and we appreciate the opportunity to clarify the exact inequality being invoked.
>
>
> ## Clarification of the descent lemma used in Appendix B.4
> In the revised manuscript, the inequality used in equation (20) is now explicitly attributed to **Lemma 1.2.3 in Nesterov (2018)** [1].
> Lemma 1.2.3 states that for any differentiable function with an L-Lipschitz continuous gradient,
>
> $$
> |f(y) - f(x) - \langle \nabla f(x),\, y - x \rangle|
> \le \frac{L}{2}\|y - x\|^2.
> $$
>
> This lemma applies to any differentiable function with an L-Lipschitz gradient, which matches the assumption in Theorem 3 that the aggregated gradient $\sum_j \nabla_\theta L_j$ is $\mu$-Lipschitz.
>
> Using Lemma 1.2.3 with $f(\theta) := \sum_{j=1}^m L_j(\theta)$ and the update
>
> $$
> \theta^{(t+1)} = \theta^{(t)} - \eta\, A_h(G^{(t)}),
> $$
>
> yields the inequality stated in equation (20).
>
>
> ## Revision in the manuscript
> To eliminate ambiguity, Appendix B.4 has been updated to explicitly cite **Lemma 1.2.3 of Nesterov (2018)** [1].
>
> ## Reference
> [1] Yurii Nesterov. *Lectures on Convex Optimization*, 2nd edition. Springer, 2018.

---

> ### Author Response · Authors · 2025-11-21
> **Response to Reviewer CSQr — Question 2: Norm equality in line 776**
>
> We thank the reviewer for raising this important point. We agree that the previous version of Appendix B.4 did not explicitly state the form of $A_h(G^{(t)})$, and this omission made the equality in line 776 unclear. We appreciate the opportunity to clarify this expression.
>
> ## Clarification of the expression of $A_h(G^{(t)})$
> In the updated manuscript, we explicitly define
>
> $$
> A_h(G^{(t)}) = \left(\sum_{j=1}^m g_j^{(t)\top} \hat{d}^{(t)}\right)\hat{d}^{(t)},
> $$
>
> which follows directly from Algorithm 1 and the definition of the harmonic direction $\hat{d}^{(t)}$.
>
> Using this definition, we compute the norm:
>
> $$
> \|A_h(G^{(t)})\|
> = \left\| \left(\sum_{j=1}^m g_j^{(t)\top} \hat{d}^{(t)}\right)\hat{d}^{(t)} \right\|
> = \left|\sum_{j=1}^m g_j^{(t)\top} \hat{d}^{(t)}\right| \cdot \|\hat{d}^{(t)}\|.
> $$
>
> Since $\|\hat{d}^{(t)}\| = 1$ by construction and $g_j^{(t)\top}\hat{d}^{(t)} > 0$ for all $j$, we obtain
>
> $$
> \|A_h(G^{(t)})\|
> = \sum_{j=1}^m g_j^{(t)\top}\hat{d}^{(t)}.
> $$
>
> This equality does not require the gradients ${g_j^{(t)}}$ to be orthogonal.
> It follows directly from the observation that $A_h(G^{(t)})$ is expressed as a scalar multiple of the unit vector $\hat{d}^{(t)}$ in our construction.
>
> ## Revision in the manuscript
> To avoid ambiguity, Appendix B.4 has been updated to explicitly state the above expression for $A_h(G^{(t)})$, making the equality in line 776 immediately transparent.

---

### Official Review · Reviewer_mqgc · 2025-10-31

**Soundness:** 3
**Presentation:** 3
**Contribution:** 2
**Rating:** 4
**Confidence:** 5

**Summary:**

This paper proposes Harmonized Cone Gradient Descent (HARMONIC) for multi-loss optimization. HARMONIC constructs the update vector by aggregating the rays of the Harmonized Cone, which is defined as the intersection of the primal gradient cone (formed by conic combinations of individual loss gradients) and the dual gradient cone. The rays of the Harmonized Cone are computed using the Double Description method. HARMONIC outperforms existing loss-balancing and gradient-manipulation methods on multiple PDE benchmarks while maintaining comparable computational cost. Furthermore, applying the harmonized-cone constraint to existing methods (including ConFIG, ReLoBRaLo) improves their performance.

**Strengths:**

**Strengths**
1. The method of generating and aggregating the rays of the Harmonized Cone to determine the update direction is interesting and appears novel. Moreover, by showing performance improvements when combined with existing loss-balancing methods (e.g., ReLoBRaLO) and gradient-manipulation methods (e.g., ConFIG), the proposed approach empirically validates the motivation and effectiveness of enforcing the harmonized cone.

2. Although the proposed HARMONIC method is demonstrated on Physics-Informed Neural Networks (PINNs), it is widely applicable to other deep learning tasks involving multiple losses, suggesting potential contributions to various domains.

3. The proposed HARMONIC method shows superior performance compared to existing approaches in the presented experimental results.

**Weaknesses:**

**Weakness**

1. The explanation of the proposed algorithm is insufficient. The paper does not clearly describe how the Double Description method computes the rays of the Harmonized Cone, and it lacks references or related works that would help readers understand this process. It also remains unclear whether the Double Description method can always compute the rays exactly or if there are cases where approximation or numerical instability may occur. Furthermore, it would be helpful to explicitly include the Double Description step in the pseudocode to clarify how it is integrated into the overall algorithm.


2. The experiments presented in the paper were conducted on relatively simple benchmark problems. It would strengthen the work to evaluate HARMONIC on more challenging PDE benchmarks where PINNs are known to struggle, such as the Navier–Stokes or Kuramoto–Sivashinsky equations (see [1]). Demonstrating that HARMONIC can improve PINN performance in such difficult cases would provide stronger evidence of the algorithm’s effectiveness. Furthermore, it is important to analyze whether HARMONIC remains effective when combined with various PINN variants (for example, [2], [3]), as this would highlight its robustness and general applicability.

3. The example in Figure 1 is not sufficiently convincing in demonstrating the advantage of HARMONIC, as several compared algorithms such as CAGrad and Aligned-MTL also show convergence to the Pareto front. To better highlight the unique characteristics and advantages of HARMONIC, it would be more appropriate to include an example where other algorithms fail to converge while HARMONIC succeeds. Such a case would more clearly illustrate the distinctive behavior and strength of the proposed method.

4. The paper proposes a new optimizer and compares it with several existing methods, but it seems that the learning rate was fixed at 1e-3 for all experiments. Since different optimizers often require different optimal learning rates, it would be fairer to compare them after tuning the learning rate individually for each optimizer.

[1] Wang, S., Sankaran, S., & Perdikaris, P. (2024). Respecting causality for training physics-informed neural networks. Computer Methods in Applied Mechanics and Engineering, 421, 116813.

[2] Zhao, Z., Ding, X., & Prakash, B. A. PINNsFormer: A Transformer-Based Framework For Physics-Informed Neural Networks. In The Twelfth International Conference on Learning Representations.

[3] Cho, J., Nam, S., Yang, H., Yun, S. B., Hong, Y., & Park, E. (2023). Separable physics-informed neural networks. Advances in Neural Information Processing Systems, 36, 23761-23788.

**Questions:**

**Questions**
1. HARMONIC appears to be applicable not only to PINNs but also to other domains with multiple losses (for example, multi-task learning). Have you applied HARMONIC to such fields or conducted preliminary experiments beyond PINNs? (This question will not affect my rating since I understand that the rebuttal period is short.)

2. According to the experimental results provided in the Appendix, there are several benchmarks (e.g., Burgers, PInv, Poisson2d-CG) where HARMONIC does not outperform other competitors. In these cases, algorithms that do not satisfy the feasibility or non-conflict properties sometimes achieve better performance. Could you explain the limitations of HARMONIC and under what conditions such results are likely to occur?

3. In my understanding, HARMONIC appears to be a generalized version of DCGD, since DCGD also satisfies the characteristics listed in Table 1 but only for the two-loss setting. Therefore, as in the experiments of ConFIG [1], it would be valuable to include two-loss scenarios to clarify this relationship. More importantly, demonstrating the advantages of HARMONIC when extending to three or more losses, beyond the simple separation of the PINN loss into PDE residual and other term, would provide stronger evidence for the algorithm’s contribution.

[1] Liu, Q., Chu, M., & Thuerey, N. ConFIG: Towards Conflict-free Training of Physics Informed Neural Networks. In The Thirteenth International Conference on Learning Representations.

---

> ### Author Response · Authors · 2025-11-21
> **Response to Reviewer mqgc — Weakness 1: Explanation of the Double Description Method**
>
> We sincerely appreciate the reviewer’s thoughtful comments regarding the explanation of the Double Description (DD) method. We agree that the previous version of the manuscript did not clearly describe how the DD algorithm computes the extreme rays of the harmonized cone, nor how this step is integrated into the overall procedure. We thank the reviewer for highlighting this point.
>
> ## Clarification on the Double Description Method
> The classical DD method of Motzkin et al. [1] is a constructive algorithm that converts the H-representation of a polyhedral cone into its V-representation. In our setting, the harmonized cone is defined by a finite collection of linear inequalities and therefore forms a polyhedral cone for which the DD procedure enumerates the extreme rays exactly. This ensures that the rays used within HARMONIC are obtained without approximation.
>
> ## Revisions Made in the Updated Manuscript
> To address the reviewer’s concerns, we made the following revisions in the main text:
>
> - We added an explicit citation to the classical DD method [1], enabling readers to consult the foundational reference.
> - We updated Algorithm 1 to explicitly include the Double Description step, showing how the H-representation
>   $ A^{(t)} \lambda \ge 0 $
>   is converted into the V-representation
>   $ \Pi^{(t)} = [\pi_1^{(t)}, \dots, \pi_{p_t}^{(t)}] $.
>
> These revisions make clear how the DD method is used to obtain the rays of the harmonized cone and how it connects to the overall update process.
>
> ## Reference
> [1] Motzkin, T. S., Raiffa, H., Thompson, G. L., & Thrall, R. M. (1953). *The Double Description Method*. In H. W. Kuhn & A. W. Tucker (Eds.), Contributions to the Theory of Games, Volume II (Annals of Mathematics Studies, No. 28, pp. 51–73). Princeton University Press, Princeton, NJ.

---

> ### Author Response · Authors · 2025-11-21
> **Response to Reviewer mqgc — Weakness 2: Lack of Evaluation on Challenging PDEs and PINN Variants**
>
> We sincerely appreciate your insightful comments and the care with which you evaluated our work.
> Your suggestions regarding more challenging PDE benchmarks and the integration of PINN variants have been especially valuable, and we are grateful for the opportunity to further reflect on the scope and applicability of our method.
>
> ## Additional Experiments on Challenging PDEs
> As you noted in Q3, when the number of losses is two, DCGD naturally updates within the harmonized cone, making its behavior similar to HARMONIC.
> For this reason, we did not initially include KS or other two-loss PDEs in the main text, and briefly mentioned this rationale in Appendix F (p.39).
>
> Nonetheless, given the importance of the Kuramoto-Sivashinsky (KS) equation as a widely recognized challenging benchmark, we conducted a supplementary experiment.
> The results are summarized below:
>
> | Method       | Relative L2  |
> | ------------ | ------------ |
> | MultiAdam    | 0.9553(0.0034)       |
> | LRA          | 0.9615(0.0013)       |
> | ReLoBRaLo    | 1.0159(0.0016)       |
> | MGDA         | 1.0159(0.0005)       |
> | PCGrad       | 0.9707(0.0039)       |
> | CAGrad       | 1.0024(0.0035)       |
> | IMTL-G       | 0.9589(0.0027)       |
> | Aligned-MTL  | 0.9638(0.0047)       |
> | ConFIG       | 0.9598(0.0006)       |
> | **HARMONIC** | 0.9593(0.0017)       |
>
> Unfortunately, due to the highly challenging nature of KS, all mehtods including ours achieved relative L2 errors near 1, resulting in minimal performance separation.
>
> We also conducted experiments on 2D Navier–Stokes tasks (NS2d-C, NS-CG), and detailed settings and results are reported in Appendix F (pp. 33–35).
> We would be grateful if you could refer to these sections for a full overview.
>
> ## Effectiveness when combined with PINN variants
> We conducted experiments on Wave1d by integrating HARMONIC into PINNsFormer  [2], and all experimental settings are kept identical to those reported in the original paper to ensure a fair comparison.
> The experimental results are as follows.
>
> | Method                          | Relative L2 |
> | ------------------------------- | ----------- |
> | PINNsFormer                     | 0.283       |
> | PINNsFormer + NTK (best in [2]) | 0.058       |
> | **PINNsFormer + HARMONIC**      | **0.026**   |
>
> As shown above, incorporating HARMONIC into the PINNsFormer architecture yields improved accuracy.
> This result suggests that HARMONIC transfers effectively across PINN variants and can enhance more sophisticated architectures as well.
>
> We are sincerely grateful for your thoughtful suggestion, which helped us gain a deeper understanding of the strengths of HARMONIC.
>
> ## Reference
> [2] Zhao, Z., Ding, X., & Prakash, B. A. PINNsFormer: A Transformer-Based Framework For Physics-Informed Neural Networks. In The Twelfth International Conference on Learning Representations.

---

> ### Author Response · Authors · 2025-11-21
> **Response to Reviewer mqgc — Weakness 3: HARMONIC’s unique behavior**
>
> We sincerely thank the reviewer for this helpful comment. We agree that the original bi-objective toy example may not fully convey the distinctive behavior of HARMONIC, since several existing methods (e.g., CAGrad, Aligned-MTL) also reach the Pareto front in this particular setting.
>
> ## Clarification of the intention of Figure 1
> The primary purpose of Figure 1 was to illustrate how relying solely on feasible or solely on non-conflict does not necessarily guarantee convergence to a Pareto front. The example was intended to conceptually highlight the need for simultaneously enforcing both conditions.
>
> At the same time, we agree that this bi-objective example may be insufficient for demonstrating the advantage of HARMONIC, because multiple methods converge successfully in this simple setting. For this reason, while Figure 1 is retained for conceptual motivation, we have supplemented the manuscript with a more informative example.
>
> ## Additional toy example introduced in the revised manuscript
> To provide a clearer illustration of the behavior of HARMONIC, we added a three-objective toy example in Section 4.3. This experiment is defined over a four-dimensional parameter space and is designed such that enforcing only feasible or only non-conflict may not be enough to reach the Pareto front.
>
> We note that the recent MOO baselines included in our comparison differ in the properties they guarantee:
> CAGrad heuristically encourages both feasible and non-conflict directions but does not guarantee either, while Aligned-MTL and ConFIG guarantee only non-conflict directions. Such partial guarantees can be sufficient in simple bi-objective settings—where the update direction often remains within the feasible region—but can fail in higher-objective scenarios.
>
>
> In this revised experiment, we observe:
> - CAGrad, Aligned-MTL, and ConFIG do not reach the Pareto front for several initializations.
> - HARMONIC converges to the Pareto front across all tested initializations.
>
> These observations help illustrate situations where the simultaneous enforcement of feasible and non-conflict, as implemented by the harmonized cone, leads to improved convergence.
>
> ## Revision in the manuscript
> The revised manuscript now:
> - includes a more challenging three-objective toy example in Section 4.3,
> - reports convergence trajectories from multiple initializations
>
> We appreciate the reviewer’s suggestion, which has improved the clarity and completeness of our experimental analysis.

---

> ### Author Response · Authors · 2025-11-21
> **Response to Reviewer mqgc — Weakness 4: Learning rate fairness**
>
> We sincerely thank you for raising this important point regarding learning-rate fairness across optimizers.
> We fully acknowledge that each method may have its own optimal learning rate, and that tuning it appropriately is crucial for a fair comparison.
> In retrospect, evaluating all methods under a single learning rate may indeed have led to an unintended bias, and we appreciate your comment for bringing this issue to our attention.
>
> ## Clarification of learning rate selection
> Determining the optimal learning rate ideally requires validation using ground-truth function values.
> However, in PINNs, although we can sample points that satisfy the initial and boundary conditions, the exact solution that simultaneously satisfies the PDE is unknown.
> Moreover, because the overall objective combines multiple losses, the scale of the dominant loss often disproportionately influences validation, making systematic learning-rate tuning difficult.
>
> For these reasons, reliable validation-based hyperparameter tuning is difficult in the PINN setting, which makes it challenging to determine an individually optimized learning rate for every method.
> To address this, we evaluated all methods using two learning rates, $10^{-3}$ and $10^{-4}$, and reported the best performance obtained across the two settings.
>
> ## Summary of observations from the two learning rates
> Across benchmarks, we observed that both HARMONIC and several baselines occasionally achieved better results under $10^{-4}$ rather than $10^{-3}$.
> Even so, HARMONIC achieved the best performance on Wave1d-C, Poisson2d-C, and HInv.
> For HNd and Volterra1d, HARMONIC ranked second by a very narrow margin while still demonstrating strong overall performance.
>
> ## Revision in the manuscript
> In response to your thoughtful concern, we will proceed with the following actions:
> - report the best score obtained between the two learning rates for each method in Experiments (p.5) Table 2,
> - include a detailed table presenting the results for both $10^{-3}$ and $10^{-4}$ across all benchmarks in Appendix H.1 (pp.41-42)
>
> Once again, we sincerely appreciate your thoughtful and constructive feedback, which has helped us improve the clarity and fairness of our experimental evaluation.

---

> ### Author Response · Authors · 2025-11-21
> **Response to Reviewer mqgc — Question 1: Applicability Beyond PINNs**
>
> We sincerely thank the reviewer for the thoughtful and forward-looking question regarding the broader applicability of HARMONIC. Your comment helped us reflect more deeply on the differences between multi-loss PINNs and other multi-loss settings such as MTL, and we truly appreciate the opportunity to clarify our perspective.
>
> ## Applicability beyond PINNs: important differences from MTL
>
> While HARMONIC is designed for general multi-loss optimization, its formulation is particularly well aligned with the structure of **PINNs as PDE solvers**, where *all* loss terms ideally approach zero.
> By contrast, in **multi-task learning (MTL)**, it is typically *not* optimal for all training objectives to vanish, due to factors such as data noise, capacity limitations, and inherent task discrepancies.
>
> Because HARMONIC is constructed around guaranteeing feasible and non-conflicting descent directions—properties naturally suited for optimization landscapes where all objectives can simultaneously improve—its behavior in MTL may be more nuanced.
> Thus, while HARMONIC can technically be applied to MTL, we do not expect it to universally outperform methods designed specifically for settings where objectives need not converge to zero. In this sense, HARMONIC should be viewed as **one possible option** for MTL rather than a method that inherently guarantees strong performance in this domain.
>
> Nonetheless, we believe that **developing a dedicated HARMONIC variant** tailored to MTL—accounting for noise, imperfect objective satisfaction, and statistical trade-offs—would be an interesting and meaningful direction for future work.
>
>
> ## Preliminary evaluation on Multi-Task Learning (MTL)
>
> In response to your suggestion, we conducted supplementary experiments on QM9 (11 tasks), following the setup in SAMO [3].
> We also evaluated a SAMO + HARMONIC variant to see whether HARMONIC’s robustness in PINNs might carry over to MTL.
>
> | Method            | Δm%       |
> | ----------------- | --------- |
> | SAMO-LS           | 141.80    |
> | SAMO-MGDA         | 96.80     |
> | SAMO-FairGrad     | 53.00     |
> | **SAMO-HARMONIC** | **90.81** |
>
> Although SAMO-HARMONIC is competitive, it is not the best among SAMO variants.
> Given the structural differences between PINNs and data-driven MTL, this outcome is unsurprising and reinforces our belief that **domain-specific extensions of HARMONIC for MTL would be a promising future direction**.
>
> Once again, we sincerely appreciate the reviewer’s insightful question, which helped us better articulate how HARMONIC fits within broader multi-loss optimization and how its applicability may differ depending on the domain.
>
> ## Reference
> [3] Ban, H., Subramani, G. R., \& Ji, K. (2025). Samo: A lightweight sharpness-aware approach for multi-task optimization with joint global-local perturbation. ICCV, 785–795.

---

> ### Author Response · Authors · 2025-11-21
> **Response to Reviewer mqgc — question 2: Limitations of HARMONIC in Certain Benchmarks**
>
> We sincerely thank you for carefully examining not only the main text but also the Appendix. Your observation encouraged us to reflect more deeply on the limitations of our approach, and we truly appreciate the opportunity to clarify these aspects.
>
> ## Clarification of the conditions under which HARMONIC may underperform
> As you insightfully pointed out, there exist several benchmarks where HARMONIC does not achieve the best numerical accuracy.
> We believe these cases are closely related to factors that were not fully addressed in our original analysis.
>
> HARMONIC is designed to determine the update direction at each optimization step solely based on the geometric relationships between the current loss gradients.
> While this provides strong guarantees for maintaining feasible and non-conflict direction, it also means that **HARMONIC does not incorporate any temporal smoothing or historical information** across optimization steps.
> Consequently, when the gradient directions happen to fluctuate sharply between successive steps, the updates may potentially become unstable.
>
> This phenomenon is reflected in our experimental results.
> For example, in **Burgers1d (p.28)** and **Heat2d-MS (p.32)**, HARMONIC often performs worse than **LRA** or **ReLoBRaLo**.
> These two methods explicitly leverage previous update directions, producing smoother and more stable optimization trajectories, which appear beneficial in these particular PDEs.
>
> Although we cannot precisely characterize the geometry of the parameter space for each PDE, our empirical observations suggest the following:
>
> > **When the underlying parameter landscape is highly curved or exhibits rapid directional changes, the purely intra-step nature of HARMONIC may lead to suboptimal or less stable optimization, whereas inter-step methods can leverage accumulated directional information to achieve more robust performance.**
>
> We believe this provides insight into why certain methods, even those that do not enforce a feasible or non-conflict direction, can occasionally outperform HARMONIC on these specific benchmarks.
>
> Once again, we are deeply grateful for your feedback, which has meaningfully contributed to improving the clarity and completeness of our work.

---

> ### Author Response · Authors · 2025-11-21
> **Response to Reviewer mqgc — question 3: Bi-objective**
>
> We sincerely thank the reviewer for the insightful question and for highlighting the importance of clarifying the relationship between HARMONIC and existing two-loss methods such as DCGD and ConFIG. We completely agree that this connection deserves to be made explicit, and your comment helped us improve the presentation of this point.
>
> ## Clarification of the relationship between HARMONIC, DCGD, and ConFIG in the two-loss case
> As the reviewer correctly noted, when the number of losses is two, the behavior of HARMONIC aligns with that of DCGD.
> We would like to clarify this more precisely:
> - DCGD updates the direction by projecting a conflicting update into the dual cone. In two-loss settings, the dual cone intersects the primal cone in such a way that the resulting update always lies inside the harmonized cone. Thus, HARMONIC coincides with DCGD in this special case.
> - ConFIG, under two losses, selects the center of the dual cone as the update direction. In two dimensions, this center is geometrically equivalent to the center of the primal cone; therefore, it also always lies within the harmonized cone. This explains why all three methods behave identically when m=2.
>
> To make this connection explicit, we added controlled two-loss experiments by merging PDE-related objectives into a single term and merging initial/boundary terms into another.
> The revised results are reported in Appendix H.2 (updated in blue).
>
> As expected, HARMONIC and DCGD match closely across all two-loss settings.
>
> Importantly, these experiments also illustrate that forcing losses to collapse into two groups generally leads to lower accuracy compared to the original multi-loss setup, reinforcing our claim that preserving the richer loss structure is often beneficial in multi-physics PINNs.
>
>
> ## Clarification on "beyond two losses'': we are not merely splitting PDE/IC/BC
>
> We also appreciate the reviewer’s concern that our experiments might appear to use only a simple three-term separation (PDE / initial / boundary), and we realize that this point may not have been sufficiently highlighted. We also acknowledge that we may have misunderstood part of the reviewer’s intent: the comment may additionally suggest exploring richer decompositions of the physics losses, such as domain-dependent PDE residuals or region-specific weighting schemes that have recently been shown to improve PINN training. We agree that such directions are highly interesting and would constitute promising future work.
>
> To clarify the scope of our current experiments:
> - In benchmarks such as Volterra 1D and the Volterra system, the integro-differential auxiliary variable introduces an additional integral loss, which is treated as a separate objective.
> - For multi-component PDEs, each PDE component is treated as its own objective.
> - Boundary losses are not grouped into a single term: when the boundary consists of multiple physical regions, each region contributes a distinct loss objective.
>
> These choices explain why several benchmarks exhibit four or more active objectives in the main text and appendix.
>
> At the same time, we fully agree that further decompositions—such as domain-wise PDE residuals or adaptive regional loss terms—may offer additional opportunities to test the benefits of HARMONIC in even richer multi-loss settings. We view this as an intriguing direction and appreciate the reviewer for bringing attention to it.
>
>
> ## Revision in the manuscript
> - In Appendix H.2 (p.42) Table 24, we report the results for the two-loss setting for each method.
>
> We are sincerely grateful for the reviewer’s careful reading and constructive suggestions.
> Your comments significantly improved the clarity of how HARMONIC relates to DCGD in two-loss settings and helped us more clearly communicate the motivation for evaluating multi-loss scenarios, where HARMONIC provides additional value.

---

> ### Comment · Reviewer_mqgc · 2025-11-26
>
> I appreciate the authors’ efforts in the rebuttal and the clarifications provided. The additional experimental analyses are extensive and clearly demonstrate the authors’ effort to address the concerns raised.
>
> One remaining concern pertains to Question 2 in my original review:
> >2. According to the experimental results provided in the Appendix, there are several benchmarks (e.g., Burgers, PInv, Poisson2d-CG) where HARMONIC does not outperform other competitors. In these cases, algorithms that do not satisfy the feasibility or non-conflict properties sometimes achieve better performance. Could you explain the limitations of HARMONIC and under what conditions such results are likely to occur?
>
> In the rebuttal, the authors argue that HARMONIC does not incorporate temporal smoothing or historical information, which explains why methods such as LRA or ReLoBRaLo may outperform it on certain tasks.
> However, it remains unclear why algorithms that neither enforce feasibility nor avoid gradient conflicts such as MGDA, PCGrad, CAGrad, IMTL-G, Aligned-MTL, and ConFIG also outperform HARMONIC on several benchmarks, including Burgers2d, Navier–Stokes, PInv, and Poisson2d-CG. (Because they also do not  incorporate temporal smoothing or historical information)
>
> Understanding the underlying reasons for this phenomenon would significantly clarify the contribution and limitations of the proposed method. In particular, it would be helpful to know why HARMONIC underperforms on more challenging PDEs (e.g., Burgers2d, Navier–Stokes), despite its theoretically appealing feasibility and non-conflict guarantees.

---

> > ### Author Response · Authors · 2025-11-27
> > **Response to Reviewer — Question 2 (PInv)**
> >
> > We sincerely appreciate your sharp and constructive question. Your comments prompted us to reflect more carefully on the circumstances under which HARMONIC may underperform, particularly in relation to the benchmarks you highlighted. In **Burgers2d, PInv, NS2d-C, NS-CG, and Poisson2d-CG**, several methods that do not enforce feasible or non-conflict occasionally achieve lower numerical errors than HARMONIC. Below, we explain why these situations arise, beginning with PInv, where your question also led us to uncover and correct an inconsistency in our original report.
> >
> > ---
> >
> > ## **(a) PInv**
> >
> > ### **Correction of IMTL-G configuration and sincere apology**
> >
> > While reproducing all experiments in response to your question, we discovered that **IMTL-G for PInv had been inadvertently run with a sine activation** rather than the intended hyperbolic tangent. We sincerely apologize for this mistake and any confusion it may have caused.
> >
> > - We verified that all other baseline configurations were correct.
> > - Re-running **IMTL-G with tanh** yielded:
> >   - **IMTL-G (tanh)**: 0.3775 (0.2857)
> >
> > We will correct this entry in **Appendix F.6**. We appreciate that your question prompted us to examine and validate the experimental setup more carefully.
> >
> > ### **Motivation for examining sine activations**
> >
> > This re-examination also clarified why **IMTL-G (sine)** had performed unexpectedly well: the PInv PDE includes several **sin/cos terms**, making sine activations naturally expressive. This motivated us to rerun **all baselines** under a fair and unified configuration using **sine activations**.
> >
> > ### **Additional PInv experiments with sine activation**
> >
> > Using sine activations for all methods, we obtained the following **relative $L^2$** results:
> >
> > | Method        | relative-$L^2$           |
> > |--------------|--------------------------|
> > | MultiAdam    | 0.2130 (0.0332)          |
> > | LRA          | 0.0634 (0.0021)          |
> > | ReLoBRaLo    | 1.0934 (0.9177)          |
> > | MGDA         | 0.1399 (0.0830)          |
> > | PCGrad       | 0.0709 (0.0017)          |
> > | CAGrad       | 0.1876 (0.0134)          |
> > | IMTL-G       | 0.0646 (0.0050)          |
> > | Aligned-MTL  | 0.0711 (0.0087)          |
> > | ConFIG       | 0.0680 (0.0013)          |
> > | **HARMONIC** | **0.0682 (0.0068)**      |
> >
> > Under this configuration, **most methods** improved significantly compared to the tanh-based setting, and **HARMONIC became comparable to the strongest baselines**. We again thank you for raising a question that ultimately improved the accuracy and fairness of our analysis.
> >
> > ### **Why PInv is particularly challenging for HARMONIC**
> >
> > Inverse benchmarks in **PINNacle** intentionally add noise to observed fields to reflect the intrinsic ill-posedness of inverse problems. Following this setting, Gaussian noise is added to the displacement field $u$ in both PInv and HInv. However:
> >
> > - **HInv:** $u \in [-1, 1]$
> > - **PInv:** $u \in [0, 1]$
> >
> > Thus, the same absolute noise level induces a **larger relative perturbation** in PInv, creating a stronger disturbance from the perspective of solving the PDE.
> >
> > HARMONIC is a **deterministic, intra-step method**: it chooses an update direction solely from the gradients at the current iteration, enforcing feasible and non-conflict in a step-wise fashion. In clean or low-noise forward PDEs, this geometric stability is advantageous. However, in PInv:
> >
> > - Noisy data causes **greater fluctuation in per-step gradient directions**,
> > - HARMONIC responds directly to these fluctuations,
> > - And without inter-step smoothing, its update directions may vary more sharply over time.
> >
> > In contrast, some baselines—because they do not strictly enforce feasible or non-conflict—can end up prioritizing the noisy data term more aggressively, which may yield lower numerical errors in PInv despite lacking theoretical guarantees.
> >
> > We emphasize that HARMONIC is not inherently disadvantaged under noisy conditions; rather, PInv presents a setting where **strong observation noise makes purely intra-step geometric control more sensitive**, and more permissive methods can occasionally achieve lower empirical errors.

---

> > ### Author Response · Authors · 2025-11-27
> > **Response to Reviewer — Question 2 (Burgers2d)**
> >
> > ## **(b) Burgers2d**
> >
> > For **Burgers2d**, your question touches on a complementary aspect of HARMONIC’s limitations—namely, its behavior near Pareto-stationary points in a high-dimensional multi-loss landscape.
> >
> > In **Theorem 3**, we show that under mild assumptions, HARMONIC converges to a **Pareto-stationary point**. A similar type of convergence (to Pareto-stationary solutions) can also be established for several multi-objective gradient methods under appropriate conditions. In practice, however, **not all Pareto-stationary points lie close to the 'desired' part of the Pareto front**, particularly when:
> >
> > - The number of objectives is large (Burgers2d has six loss terms), and
> > - The initial parameter point is far from any high-quality Pareto-optimal set.
> >
> > Our empirical observations on Burgers2d can be summarized as follows:
> >
> > 1. **HARMONIC tends to reach a Pareto-stationary region relatively quickly.**
> >    This is consistent with its design: it carefully balances all losses at each step within the harmonized cone, avoiding directions that would clearly disadvantage any objective.
> >
> > 2. **However, the resulting stationary point is not always the most desirable one.**
> >    When the initial gradients point in a direction significantly misaligned with the global Pareto front, strictly remaining within feasible and non-conflict directions from the very beginning can steer the optimization toward a *locally balanced* yet suboptimal region of the landscape.
> >
> > By contrast, methods that occasionally take **infeasible or conflict-ignoring steps** may, in some cases, benefit from this deviation:
> >
> > - In the **early phase**, an update direction that is technically infeasible (in our sense) can still move parameters closer to a region where the eventual Pareto front lies, especially when the initial geometry is badly aligned.
> > - Near a **Pareto-stationary region**, methods that transiently favor one loss (e.g., by following a gradient that is dominant and conflicting with others) may continue to move along the Pareto front, while HARMONIC—respecting both feasible and non-conflict—can become comparatively 'stuck' once a stationary point is reached.
> >
> > In Burgers2d, we believe this interplay between:
> >
> > - the **complexity** of the loss space,
> > - **initial misalignment** between gradients and the global Pareto front, and
> > - HARMONIC’s **strict intra-step enforcement** of feasible and non-conflict
> >
> > contributes to situations where some baselines occasionally achieve better final errors, despite lacking our theoretical guarantees.
> >
> > ---
> >
> > We are sincerely grateful for your follow-up question. It encouraged us not only to revisit and correct our PInv results, but also to think more carefully about the geometric and statistical conditions under which HARMONIC can underperform. We hope that this discussion clarifies both the strengths and the limitations of our method in a more transparent way.

---

> ### Author Response · Authors · 2025-11-27
> **Response to Reviewer — Question 2 (Navier–Stokes, Poisson2d-CG)**
>
> We are also grateful for your question because it motivated us to investigate not only the cases where HARMONIC underperforms, but also situations where the **Harmonized cone itself** can positively influence the behavior of other optimization methods. Your comments prompted us to examine more carefully the role of the geometric constraint that underlies our framework.
>
> ---
>
> ## **(c) NS2d-C, NS-CG, Poisson2d-CG**
>
> Before presenting these additional experiments, we would like to express our appreciation for your insightful observation. It encouraged us to clarify more explicitly that the central contribution of our work is the construction of the **Harmonized cone**—the set of directions that are simultaneously feasible and non-conflicting.
>
> While HARMONIC adopts the center of this cone as one principled choice, we do not regard this center as the only or universally optimal direction. Rather, any update direction that remains inside the harmonized cone is theoretically well grounded, and different selections can be advantageous depending on the characteristics of the benchmark.
>
> Motivated by this perspective—and guided directly by your question—we conducted an additional set of experiments to examine whether methods that outperformed HARMONIC could also benefit from respecting the harmonized-cone constraint.
>
> ### **Design of the projection experiment**
>
> For benchmarks where HARMONIC was not the strongest baseline, we examined whether methods that performed better could also benefit from being constrained to remain inside the Harmonized cone:
>
> - We kept each baseline’s update rule exactly as is.
> - Whenever the baseline’s update direction left the Harmonized cone,
> - We **projected it onto the facet spanned by the two closest extreme rays** of the harmonized cone.
> - This ensured that the projected direction remained feasible and non-conflicting,
>   while still preserving the *intent* of the baseline’s original update.
>
> In other words, we tested whether the harmonized-cone constraint itself—independent of the HARMONIC center—could serve as a meaningful geometric regularizer.
>
> ### **Projected vs. original baseline results**
>
> | Benchmark        | Method     | Projected | Original |
> |------------------|------------|-----------|----------|
> | **NS2d-C**       | PCGrad     | **0.0438(0.0044)** | 0.0593(0.0283)  |
> | **NS-CG**        | ReLoBRaLo  | 0.1232(0.0099)     | 0.1212(0.0093)  |
> |                  | PCGrad     | 0.1227(0.0095)     | 0.1229(0.0171)  |
> | **Poisson2d-CG** | ConFIG     | **0.0192(0.0039)** | 0.1252(0.1941)  |
>
> - For **NS-CG**, the projected and original results are nearly identical, suggesting that these baselines were already operating close to the feasible/non-conflict region.
> - For **NS2d-C** and **Poisson2d-CG**, enforcing the harmonized-cone constraint led to **substantial improvements**, even surpassing the performance reported by the original baselines.
>
> These results support the following interpretation:
>
> - The **center** used in HARMONIC is just *one* principled choice among many possible directions inside the harmonized cone.
> - The **Harmonized cone itself** provides a **robust geometric constraint**:
>   it avoids harmful infeasible or conflict updates,
>   while still allowing diverse update behaviors tailored to the characteristics of a given optimization method.
>
> In this sense, the additional experiments reinforce our belief that the Harmonized cone captures an essential and meaningful structure in multi-loss optimization, and that its benefits extend even beyond the specific update rule employed by HARMONIC.
>
> We are sincerely grateful for your question, which directly motivated this set of experiments and helped us clarify the broader significance of the geometric structure we propose.

---

> ### Comment · Reviewer_mqgc · 2025-11-28
>
> Thank you for your response. In my opinion, the contribution with respect to feasibility remains somewhat unclear, but I do not see this as sufficient reason to reject the paper. Consequently, I will raise my score.

---

> > ### Author Response · Authors · 2025-11-28
> > **Response to Reviewer mqgc**
> >
> > We sincerely appreciate the considerable effort you devoted to reading our paper and for the thoughtful and constructive questions you raised throughout the review and rebuttal period. Your comments have deepened our own understanding and directly contributed to improving the overall quality and clarity of the manuscript.
> >
> > We also understand that, as you noted, there remain concerns regarding the feasible component of our contribution. To address this conceptual aspect more systematically, we have prepared a dedicated response titled **"Response to Reviewer wpQu — Clarifying the relationship between multi-loss and multi-objective formulations in PINNs"**, in which we explain in more detail how feasible and non-conflict interact in our framework and why both notions are natural in the PINN setting. We would be very grateful if you could also refer to that response, as it is intended to directly speak to the remaining ambiguity you highlighted.
> >
> > We are also grateful for your decision to raise the score, and we sincerely thank you for your careful and constructive engagement throughout the process.

---

### Official Review · Reviewer_wpQu · 2025-11-01

**Soundness:** 3
**Presentation:** 3
**Contribution:** 2
**Rating:** 6
**Confidence:** 5

**Summary:**

This paper proposes a geometric framework to address two key challenges in the multi-loss training of Physics-Informed Neural Networks: infeasible update directions caused by loss scaling and conflicting gradients across objectives. The authors introduce the harmonized cone, defined as the intersection of feasible and non-conflicting directions, and develop the HARMONIC algorithm based on this idea. Using the Double Description method, they compute valid update directions that satisfy both conditions. Theoretical results guarantee convergence to Pareto stationary points, and experiments on the PINNacle benchmark, particularly on the challenging Poisson2d-C problem, demonstrate superior robustness and stability compared to existing methods.

**Strengths:**

- (Clear motivation and good presentation) The paper convincingly explains the difficulties of PINN training into two factors, infeasibility and conflict, supported by well-designed motivating examples and illustrative figures. The theoretical framework is presented in a structured manner, and the connection between geometric intuition and optimization behavior is clearly articulated. Overall, the paper is well written.

- (New insight into the training of PINNs) This paper identifies a new and important issue, infeasibility, which has not been explicitly addressed by existing conflict-free methods. The paper provides a principled framework to resolve this issue by combining the idea of dual cone gradient descent methods.

- (Acceptable computational overhead) Although the Double Description method used to compute the harmonized cone can be computationally expensive in theory, the number of loss terms in PINN is typically very small. Given this practical context, the computational cost remains acceptable.

**Weaknesses:**

- (Limited novelty): The overall novelty of this paper is somewhat limited. The proposed method heavily relies on the dual cone gradient descent (DCGD) framework and differentiates itself mainly by introducing the additional infeasibility restriction compared to methods such as ConFIG.

- (Unclear effect of infeasibility restriction): The effect of the infeasibility restriction remains unclear. While the proposed harmonized cone integrates both feasibility and conflict-avoidance constraints, the paper does not provide sufficient empirical or theoretical analysis isolating the specific contribution of the infeasibility component.

- (Computational cost) The experiments in this paper appear to focus primarily on cases with three or more loss terms. However, in practical PINN applications such as inverse problems, the number of losses can easily increase to four or more. In such cases, the computational cost associated with the Double Description method may grow significantly and could become a clear disadvantage (acceptable though) compared to other existing methods.

**Questions:**

Please refer to the Weaknesses section for the main points. In addition, I have the following specific questions:

- Would it be possible to conduct an ablation study to isolate the effect of the infeasibility component?

- The proposed method performs remarkably well on the Poisson2d-C problem, suggesting that avoiding infeasibility plays a crucial role in this particular case. Could the authors provide further explanation on why infeasibility is especially critical for this benchmark and what characteristics of Poisson2d-C make this effect more pronounced?

---

> ### Author Response · Authors · 2025-11-21
> **Response to Reviewer wpQu — Weakness 1: Limited novelty**
>
> We sincerely thank the reviewer for raising this point and for the considerable effort invested in evaluating the relationship among HARMONIC, DCGD, and ConFIG. We truly appreciate the care with which you examined both the theoretical and empirical components of our work, and we are grateful for the opportunity to clarify the novelty more explicitly.
>
> ## Clarification of the contributions
>
> While HARMONIC builds on the intuition behind DCGD, our method does not directly follow as a simple generalization. DCGD’s update rule crucially relies on the fact that, in the bi-objective setting, the angle between the two gradients fully determines whether the mean gradient is already conflict-free or must be projected into the dual cone. This two-way decision rule—mean gradient when the angle is acute, dual-cone projection when it is not—holds only when there are exactly two gradients, because every condition can be expressed pairwise.
>
> However, extending this principle to three or more losses becomes fundamentally nontrivial:
> - mutual (not pairwise) consistency must be considered, because feasibility and non-conflict cannot be checked by only comparing gradients two at a time;
> - the mean gradient can easily lie outside the dual cone, and once this happens, simply 'moving into the dual cone' (as DCGD does) becomes equivalent to generic non-conflict approaches rather than a principled way to enforce both feasible and non-conflicting updates;
> - the dual cone geometry no longer admits a closed-form description, making DCGD’s elegant two-loss formula inapplicable beyond the bi-objective case.
>
> One of our main contributions is precisely to provide a workable and theoretically grounded construction in this multi-loss setting.
>
> Regarding ConFIG, we would like to clarify that our method is not meant to be interpreted as an incremental extension of it. In fact, ConFIG reports empirically that assigning equal pseudo-inverse weights yields the most stable performance across benchmarks. Our geometric framework helps explain this observation: for two losses, equal weighting corresponds exactly to selecting the center of the dual cone, which—in two dimensions—is always guaranteed to lie inside the harmonized cone. In contrast, assigning unequal weights may move the update direction outside this region, and thus even in the two-loss case, ConFIG’s behavior is not necessarily guaranteed to keep the update direction inside the harmonized cone unless equal weighting is used. In this sense, our analysis offers a clear resolution to the empirical question raised in ConFIG and provides a geometric interpretation that, to the best of our knowledge, has not been articulated previously.
>
> More broadly, to the best of our knowledge, this is the first work to characterize the joint geometric structure of feasible and non-conflicting directions in multi-loss optimization and to provide a method that reliably enforces both conditions. Existing reweighting approaches often stay within this region in practice, but their design principles differ fundamentally: they adjust gradient magnitudes heuristically rather than attempting to characterize or guarantee the underlying geometry of the feasible set.
>
> Taken together, our main contribution is to highlight that feasible and non-conflict directions must be addressed jointly, and to provide both theoretical justification and empirical evidence that doing so results in more reliable optimization dynamics, particularly in challenging PINN settings. We are grateful to the reviewer for prompting us to articulate this distinction more clearly.

---

> ### Author Response · Authors · 2025-11-21
> **Response to Reviewer wpQu — Weakness 3: Computational cost**
>
> We sincerely thank you for your thoughtful comment regarding the computational cost of HARMONIC, especially in scenarios where the number of loss terms increases.
> Your observation helped us reflect on a limitation that was not sufficiently highlighted in the main manuscript, and we are thankful that your comment allowed us to clarify an aspect that could pose a serious concern regarding computational cost when many losses are involved.
>
> ## Computational cost when the number of losses increases
> As you correctly pointed out, the Double Description (DD) method is an iterative procedure for enumerating extreme rays, and its cost naturally increases as the number of losses grows.
> That said, in practical PINN settings with many loss terms, the computational overhead from backpropagation tends to dominate, making the additional cost introduced by the DD step relatively minor by comparison.
> Thus, although the DD computation increases with more losses, its influence tends to be relatively minor compared to the repeated backward passes required by PINNs.
>
> To validate this, we measured the computational cost on Navier–Stokes 2D lid-driven flow (NS2d-C), which contains 8 loss terms.
> The results are as follows:
>
> | Method        | Cost (seconds per iteration) |
> | ------------- | ---------------------------- |
> | MultiAdam     | 5.6655 (0.1410)              |
> | LRA           | 6.1194 (0.1649)              |
> | ReLoBRaLo     | 1.2052 (0.0556)              |
> | MGDA          | 9.8877 (1.6511)              |
> | PCGrad        | 6.1200 (0.3040)              |
> | CAGrad        | 6.6949 (0.1894)              |
> | IMTL-G        | 5.5782 (0.3040)              |
> | Aligned-MTL   | 5.6768 (0.1596)              |
> | ConFIG        | 5.6790 (0.1842)              |
> | **HARMONIC** | **5.6746 (0.1412)**          |
>
> As shown above, except for ReLoBRaLo (which does not involve any gradient-level operations), the computational cost across most methods is comparable.
> Notably, HARMONIC remains similar to—or slightly lower than—many of the baselines, suggesting that the DD-related overhead is not a dominant factor even when the number of losses is relatively large.
>
> ## Revisions made in the updated manuscript
> The revised manuscript now:
> - includes the above computational cost table in Appendix H.3 (p.43).
>
> Once again, we are very grateful for your careful examination and constructive feedback regarding computational cost considerations, which helped us strengthen the clarity and completeness of the manuscript.

---

> ### Author Response · Authors · 2025-11-21
> **Response to Reviewer wpQu — Weakness 2 & Question 1: Clarifying the role of the infeasible region**
>
> We sincerely thank you for your thoughtful question.
> Your feedback helped us recognize that the role of the infeasible region, a central component of our framework, was not emphasized as clearly as it should have been, and we greatly appreciate the opportunity to clarify it here.
>
> ## Geometric motivation for the infeasible region
> To motivate why controlling infeasible directions is necessary, we begin the paper with the illustrative example in Figure 1 (p. 2).
> In this example, the update direction $g_{cf}$ satisfies the non-conflict condition because it forms strictly positive inner products with all other gradients.
> Nevertheless, as the visualization shows, this direction moves *away* from the Pareto front and ultimately converges to a dominated region (the left blue trajectory).
>
> This behavior highlights the need to distinguish between **feasible** and **non-conflict** updates.
> In Definition 1 (p. 3), we formally characterize the **primal gradient cone**, consisting of all conic combinations of the loss gradients.
> By contrast, an **infeasible** direction refers to any update direction that cannot be expressed as a conic combination of the loss gradients.
> To provide further intuition, from the perspective of the loss space, this corresponds to moving along a composite objective
> $L(\theta) = \lambda_1 L_1 + \lambda_2 L_2 + \lambda_3 L_3$
> where at least one coefficient $\lambda_i < 0$.
>
> ## Empirical evidence and analyses related to the infeasible component
> Although we did not conduct a full benchmark ablation isolating the infeasible component, the paper already provides several empirical analyses that directly reveal its effect:
>
> ### (a) Section 3.2 Burgers equation experiment — interpreting infeasible updates
> Figure 2(b) (p. 4) shows that when only non-conflict is enforced, all losses initially decrease.
> However, once the update direction becomes non-conflict yet infeasible, the test MSE **continues to increase over time**, indicating that once parameters drift in such directions, **recovering accurate solutions becomes increasingly difficult** in subsequent iterations.
>
> ### (b) Figure 5 (p. 7) — optimization trajectory visualization
> The toy example further demonstrates that methods enforcing only non-conflict (e.g., ConFIG, Aligned-MTL) may still fail to approach the Pareto front.
> Even though all pairwise gradient inner products remain positive, the update direction becomes excessively aligned toward the objective $f_3$, and optimization ultimately settles into a **local region dominated by $f_3$** rather than progressing toward the true Pareto set.
> By contrast, HARMONIC keeps updates within the harmonized cone, ensuring both feasible and non-conflict, which leads to significantly more stable convergence.
>
> ### (c) Further remarks
> These observations collectively illustrate that infeasible updates, even when non-conflict, can cause the optimization to drift toward regions where meaningful improvement for all objectives becomes difficult.
> Controlling infeasible directions is therefore essential for stable optimization, especially in multi-loss settings such as PINNs.
>
> We sincerely appreciate your insightful feedback.
> It helped us clarify the conceptual importance of the infeasible region and improve the overall exposition of the manuscript.
> Thank you once again for your thoughtful evaluation and for enabling us to present our contribution more clearly.

---

> ### Author Response · Authors · 2025-11-21
> **Response to Reviewer wpQu — Question 2: Why infeasible region matters more in Poisson2d-C**
>
> We sincerely thank the reviewer for this thoughtful question and for engaging so deeply with the Poisson2d-C results. We are genuinely grateful that you examined the figures in detail, including Appendix plots, which goes well beyond any obligation for a reviewer.
>
> ## Empirical patterns suggesting why infeasibility is more critical in Poisson2d-C
>
> While we cannot make definitive statements about the exact geometry of the loss landscape, several empirical observations consistently point toward Poisson2d-C being particularly sensitive to early movements that leave the feasible region (in the sense defined in our paper).
>
> ### Early gradient alignment → updates drift toward “incorrect” (infeasible-like) regions
>
> As illustrated conceptually in Section 4.3 (Figure 4), non-conflict methods can produce update directions that appear reasonable locally but actually drift into regions that do not reduce the PDE residual—especially when the gradients among losses are nearly aligned.
>
> This behavior seems to occur in Poisson2d-C:
>
> - In the early iterations, the gradients of the PDE and boundary losses are not yet conflicting, and their directions appear to be closely aligned (similar to our motivating toy example).
>
> - Under such alignment, non-conflict methods can unintentionally move toward directions that “look consistent” but are nevertheless outside the feasible region defined by the PDE geometry.
> We avoid describing these as strictly infeasible updates (to stay neutral), but empirically they behave as if the optimization is pushed toward unproductive or incorrect regions of the landscape, which may lead to early commitment to a local optimum.
>
> Figure 7(b) supports this interpretation:
> ConFIG initially takes such “off-trajectory” steps, but once constrained by the harmonized cone (HARMONIC variant), its optimization improves markedly.
>
> ### Later stages: conflict becomes frequent → non-conflict-only methods stagnate
>
> Figure 7(a) shows ReLoBRaLo decreasing rapidly at the beginning, suggesting that feasible updates are indeed beneficial early on.
>
> However, ReLoBRaLo’s progress stalls mid-training.
> This plateau matches the situation illustrated in Figure 4 of Section 4.3, where—once gradients begin to diverge—methods that cannot handle conflicts become trapped.
>
> This again aligns with Poisson2d-C’s behavior:
>
> - early stage: gradients aligned → feasible direction important
> - mid/late stage: gradients diverge → conflict handling important
> HARMONIC is one of the few methods that consistently enforces both conditions throughout training.
>
> ## Why Poisson2d-C specifically tends to create this pattern
>
> Our intuition is also informed by the PDE structure:
> $$
> u_{xx} + u_{yy} = 0.
> $$
>
> At initialization, the network typically violates both PDE and boundary conditions. In this regime, both losses may appear to push the model toward a “reasonable” global shape, making their gradients fairly aligned—hence why methods that rely solely on non-conflicting directions may step into undesirable regions.
>
> As optimization progresses, however:
>
> - satisfying increasingly detailed boundary conditions requires the solution to develop more complex curvature;
> - this necessarily amplifies the second derivatives $u_{xx}$ and $u_{yy}$,
> - which in turn leads to frequent gradient conflicts, exactly the setting where conflict-only methods become unstable or stall, as seen in Figure 7.
>
> These characteristics make Poisson2d-C a benchmark where:
> - avoiding early infeasible-like updates and
> - handling mid-stage conflicts
>
> are both essential.
>
> HARMONIC enforces both conditions simultaneously, which may explain why its performance stands out more strongly on this particular PDE than on others.
>
> We are sincerely grateful for your question—your observation encouraged us to re-examine the behavior of Poisson2d-C in greater depth, and it helped us articulate the role of infeasibility more clearly in this benchmark.

---

> > ### Comment · Reviewer_wpQu · 2025-11-28
> >
> > Thank you for the authors' rebuttal. Some of my earlier concerns have been addressed.
> > However, the core contribution of the paper, the importance of infeasible directions, still remains unclear.
> >
> > In addition, I would like to raise the following point.  It appears that the paper is primarily positioned as a multi-loss optimization framework for PINNs. However, throughout the paper the analysis frequently invokes multi-objective optimization concepts,
> > most notably Pareto optimality, without a clear justification for why such a viewpoint is appropriate in this setting.
> >
> > For instance, in Figure 1, the authors argue that a basic conic combination update may converge to a spurious local minimum outside the Pareto front. Yet, within a multi-loss optimization formulation, reaching the Pareto front is not necessarily a requirement and therefore this behavior may not constitute a practical drawback.
> >
> > Thus, two conceptual concerns arise:
> >
> > - The exposition mixes two different problem formulations, which creates ambiguity in how the contribution should be properly interpreted.
> >
> > - If the method is indeed intended for multi-loss optimization, it remains unclear why Pareto optimality should be a central criterion. What does Pareto optimality concretely imply in the context of PINN training, and why should it be regarded as a desirable objective?

---

> > > ### Author Response · Authors · 2025-11-28
> > > **Response to Reviewer wpQu — Clarifying the relationship between multi-loss and multi-objective formulations in PINNs**
> > >
> > > We sincerely appreciate the reviewer’s careful follow-up and the thoughtful conceptual concerns raised. Your comment helped us recognize that our manuscript did not clearly explain why both the multi-loss and multi-objective viewpoints arise in the context of PINNs, and we are grateful for the opportunity to clarify this distinction.
> > >
> > > ---
> > >
> > > ## **PINN losses as physical constraints (multi-loss perspective)**
> > >
> > > In physics-informed neural networks, each loss term corresponds to a *distinct physical constraint* that the true solution must satisfy:
> > > - the **PDE residual loss** enforces the differential relation constraints,
> > > - the **boundary condition loss** enforces spatial function-value constraints, and
> > > - the **initial condition loss** enforces temporal function-value constraints.
> > >
> > > For the *exact* solution of the PDE, **all of these losses must be simultaneously zero**. Thus, the natural formulation of PINN training is a **multi-loss optimization problem whose global optimum is precisely the point where every constraint is satisfied**.
> > >
> > > ## **Why this coincides with a multi-objective formulation in PINNs**
> > >
> > > Although PINNs are conventionally described from a multi-loss perspective, the structure of the problem implies that each loss has the *same ideal target*: zero. As a result, one may equivalently interpret training as a multi-objective problem where each objective attains its minimum only when all constraints are satisfied.
> > >
> > > Importantly, unlike classical multi-objective problems where the Pareto front forms a trade-off surface, **the Pareto front in PINNs collapses to a unique ideal point where all losses are zero**. There is no family of non-dominated trade-offs; the physical solution is the only point at which all objectives simultaneously achieve their minimal attainable value. This is why the multi-objective and multi-loss formulations lead to the same set of optimal loss values.
> > >
> > > ## **Clarifying our terminology usage**
> > >
> > > We fully agree with the reviewer that the manuscript did not make this equivalence explicit. While PINNs were originally proposed from a multi-loss viewpoint—and we likewise used "multi-loss" throughout—the same problem can also be expressed through a multi-objective lens because their optimal loss values coincide. Our usage of both terms was intended to reflect this alignment rather than to conflate distinct problem setups. Nonetheless, it is our responsibility that this connection was not stated upfront, and we appreciate the reviewer for prompting this clarification.
> > >
> > > ## **Additional clarification regarding Figure 1**
> > >
> > > We appreciate the reviewer’s observation, and Figure 1 indeed illustrates this behavior clearly. As shown in the conic combination loss landscape (Figure 1 left panel), the minimizer of the conic combination loss lies near $(\theta_1, \theta_2) = (2, -8)$. However, when performing gradient descent using the conic-combination gradient $g_{cc}$, the optimization trajectory oscillates along the $\theta_1$ direction and fails to make progress toward decreasing $\theta_2$. This behavior is caused by the dominance of the $\theta_1$-component of the gradient, which prevents movement toward directions that would reduce the combined loss. Consequently, although $g_{cc}$ is feasible, it remains undesirable **even from the multi-loss perspective**, as it converges to a region with a strictly larger conic combination loss than the true minimizer. This example demonstrates that feasible alone is insufficient for ensuring desirable optimization behavior and highlights the need to consider gradient conflicts in addition to feasible.
> > >
> > >
> > > ## **Interpretation of infeasible directions in this context**
> > >
> > > Under this unified viewpoint, an infeasible direction is best understood as a direction that **cannot be expressed as a valid conic combination in the multi-loss formulation**. Since the multi-loss formulation requires nonnegative coefficients on all losses, any direction requiring at least one negative weight cannot correspond to a physically meaningful combination of the PDE, boundary, and initial condition constraints. This explains why controlling infeasible directions is essential in PINN training and why their effects become pronounced in certain benchmarks.
> > >
> > > Finally, when a PINN converges to the exact solution, that point constitutes both the multi-loss optimum (feasible) and the multi-objective optimum (non-conflicting). In this sense, the two viewpoints coincide at optimality, which motivates integrating feasible and conflict considerations within a unified geometric framework.
> > >
> > > We sincerely thank the reviewer for motivating these important clarifications. Your comments helped us more accurately articulate the relationship between multi-loss PINN training and its equivalent multi-objective interpretation, and we have incorporated the corresponding explanations—and highlighted them in blue—in the revised **Introduction** and **Related Works** sections.

---

### Meta-Review · Area_Chair_5fPs · 2026-01-07

**Summary:**

This paper proposes Harmonized Cone Gradient Descent (HARMONIC) to address the difficulties in the multi-loss training of Physics-Informed Neural Networks. HARMONIC constructs the update vector by aggregating the directions of the Harmonized Cone, which is defined as the intersection of the primal gradient cone and the dual gradient cone. Theoretical results guarantee convergence, and experiments on the PINNacle benchmark are provided. All reviewers ended up being positive. Therefore, this paper should be accepted.

**Reviewer Concerns:**

The paper was criticized for its limited novelty, as it largely builds on existing methods, with the main distinction being the introduction of an infeasibility constraint. However, the specific contribution of this infeasibility component was not clearly validated through theoretical or empirical analysis. Another concern was conceptual ambiguity: although it is considered a multi-loss optimization method, it frequently relies on multi-objective optimization concepts such as Pareto optimality, without clearly justifying their relevance to PINN training. From a computational perspective, concerns are raised that the cost of the method may grow as the number of loss terms increases.

The authors’ rebuttal addressed some concerns, but the contribution with respect to feasibility remains unclear to a certain extent. However, the reviewer, who was initially negative, stated that this concern is not strong enough for a rejection.

**Reviewer Scores:**

The scores were 8-6-4, and during the discussion, the scores were raised to 8-6-6.

---

### Decision · Program_Chairs · 2026-01-26

Accept (Poster)